# Variable C/P composition of organic production and its effect on ocean carbon storage in glacial–like model simulations

Malin Ödalen[1], Jonas Nycander[1], Andy Ridgwell[2,3], Kevin I. C. Oliver[4], Carlye D. Peterson[2], and Johan Nilsson[1]

[1]Department of Meteorology, Bolin Centre for Climate Research, Stockholm University, 106 91 Stockholm, Sweden
[2]Department of Earth Sciences, University of California–Riverside, Riverside, CA 92521, USA
[3]School of Geographical Sciences, Bristol University, Bristol BS8 1SS, UK
[4]National Oceanography Centre, Southampton, University of Southampton, Southampton SO14 3ZH, United Kingdom

**Correspondence:** Malin Ödalen (malin.odalen@misu.su.se)

**Abstract.** During the four most recent glacial maxima, atmospheric $CO_2$ has been lowered by about 90–100 ppm with respect to interglacial concentrations. It is likely that most of the atmospheric $CO_2$ deficit was stored in the ocean. Changes of the biological pump, which are related to the efficiency of the biological carbon uptake in the surface ocean and/or of the export of organic carbon to the deep ocean, have been proposed as a key mechanism for the increased glacial oceanic $CO_2$ storage. The biological pump is strongly constrained by the amount of available surface nutrients. In models, it is generally assumed that the ratio between elemental nutrients, such as phosphorus, and carbon (C/P ratio) in organic material is fixed according to the classical Redfield ratio. The constant Redfield ratio appears to hold approximately when averaged over basin scales, but observations document highly variable C/P ratios on regional scales and between species. If the C/P ratio increases when phosphate availability is scarce, as observations suggest, this has the potential to further increase glacial oceanic $CO_2$ storage in response to changes in surface nutrient distributions. In the present study, we perform a sensitivity study to test how a phosphate–concentration dependent C/P ratio influences the oceanic $CO_2$ storage in an Earth system model of intermediate complexity (cGENIE). We carry out simulations of glacial–like changes in albedo, radiative forcing, wind–forced circulation, remineralisation depth of organic matter, and mineral dust deposition. Specifically, we compare model versions with the classical constant Redfield ratio and an observationally–motivated variable C/P ratio, in which the carbon uptake increases with decreasing phosphate concentration. While a flexible C/P ratio does not impact the model's ability to simulate benthic $\delta^{13}C$ patterns seen in observational data, our results indicate that, in production of organic matter, flexible C/P can further increase the oceanic storage of $CO_2$ in glacial model simulations. Past and future changes in the C/P ratio thus have implications for correctly projecting changes in oceanic carbon storage in glacial–to–interglacial transitions as well as in the present context of increasing atmospheric $CO_2$ concentrations.

## 1 Introduction

During the last four glacial maxima, atmospheric $CO_2$ (henceforth $pCO_2^{atm}$ was lowered by $\sim$ 90–100 ppm compared to the interglacials (Petit et al., 1999; Lüthi et al., 2008). Due to the difference in size between the oceanic, terrestrial and atmospheric

carbon reservoirs, where the oceanic reservoir is by far the largest with >90% of their summed carbon contents (reviewed by, Ciais et al., 2013), it is likely that most of the $CO_2$ that was removed from the atmosphere was stored in the glacial ocean. In addition, studies of paleoproxy records indicate that carbon storage in the glacial terrestrial biosphere was smaller compared to in interglacial climate (Shackleton, 1977; Duplessy et al., 1988; Curry et al., 1988; Crowley, 1995; Adams and Faure, 1998; Ciais et al., 2012; Peterson et al., 2014). During deglaciation, radiocarbon evidence indicate that $CO_2$ was rapidly released from the ocean back to the atmosphere (Marchitto et al., 2007; Skinner et al., 2010).

Numerous processes, both physical and biological, have been identified as possible contributers to increased glacial oceanic storage. As glacial climate was substantially colder than interglacial climate; the global averages of surface, and ocean temperature at the LGM are estimated to have been 3–8 °C and 2.0–3.2 °C colder, respectively, than the pre–industrial (Stocker, 2014; Headly and Severinghaus, 2007; Bereiter et al., 2018). Due to the temperature effect on solubility, a colder ocean can hold more carbon. However, glacial changes in salinity partly offset the temperature effect on solubility (reviewed by, Kohfeld and Ridgwell, 2009). A colder climate is also drier, and the dry conditions led to increased glacial dust deposition compared to interglacial climate (Mahowald et al., 2006). It has been hypothesised that the addition of dust contributed to increased iron availability in the surface ocean, and that this contributed to a strengthening the biological sequestration of carbon in the glacial ocean compared to interglacials (Martin, 1990). The addition of iron would allow for more complete usage of other nutrients in regions where iron is limiting for biological production. Such strengthening of the retention of biologically–sourced carbon in the deep ocean (or the so–called biological pump), through changes in nutrient availability, light conditions, and/or ocean circulation, has long been considered an important player in the glacial increase in ocean $CO_2$ storage (Broecker, 1982a; Sarmiento and Toggweiler, 1984; Archer et al., 2000; Sigman and Boyle, 2000). Other studies have pointed to changes in carbonate preservation in coral reefs and deep–sea marine sediments (Berger, 1982; Broecker, 1982a; Archer and Maier-Reimer, 1994). It is likely that reduced ventilation of the deep water, through changes in ocean circulation and expanded sea ice cover acting as a barrier for air–sea gas exchange, contributed to increasing the glacial ocean carbon retention (e.g., Boyle and Keigwin, 1987; Duplessy et al., 1988; Stephens and Keeling, 2000; Marchitto and Broecker, 2006; Adkins, 2013; Menviel et al., 2017; Skinner et al., 2017). Model studies by (Menviel et al., 2017) show that reduced Southern Hemisphere westerly winds produce reduced ventilation of Antarctic Bottom Water (AABW) in line with evidence from proxy records of $\delta^{13}C$ . In addition, it has been shown that strengthening of the winds over the Southern Ocean was a likely contributer to deglacial outgassing of $CO_2$ from the ocean to the atmosphere (Mayr et al., 2013). Extensive summaries of the processes responsible for high glacial ocean carbon storage, and examples of their interactions, are given by Brovkin et al. (2007); Kohfeld and Ridgwell (2009); Hain et al. (2010); Sigman et al. (2010). Despite the efforts of identifying the responsible processes, models have been struggling to achieve the full lowering of $pCO_2^{atm}$ expected for a glacial.

In this paper, we focus on the biological pump and how it responds to glacial–like changes in climate. Our aim is to investigate how the level of simplification of the biological carbon uptake in an Earth system model model may affect the glacial drawdown of $pCO_2^{atm}$. Most biogeochemical models used in glacial climate studies have a simple representation of biological production, which assumes that carbon, C, and inorganic nutrients such as phosphorus, P, are taken up in fixed proportion to

each other. This is modelled using the average ratio of C/P of the ocean organic matter originally observed by Redfield (1963) (C/P = 106/1), or adjustments to these suggested in follow–up studies (Takahashi et al., 1985; Anderson and Sarmiento, 1994).

We investigate whether allowing for a flexible C/P stoichiometric ratio increases model ocean $CO_2$ storage in a glacial–like climate. This possibility was suggested in studies by Broecker (1982b), Archer et al. (2000) and Galbraith and Martiny (2015), but the implications of Redfield versus flexible C/P for glacial ocean carbon storage has not previously been tested in an Earth system model. This type of non–Redfieldian dynamics were applied in the model used in Eggleston and Galbraith (2018) and Galbraith and de Lavergne (2018), but their results were not analysed in terms of difference from a Redfield model version. In addition, Buchanan et al. (2018) explored the importance of dynamic response of ocean biology, such as flexible stoichiometry, for modelled ocean biogeochemistry in pre–industrial simulations. They found that the dynamic response was fundamental for stabilising the response of ocean DIC to changes in the physical circulation state.

We conduct a sensitivity study, where we apply glacial–like changes in radiative forcing, albedo, wind–forced circulation, remineralisation depth and dust, separately and in combination, to an interglacial control state. Here, changes in radiative forcing and albedo serve to cool the climate, to mimic glacial temperature and ice conditions. The surface wind stress is reduced in the polar regions, in order to pursue reduced AABW ventilation as suggested by paleoproxy evidence (Menviel et al., 2017; Skinner et al., 2017). Ocean cooling reduces the degradation rate of sinking particulate organic carbon, which increases the average depth of remineralisation of organic carbon (Matsumoto, 2007). Drier glacial climate resulted in increased dust deposition, and thereby iron flux, to the ocean (Martin, 1990). All these changes act to increase ocean carbon storage and thereby reduce $pCO_2^{atm}$. The applied changes are not expected to induce a full glacial maximum model state, but they allow us to explore several important effects on the biological and solubility carbon pumps, and produce a state with glacial–like climate conditions.

We apply the perturbations in two different versions of the Earth system model cGENIE; an original version using fixed Redfield stoichiometry of C/P (Ridgwell et al., 2007) plus a co–limitation by iron (Tagliabue et al., 2016) for biological production, and a modified version using the non–Redfieldian, nutrient concentration dependent, C/P stoichiometry suggested by Galbraith and Martiny (2015) and iron co–limitation.

We show that flexible C/P stoichiometry allows a larger glacial ocean $CO_2$ storage, as predicted by the box–model study of Galbraith and Martiny (2015), and that flexible stoichiometry has the largest impact for perturbations in remineralisation depth and dust forcing. Additionally, we show that flexible stoichiometry allows for increased ocean carbon storage without decreasing the storage of preformed nutrients in the deep ocean.

## 2 Methods

### 2.1 Model description

cGENIE is an Earth system model of intermediate complexity, with a 3D frictional–geostrophic ocean ($36 \times 36$ equal area horizontal grid, 16 depth levels), 2D energy–moisture balance atmosphere with prescribed wind fields, interactive atmospheric chemistry and ocean biogeochemistry. Model code and user handbook can be found in the cGENIE GitHub repository (cGENIE

GitHub repository, 2019). We run a version of cGENIE with the same phosphorus plus iron (Fe) co–limitation scheme as used in the iron cycle model inter–comparison study of Tagliabue et al. (2016). The model branch enabled for use with flexible C:P ratios (see 2.2) is tagged as release v0.9.5, and the model configurations used in this paper are included in this release (cGENIE release v0.9.5, 2019, see Code availability for details).

## 2.2  Stoichiometry

In the original version of the cGENIE Earth system model (Ridgwell et al., 2007), as well as in the version of Tagliabue et al. (2016), the stoichiometric ratios are based on Redfield (1963). Thus, there is a fixed relationship between the number of moles of the elements that are taken up (positive) or released (negative) during production of organic matter in the ocean. This relationship is $P : C : O_2 = 1 : 106 : -138$, where $O_2$ is dissolved oxygen (nitrogen is assumed only implicitly for the purpose of accounting for organic matter creation and remineralisation related alkalinity transformations in the ocean (Ridgwell et al., 2007)). An exception is iron, where Fe:C varies as a function of iron availability as described in Watson et al. (2000).

Although the average elemental composition of organic matter in the ocean is close to the Redfield ratios, the stoichiometry of production of new organic material has shown high in–situ variability. Variability occurs between species, but also within the same species, and has been shown to depend on environmental factors such as nutrient availability, water temperature and light (e.g., Le Quéré et al., 2005b; Galbraith and Martiny, 2015; Yvon-Durocher et al., 2015; Tanioka and Matsumoto, 2017; Garcia et al., 2018a; Moreno et al., 2018). We test the importance of this variability for glacial ocean $CO_2$ storage by running the same experiments with the fixed Redfield stoichiometry version of cGENIE and with a model version where we have implemented the linear regression model presented by Galbraith and Martiny (2015) (Eq. 1). These two model versions are henceforth denoted $RED$ and $GAM$, respectively.

The flexible stoichiometry in $GAM$ depends on the ambient concentration of dissolved phosphate ($[PO_4]$) in the water:

$$P : C = 1 : \left(\frac{[PO_4]}{144.9 \mu mol L^{-1}} + 0.0060\right)^{-1}. \tag{1}$$

This relation shows that, when $[PO_4]$ is low, organisms bind more $C$ per atom of $P$ than they do under high $[PO_4]$ conditions (Fig. 1). Eq. 1 is applied in cGENIE in the calculations of biological C uptake at the surface ocean based on the surface concentration of $PO_4$. In Section 3.1.2, Eq. 1 is also used to translate model surface $PO_4$ fields to the corresponding surface C/P ratios for the organic matter produced in each grid cell. Note that, while we change the ratio C/P, the ratio C/O$_2$ remains the same in all experiments. As a result, the P/O$_2$ ratio changes between experiments.

## 2.3  Experiments

We start all experiments from an interglacial/modern control state, which has been run for 10,000 years to steady state, using either Redfield ($Ctrl_{RED}$) or variable ($Ctrl_{GAM}$) stoichiometry. The control states have a prescribed $pCO_2^{atm}$ of 278 ppm and the same climate (Table 1), but due to the differences in C/P, they have different ocean carbon inventories (Table S.1.). In $Ctrl_{GAM}$, the export flux of organic matter (see Ridgwell et al., 2007) has a global average C/P composition of 121/1 and

thus the global ocean carbon storage is larger than in $Ctrl_{RED}$. This also suggest that a perturbation, which increases ocean storage of P through the biological pump, could cause storage of 15 (i.e. 121-106) more carbon atoms in simulations using GAM compared to RED, simply because the average composition of the formed biological material is different. To distinguish between the role of the flexibility of the stoichiometry and the change in the mean composition of organic material, we add a

control state with fixed stoichiometry where C/P = 121/1 (henceforth denoted $Ctrl_{121}$).

In order to explore the effects of variable stoichiometry, we make a sensitivity study where we apply changes to boundary conditions, individually and in combination (see Section 2.3.3), that may be representative of changes that occurred during glacial periods. All experiments are listed in Table 1.

The applied changes in boundary conditions are

– physical perturbations (colder climate):

       – radiative forcing corresponding to $LGM$ $CO_2$ = 185 ppm

       – zonal albedo profile representative of $LGM$ (calculated from the LGM climate simulation of Davies-Barnard et al., 2017)

   – physical perturbations (weaker overturning):

15        – reduced wind forcing over the Southern Ocean (Lauderdale et al., 2013, e.g.,) and north of 35°N (see Section 2.3.1).

   – biological perturbations:

       – changed remineralisation length scale (e.g., Matsumoto, 2007; Chikamoto et al., 2012; Menviel et al., 2012)

       – increased dust forcing, as simulated for $LGM$ (regridded from Mahowald et al., 2006)

By applying the above perturbations, we aim to approach, but not fully resolve, some of the characteristics of the Last Glacial

Maximum (LGM) ocean, which appears to have had a global average ocean temperature ($\overline{T_{oce}}$) 2.57 ±0.24 °C colder than the Holocene (Bereiter et al., 2018), a weakly ventilated deep ocean (e.g., Menviel et al., 2017) and a more efficient biological pump (e.g., Sarmiento and Toggweiler, 1984; Martin, 1990; Sigman and Boyle, 2000). We also aim to increase carbon retention in the deep ocean (Muglia et al., 2018).

The physical perturbations serve to achieve a colder climate ($\overline{T_{oce}}$ cools by 2.1°C c.f. $Ctrl$, thus 80 % of the observed

2.6°C, see Section3.2.1), and weaker overturning (see Section 3.2.2 and Fig. 2) with a longer residence time of the Antarctic Bottom Water (AABW) cell compared to $Ctrl$. Colder conditions achieve a stronger solubility pump, thereby strengthening the retention of carbon in the deep ocean. As the physical perturbations affect the ocean circulation and temperature, they thereby affect the nutrient distribution, and the rates of nutrient upwelling and biological growth (slower growth in colder water). They thereby affect the biological productivity (Ridgwell et al., 2007).

The biological perturbations serve to achieve a more efficient biological pump, which is connected with increased retention of nutrients and carbon in the deep ocean, and lower surface nutrient concentrations in productive regions (see Sections 3.2.3

and 3.2.4). With flexible stoichiometry, lower surface nutrient concentrations results in a higher C/P ratio, further increasing the export production, and therby the carbon retention in the deep ocean. In our experiments, we show that the flexible stoichiometry amplifies the response of the biological pump to both physical and biological perturbations.

The perturbations and the experiments are described in detail in Sections 2.3.1–2.3.3.

### 2.3.1 Physical perturbations

We change the physical conditions for climate by changing radiative forcing and albedo to LGM–like conditions and denote these changes $LGMphy$. We set the radiative forcing in the model to correspond to an atmosphere with 185 ppm $CO_2$ instead of 278. However, we allow the $pCO_2^{atm}$ to freely evolve (starting from the value of 278 ppm of the $Ctrl$ state atmosphere) in response to the cooler climate. For albedo, we apply a zonal $LGM$ albedo profile (calculated from the LGM climate simulation of Davies-Barnard et al., 2017). Assumptions of a simple zonal profile, instead of a 2D field re–gridded from PMIP LGM simulations, allows for a better consistency with the original zonal mean albedo profile developed for the modern configuration of GENIE (Marsh et al., 2011). Together, the changes in radiative forcing and albedo causes the global ocean average temperature ($\overline{T_{oce}}$) to decrease by 2.1°C compared to $Ctrl$ (see Section3.2.1).

To achieve a longer residence time of the AABW water mass, and an associated increase in carbon and nutrient retention, we apply weaker winds (denoted $WNA \times 0.5$). We use the Southern Ocean wind profile of Lauderdale et al. (2013), where the peak westerly wind strength at 50°S has been halved compared to the control state (see Section 2.3.3). The winds north and south of the peak are reduced accordingly to give a continuous profile (see Fig. 2 a of Lauderdale et al., 2013). The result is a weaker overturning (see Table 2) and a longer residence time of the AABW as may be expected for the glacial ocean (Menviel et al., 2017; Skinner et al., 2017) (see also Section 4.5). Thus, this approach is justifiable in a model of reduced complexity. However, there are studies suggesting that the Southern Ocean winds may in fact have been stronger during glacial times(Sime et al., 2013; Kohfeld et al., 2013; Sime et al., 2016). To avoid an expansion of the NADW overturning cell that would be inconsistent with the glacial ocean (Duplessy et al., 1988; Lynch-Stieglitz et al., 1999; Curry and Oppo, 2005; Marchitto and Broecker, 2006; Hesse et al., 2011), winds north of 35°N are also gradually reduced so that the wind strength north of 50°N is reduced by half compared to the control state. In cGENIE, gas transfer velocities are calculated as a function of wind speed (described in Ridgwell et al., 2007), and following Wanninkhof (1992). Consequently, weaker winds also lead to reduced gas exchange with the atmosphere.

### 2.3.2 Biological perturbations

In the ocean, phytoplankton growth rates and remineralisation of particulate organic carbon are processes that both work more slowly at colder temperatures (Eppley, 1972; Laws et al., 2000). Cooling of the ocean would thus lead to decreased production of particulate organic matter (POC), and simultaneously to a slower degradation of POC, with competing effects on export production (i.e. the amount of C captured by primary production that leaves the surface ocean without being remineralised) (Matsumoto, 2007). However, Matsumoto (2007) shows that the effect of slower remineralisation dominates the effect on export production. It has therefore been hypothesised that the cooling of the glacial ocean led to a deepening of the remineralisation

length scale (henceforth denoted RLS) in the ocean, and thereby more efficient retention of organic carbon in the deep ocean (Matsumoto, 2007; Chikamoto et al., 2012), which in turn caused a lowering of $pCO_2^{atm}$. Menviel et al. (2012) find that such a deepening results in model changes in export production in poor agreement with paleo–proxies, while Chikamoto et al. (2012) find improved model agreement with the glacial proxy records of export production and stable carbon isotopes for temperature–dependent growth rates and remineralisation. Deeper remineralisation also results in increased nutrient retention in the deep ocean, thus causing changes in surface nutrient fields and in C/P ratios of $GAM$. We test the effect of changes in RLS by multiplying the model default RLS by a factor Y ($RLS \times Y$, see Section 2.3.3 ).

Increased dust forcing leads to increased iron ($Fe$) availability. This allows for increased productivity (and hence more efficient usage of other nutrients) in the high-nutrient, low-chlorophyll (HNLC) regions in the North Pacific, Equatorial Pacific and Southern Ocean, where iron (Fe) is the limiting micronutrient (Martin, 1990; Moore et al., 2013). The variable stoichiometry in $GAM$ is expected to be influential if the concentrations of $P$ decrease in such regions as a result of increased Fe availability. This process may hence be of importance in a glacial scenario where dust forcing increases as a result of the drier conditions (Martin, 1990; Moore et al., 2013). We apply the regridded LGM dust fields of Mahowald et al. (2006) and denote this change $LGMdust$.

### 2.3.3 Sensitivity experiments and combined simulations

In the sensitivity study, for each of the three C/P parametrisations, we first change one forcing at a time (see Table 1). We run simulations where we apply individually the LGM boundary conditions for radiative forcing ($LGMrf$), albedo ($LGMalb$) and dust ($LGMdust$) and one simulation with halved wind stress near the poles (-50 and 50°N) ($WNS \times 0.5$). For the remineralisation length scale (RLS), we test a range of values of the multiplication factor $RLS \times fr$, where $fr = \{0.75, 1.25, 1.75\}$. This allows us to test the sensitivity to deep ocean retention of organic carbon.

We then run simulations where we combine several changes in forcing. We get a colder climate simulation ($LGMphy$) by combining $LGMrf$ and $LGMalb$. In simulation $Acomb$, we combine $LGMrf$, $LGMalb$, $LGMdust$, and $RLS \times 1.25$ (see Table 1). Kwon et al. (2009) show that small changes in remineralisation depth can cause substantial changes in $pCO_2^{atm}$. With the RLS deepening of 25%, we keep the corresponding changes in $pCO_2^{atm}$ from exceeding the $\sim 20 - 30$ ppm obtained in other studies (Matsumoto, 2007; Menviel et al., 2012). We finally run a glacial–like simulation $GLcomb$ (see Section 3.3), which is similar to $Acomb$ but also includes the change in wind stress $WSN \times 0.5$. The achieved $GLcomb$ model state does not represent a full glacial maximum state, but is more glacial–like compared to the control state; it has a colder climate (see Table 2), reduced deep ocean ventilation and more carbon retention in the deep ocean.

### 2.4 Observations

For comparison and validation of model results, we use records of ocean state variables from observations of modern data and proxy observations from the LGM.

Modern data of ocean temperature, oxygen and nutrients are retrieved from the World Ocean Atlas 2018 (Locarnini et al., 2018; Garcia et al., 2018c, b) and we use the proxy estimates of LGM ocean temperature from (Bereiter et al., 2018). Average

modern day strength of the Atlantic meridional overturning circulation (AMOC) is estimated by McCarthy et al. (2015) from the RAPID-MOCHA array at 26 °N.

We use model–data comparison of benthic $\delta^{13}C$ to assess the statistical similarity (correlation) between both the model control state and glacial–like state (see Section 2.3) to benthic $\delta^{13}C$ data representing the Late Holocene (0-6 ka, HOL) and Last Glacial Maximum (19-23 ka, LGM), respectively Peterson et al. (2014). Locations of core sites can be seen in Fig. S.1 (see also Fig. 1 in Peterson et al. (2014)). Note that LGM benthic $\delta^{13}C$ is more $^{13}C$-depleted than the Holocene due to the addition of $^{13}C$-depleted terrestrial carbon to the glacial ocean (Shackleton, 1977; Curry et al., 1988; Duplessy et al., 1988), which is not simulated in our model experiments. Therefore, to compare our glacial–like simulations ($GLcomb$) to LGM observations, we subtract a Holocene-LGM global average difference of 0.32 ‰ (Gebbie et al., 2015) from the $GLcomb$ experiments. Gebbie et al. (2015) state that the wide range of error for the estimate of glacial–to–modern change in benthic $\delta^{13}C$ of of 0.32 $\pm$0.20‰ suffers from a lack of observations in all ocean basins but the Atlantic. Therefore, we place more emphasis on the results of the model–data comparison in the Atlantic than in the Indo–Pacific sector.

## 2.5 Nutrient utilisation efficiency

The extent to which biology succeeds to use the available nutrients can be determined by calculating the nutrient utilisation efficiency $\overline{P^*}$ (Ito and Follows, 2005; Ödalen et al., 2018),

$$\overline{P^*} = \frac{\overline{P_{rem}}}{\overline{P_{tot}}}, \tag{2}$$

which is the fraction of remineralised ($P_{reg}$) to total ($P_{tot}$) nutrients (in this case, $PO_4$) in the ocean. Overlines denote global averages. Remineralised nutrients have been transported from the surface to the interior ocean by the biological pump, and $P_{rem}$ is given by

$$P_{rem} = P_{tot} - P_{pre}. \tag{3}$$

Here, $P_{pre}$ is preformed $PO_4$ – the concentration of $PO_4$ that was present in the water parcel as it sank, thus the fraction that was not used by biology in the surface ocean. In cGENIE, the concentration of preformed tracers is set in the surface ocean and then passively advected through the ocean interior (Ödalen et al., 2018). The biological pump also captures carbon, and a similar relationship can be used for concentrations of DIC, where

$$DIC_{rem} = DIC_{tot} - DIC_{pre}. \tag{4}$$

$DIC_{rem}$ is used to compute the ocean storage of remineralised acidic carbon ($AC_{rem}$, see Appendix A ). $AC_{rem}$ is biological carbon that entered the ocean in the form of $CO_2$ in soft tissue (as opposed to carbonates in hard tissue), measured independent of oxygen consumption and/or remineralised phosphate.

In cGENIE, $P_{pre}$ and $DIC_{pre}$ are modelled as passive tracers (Ödalen et al. (2018)). Hence, we can use the model output for $P_{pre}$ in Eq. 3 to compute $\overline{P^*}$ (Eq. 2).

In a model with fixed Redfield ratio, $\overline{P^*}$ determines the effect of the biological pump on $pCO_2^{atm}$. For example, it has been found that a higher $\overline{P^*}$ in the initial state gives a lower potential for drawdown of $pCO_2^{atm}$ in response to similar perturbations (Marinov et al., 2008; Ödalen et al., 2018). However, with variable stoichiometry this is no longer true, since the amount of carbon retained in the deep ocean is not necessarily proportional to $\overline{P^*}$.

## 3 Results

### 3.1 Control states

#### 3.1.1 Ocean temperature and circulation

As the three control states, $Ctrl_{RED}$, $Ctrl_{GAM}$ and $Ctrl_{121}$ are driven by the same physical forcings and have the same $pCO_2^{atm}$, they have the same ocean circulation pattern (Fig. 2 a, c, e and Table 2, Table S.2) and climate (exemplified by global ocean average temperature ($\overline{T_{oce}}$) in Table 2 and Table S.2). The surface ocean nutrient fields are fairly similar, with small differences due to the different C/P parametrisations (compare Fig. 3 a, and Fig. S.2). The strength of the Atlantic meridional overturning circulation (AMOC), diagnosed in the model as the maximum of the Atlantic meridional overturning streamfunction deeper than 1000 m, is 14 Sv ($1Sv = 1 \cdot 10^6 m^3 s^{-1}$) in all control states (Table 2, Table S.2). Results from the RAPID-MOCHA array at 26 °N suggest an average AMOC strength of $17.2 \pm 0.9$ Sv (McCarthy et al., 2015), thus our control state AMOC is a little bit weaker than in present day climate. The observational estimate for $\overline{T_{oce}}$ according to the World Ocean Atlas 2018 (Locarnini et al., 2018) is 3.49 °C, thus comparable to the 3.56 °C of our $Ctrl$ simulations. The surface nutrient concentrations of our control state $Ctrl_{GAM}$ (Fig. 3 a) compare reasonably well with observed surface ocean concentrations of PO$_4$ (Fig. 3 c), with some underestimation in the Pacific equatorial region, the North Pacific ocean, and the Labrador Sea. The agreement with observations is better for $Ctrl_{GAM}$ than for $Ctrl_{RED}$ (Fig. S.2).

#### 3.1.2 Surface nutrient distribution and C/P ratios

In Fig. 3 we see that surface $PO_4$ fields (left hand column) and the corresponding fields of surface C/P ratios (as given by Eq. 1, right hand column) of $Ctrl_{GAM}$ (panels a, b) and of observations (panels c, d) are similar in their pattern as well as in the magnitudes of values. Note that high concentrations of PO$_4$ correspond to low C/P ratios, and vice versa. The highest observed $PO_4$ concentrations in the Northern and equatorial Pacific are not fully reproduced by the model, but the pattern is well reproduced. In the surface C/P field of $Ctrl_{GAM}$ (Fig. 3 b) we see the signature of very high nutrient concentrations in the Southern Ocean ($> 1 \mu mol L^{-1}$, Fig. 3 b) as a band of low ratios, with the most extreme values near the Antarctic continent, as seen in observations (Fig. 3 d).

The nutrient utilisation efficiency $\overline{P^*}$ (Eq. 2) in the three control states differs by a few percent; 0.43, 0.46 and 0.42 in $Ctrl_{RED}$, $Ctrl_{GAM}$ and $Ctrl_{121}$ respectively (Table 2). The fraction of $DIC_{rem}$ in $DIC_{tot}$ (see Section 2.5, Eq. 4, Table S.1) is 0.065, 0.077 and 0.072 in $Ctrl_{RED}$, $Ctrl_{GAM}$ and $Ctrl_{121}$ respectively.

### 3.1.3   Ocean dissolved $O_2$

The most apparent difference between $Ctrl_{RED}$ and $Ctrl_{GAM}$ is in deep ocean oxygen concentrations, where the global ocean average dissolved $O_2$ concentration $(\overline{O_2})$ in $Ctrl_{GAM}$ ($144 \mu mol kg^{-1}$) is lower than in $Ctrl_{RED}$ and $Ctrl_{121}$ (166 and 152 $\mu mol kg^{-1}$, respectively). Compared to observations (World Ocean Atlas 2013, Fig. 4 a–b), both $Ctrl_{RED}$ (Fig. 4 c–d) and $Ctrl_{GAM}$ (Fig. 4 e–f) agree reasonably well with the real ocean. $Ctrl_{GAM}$ appears to capture better than $Ctrl_{RED}$ the equatorial oxygen minimum in the Atlantic basin, but goes too low in the North Pacific. In $Ctrl_{GAM}$ (Fig. 4 f), the North Pacific

is markedly lower in oxygen than in $Ctrl_{RED}$ (Fig. 4 d) and even anoxic in the oxygen minimum zone (OMZ). This should be kept in mind when analysing the oxygen sections of the glacial–like states $GLcomb_{RED}$ (Fig. 4 g–h) and $GLcomb_{GAM}$ (Fig. 4 i–j). Global averages for dissolved $O_2$ are given in Table 2.

### 3.1.4   Ocean $\delta^{13}C$

By comparing $Ctrl_{RED}$ $\delta^{13}C$ and Holocene (0–6 ka, HOL) benthic $\delta^{13}C$ values, we estimate a global model–data correlation

of 0.78 (Table S.3). The modern–day Atlantic Ocean has a distinctive spatial $\delta^{13}C$ pattern (Fig. 5 a, Fig. S.1) with $^{13}C$-enriched values in the intermediate depth (<2 km) North Atlantic and Nordic Seas and $^{13}C$-depleted values in the deep (>2.5 km) South Atlantic. While the model produces a weaker gradient than the observed HOL Atlantic Ocean (corr. 0.50, Table S.3) the model correlates well with Eastern Atlantic $\delta^{13}C$ records (Fig. S.1). For the Indo–Pacific, the weaker benthic $\delta^{13}C$ gradient is well represented by the model (Fig. 5 e). This pattern emerges mainly due to $^{13}C$-depleted, biologically sourced carbon that is

accumulated in the weak circulation region of the interior North Pacific (Matsumoto et al., 2002). However, Indo–Pacific $\delta^{13}C$ values of $Ctrl_{RED}$ are overall lower than the HOL observations. The overall model–data correlation for the Indo–Pacific is 0.39 (Table S.3). Comparing the control states of the $RED$ and $GAM$ model versions, $\delta^{13}C$ patterns (Fig. 5 a, e and Fig. S.3 a, e) and model–data correlations with HOL observations (Table S.3) are similar between the model versions, with somewhat lower correlations for $GAM$.

## 3.2   Sensitivity experiments

The applied changes listed in Table 1 cause changes in ocean characteristics such as overturning circulation, temperature, surface nutrient distributions and biological productivity, which result in changed $pCO_2^{atm}$. The resulting steady state global average values for temperature $(\overline{T_{oce}})$, dissolved oxygen $(\overline{O_2})$, and nutrient utilisation efficiency $\overline{P^*}$, as well as the maximum and minimum of the Atlantic meridional overturning streamfunction, are listed in Table 2.

### 3.2.1 Radiative forcing and albedo

In the simulations where radiative forcing and albedo are changed to represent LGM conditions ($LGMrf + LGMalb = LGMphy$), the reductions in $pCO_2^{atm}$ are similar in the $RED$ and the $GAM$ model versions. In $LGMphy$, the resulting $pCO_2^{atm}$ is 245.4 and 244.9 ppm respectively, thus a reduction of 33 ppm compared to the $Ctrl$ 278 ppm (Fig. 6). Here, variable C/P does not impact the results, because changes in the surface nutrient distribution (Fig. 7 a), and the associated changes in C/P (Fig. 8 a), are limited to very high latitudes where productivity is already low in the control state, due to low temperatures and a lack of light and iron. The drawdown of $pCO_2^{atm}$ can mainly be attributed to the increase in solubility carbon ($C_{sat,T}$) due to ocean cooling, and to an increase in sea ice, which prevents air–sea gas exchange and therefore causes an increase in disequilibrium carbon ($C_{dis}$) (Ödalen et al., 2018). Ocean cooling amounts to 2.1°C in $LGMphy$ compared to $Ctrl$ ($\overline{T_{oce}}$ in Table 2). In cGENIE, the increase in $C_{sat,T}$ associated with ocean cooling corresponds to $\sim 7\pm1.5$ ppm $°C^{-1}$ (Ödalen et al., 2018, supplementary Fig. S.1). Thus, in $LGMphy$ $C_{sat,T}$ and $C_{dis}$ should contribute roughly 40 and 60 % respectively of the change in $pCO_2^{atm}$. Note that cGENIE underestimates the true effect of ocean cooling on solubility, due to a temperature restriction on the solubility constants (Ödalen et al., 2018). This temperature restriction limits solubility from changing below 2.0 °C. In this case, the solubility effect on $pCO_2^{atm}$ of reducing $\overline{T_{oce}}$ by 2.1 °C (from 3.6 °C to 1.5°C, $\sim 15$ ppm) is thus comparable to only 1.6 °C of cooling (from 3.6 °C to 2.0 °C, $\sim 11$ ppm), and the solubility effect is underestimated by $\sim 4$ ppm. As we do not change salinity, we are simultaneously likely to overestimate the increase in solubility between $Ctrl$ and a glacial–like state, by $\sim 6$ ppm (Kohfeld and Ridgwell, 2009). This effect is consistent for any choice of C/P parametrisation, and is therefore not explored further.

### 3.2.2 Reduced wind forcing

When the peak of Southern Ocean (henceforth SO) winds is reduced, the strength of the overturning circulation of AABW decreases (see difference in Southern Hemisphere overturning streamfunction between Fig. 2 a and b). Thus, given that the volume of AABW does not change, its residence time increases. This also means that the upwelling nutrient–rich water in the SO stays a longer time near the surface and loses more nutrients before being subducted. This decreases the SO concentration of preformed phosphate in $WNS \times 0.5_{RED}$ compared to $Ctrl$, as seen in Fig. 7 b, and increases the nutrient utilisation efficiency $\overline{P^*}$ (Table 2, Fig. 9). This leads to a drawdown of $pCO_2^{atm}$ of 12.9 ppm compared to $Ctrl_{RED}$ (Fig. 6). As the nutrient concentration in the SO decreases (Fig. 7 b), the flexible C/P ratio (Fig. 8 b) leads to an increased carbon capture efficiency in $GAM$ compared to $RED$ (see $GLcomb - Ctrl$ of biologically sourced carbon ($AC_{rem}$) in Fig. 9), which is partly compensated by a reverse effect in the Pacific equatorial region. Consequently, in $WSN \times 0.5_{GAM}$, we get a reduction of $pCO_2^{atm}$ of 16.3 ppm compared to $Ctrl_{GAM}$ (Fig. 6). Hence, for halved peak wind stress at $\pm50°N$, the flexible stoichiometry increases the drawdown by $\sim 26$ %.

### 3.2.3 Remineralisation length scale

When the remineralisation length scale (RLS) increases, the biological material reaches deeper before it is remineralised, and it takes longer for it to be returned to the surface. Therefore, more of the biologically sourced carbon ($AC_{rem}$) and nutrients are present in the deep ocean at any given time, leading to an increase in $\overline{P^*}$ (Table 2, Fig. 9) and a decrease in $pCO_2^{atm}$ (Fig. 6). The deeper we make the RLS, the bigger the drawdown of $pCO_2^{atm}$ – in $RLS \times 1.25_{RED}$ and $RLS \times 1.75_{RED}$ $pCO_2^{atm}$ decreases by 14 and 33 ppm, respectively, compared to $Ctrl_{RED}$. In $GAM$, the drawdown in each experiment is increased by an additional $\sim 30$ %, thus $RLS \times 1.25_{GAM}$ and $RLS \times 1.75_{GAM}$ see a reduction of $pCO_2^{atm}$ of 18 and 44 ppm, respectively, compared to $Ctrl_{GAM}$ (Table S.2). Our changes in RLS cause very small, but global, changes in $PO_4$ concentrations (global average anomaly = -0.016 $\mu M$, $RLS \times 1.25$ in Fig. 7 c), which, through the small resulting changes in C/P (Fig. 8 c), still contribute to the additional drawdown of $pCO_2^{atm}$ in $GAM$.

In a sensitivity test where we make the RLS 25 % shallower (which would be representative of a warmer climate c.f. $Ctrl$), the $pCO_2^{atm}$ increases by 18 and 23 ppm in $RED$ and $GAM$ respectively compared to their control states (see $RLS \times 0.75_{RED}$ and $RLS \times 0.75_{GAM}$ in Table S.2). Interestingly, the response in $pCO_2^{atm}$ is again $\sim 30$ % larger in $GAM$. The variable stoichiometry thus amplifies the effect on $pCO_2^{atm}$ by any change in RLS. The potential implications of this result for warm climate scenarios is further discussed in Section 4.1.

### 3.2.4 Dust forcing

The simulations with LGM dust forcing ($LGMdust$, Table 1) show the largest difference in $pCO_2^{atm}$ between the $RED$ and the $GAM$. In $LGMdust_{RED}$, $pCO_2^{atm}$ decreases by 16 ppm compared to $Ctrl_{RED}$, whereas $LGMdust_{GAM}$ sees a reduction of 21 ppm compared to $Ctrl_{GAM}$ (Fig. 6). The drawdown is thus $\sim 30$ % larger with variable stoichiometry. About a third ($\sim 10$ % of $\sim 30$ %) of the increase in drawdown can be explained by a change in average C/P composition of the organic material that is exported out of the surface ocean (see Section 4.4).

As anticipated, the iron added by the dust forcing allows more efficient usage of P in the HNLC-regions, which increases the ocean storage of biologically sourced carbon $P_{rem}$ (thus, $P^*$ increases, Table 2, Fig. 9). This reduces the surface nutrient concentrations in these areas (Fig. 7 d). In the $GAM$ model version, this is followed by increased C/P ratios in these areas (Fig. 8 d), resulting in a lower $pCO_2^{atm}$ in $LGMdust_{GAM}$ than in $LGMdust_{RED}$. The largest anomalies in $PO_4$ concentrations, and consequently in C/P, are observed in subantarctic zone of the Southern Ocean, particularly in the Atlantic and Indian sectors. This subantarctic increase in biological efficiency is consistent with radionuclide proxy data ([10]Be, [230]Th, [231]Pa) from the LGM (e.g., Kumar et al., 1995; Kohfeld et al., 2005).

### 3.3 Combined experiments

We show the results of two different combined simulations; $Acomb$ and $GLcomb$. $GLcomb$ is the "glacial–like" simulation, which combines all the sensitivity experiments (Table 1). $Acomb$ omits the reduction in wind stress.

### 3.3.1 Ocean temperature and circulation

In the glacial–like simulations, $GLcomb_{RED}$ and $GLcomb_{GAM}$, the global average ocean temperature ($\overline{T_{oce}}$) is 1.7 °C lower than in the respective control states (Table 2). Headly and Severinghaus (2007) estimate LGM $\overline{T_{oce}}$ to have been $2.7 \pm 0.6$ °C colder than the modern ocean, while a more recent estimate by Bereiter et al. (2018) constrains $\overline{T_{oce}}$ to $2.57 \pm 0.24$. $GLcomb$ is thus just outside the one standard deviation limit of the warm end of the Headly and Severinghaus (2007) estimate for the LGM. In $Acomb$, $\overline{T_{oce}}$ is 2.1°C cooler than $Ctrl$, thus this simulation falls within the uncertainty of the Headly and Severinghaus (2007) estimate. Compared to the Bereiter et al. (2018) estimate, our combined experiments $GLcomb$ and $Acomb$ achieve 64 and 82 % of the glacial–interglacial difference in $\overline{T_{oce}}$, respectively. As anticipated, our combined forcings do not induce a full glacial maximum state, but a state with glacial–like climate conditions.

Ocean overturning circulation weakens in $GLcomb$ compared to $Ctrl$ (Fig. 2, Table 2), mainly as a result of the wind stress reduction. For example, the AMOC (here measured as the maximum of the Atlantic overturning streamfunction) reduces in strength by $\sim 15$ %, from 14 Sv to 12 Sv (Table 2). The global meridional overturning stream function reveals that the SO overturning cell sees a reduction in transport (Fig. 2 d), which is associated with weaker upwelling and thus longer residence time for AABW, as hypothesised for the glacial ocean (e.g., Menviel et al., 2017; Skinner et al., 2017). In $Acomb$, where the wind stress is kept at modern values, the ocean overturning circulation remains similar to the control state (Table 2).

### 3.3.2 Surface nutrient distribution and C/P ratios

In the surface nutrient anomalies ($GLcomb - Ctrl$, shown for $GAM$ in Fig. 7 f), we see the strongest response in the Southern Ocean, with different effects south and north of the so-called biogeochemical divide described by Marinov et al. (2006). Marinov et al. (2006) show that the air–sea balance of $CO_2$ is dominated by processes in the waters close to Antarctica, whereas global export production is instead controlled by the biological pump and circulation in the Subantarctic region. The border between these two regimes is referred to as the biogeochemical divide. South of the biogeochemical divide, close to the Antarctic continent, we see an increase in $GLcomb$ nutrient concentrations compared to $Ctrl$ (Fig. 7 f), which coincides with an increase in sea-ice in this area (not shown). Colder conditions due to changed albedo and radiative forcing, with more sea ice than in the control state, cause a reduction in biological production, leaving more unused P in the surface layer (Fig. 7 a). North of the biogeochemical divide, increased aeolian dust flux increases the productivity of the biology, which reduces P in the surface compared to the control (Fig. 7 d). In combination with circulation changes, resulting from the reduced SO wind stress (Fig. 7 b), and deeper remineralisation (Fig. 7 c), P concentrations in the Subantarctic region are strongly reduced (Fig. 7 f). In the North and Equatorial Pacific, there is also a reduction of P (Fig. 7 f), mainly due to to the increased dust flux (Fig. 7 d). There is an increase in P seen in the Arctic, again coincident with an increase in sea ice in the same area. As a result of these changes, we see strong positive anomalies in C/P in the HNLC-regions, and negative anomalies in the highest latitude bands (Fig. 8 f). The organic matter that is exported out of the upper layer (henceforth referred to as export production) in $GLcomb_{GAM}$ has a global average C/P ratio of 134/1.

### 3.3.3  Ocean dissolved $O_2$

Despite colder conditions, which allow for more dissolution of $O_2$, the reduction in $\overline{O_2}$ is evident in the $GLcomb$–simulations (Fig. 4). This mirrors the increase of $AC_{rem}$ (Fig. 9). The $\overline{O_2}$ reduction is about 50 % larger in $GAM$ compared to $RED$. As the initial state $Ctrl_{GAM}$ is already lower in oxygen than $Ctrl_{RED}$ ($144 \mu mol kg^{-1}$ compared to $166 \mu mol kg^{-1}$), and variable
stoichiometry allows for additional ocean storage of organic carbon, the end state $\overline{O_2}$ is drastically lower in $GLcomb_{GAM}$ ($74 \mu mol kg^{-1}$) compared to $GLcomb_{RED}$ ($122 \mu mol kg^{-1}$).

### 3.3.4  Ocean $\delta^{13}C$

In $GLcomb_{RED}$ (contours in Fig. 5 b,d), the Atlantic North–South gradient in $\delta^{13}C$ is stronger than in $Ctrl_{RED}$ (contours in Fig. 5 a,c). This strong gradient is not observed in the Holocene Atlantic $\delta^{13}C$-data (dots in Fig. 5 a,b), but is prominent
in the LGM time slice (dots in Fig. 5 c,d). The LGM observations are well reproduced in $GLcomb_{red}$ (corr. 0.62, Table S.3), especially in the East Atlantic (corr. 0.77). When we correct $GLcomb_{RED}$ for the absence of injected terrestrial carbon, we see clear similarities with LGM observations (Fig. 5 d), though the southernmost cores still indicate more $^{13}C$-depleted conditions than the model. In the Indo–Pacific, $GLcomb_{RED}$ is too $^{13}C$-depleted compared to LGM observations, particularly with the correction for the absence of a terrestrial signal (Fig. 5 h), and the model–data correlation is poor (0.05). For the Indo–Pacific,
the model–data correlation for Holocene data is similar between $GLcomb_{RED}$ and $Ctrl_{RED}$ (0.24 and 0.39 respectively, Table S.3). This suggests that the poor correlation with LGM data is simply due to our changes in forcings being insufficient to achieve the required rearrangements in Indo–Pacific circulation patterns. In $GLcomb$, similarly as in $Ctrl$, there is very little difference between the $RED$ and $GAM$ model versions in terms of $\delta^{13}C$ patterns (compare Fig. 5 d,h to Fig. S.3 d,h) and model–data correlations (Table S.3). However, there are overall lower values of $\delta^{13}C$ in $GAM$, reflecting the larger storage of
biologically sourced carbon in the deep ocean in this model version.

### 3.3.5  Atmospheric $CO_2$

In the combined experiments $Acomb$ and $GLcomb$, $pCO_2^{atm}$ decreases strongly compared to the control state (reduction of between $-56$ to $-80$ ppm, Fig. 6). This is partly a result of colder conditions (Table 2), which lead to increased solubility for $CO_2$ in sea water and an expanded sea–ice cover (Figure S.4) which restricts air–sea gas exchange. The latter slows down
the equilibration of the $CO_2$–rich upwelling water with the atmosphere before it is subducted into the deep ocean, and thus causes increased $C_{dis}$. In $Acomb$ and $GLcomb$, changes in biological production (see Section 3.3.2) and storage of biologically sourced carbon (Fig. 9) also contribute strongly to the reduced $pCO_2^{atm}$.

    The combined experiments show a striking difference in $pCO_2^{atm}$ between the model versions $RED$ and $GAM$; in $GLcomb_{RED}$ and $GLcomb_{GAM}$, we achieve drawdown of $pCO_2^{atm}$ of $-64$ and $-80$ ppm, respectively, from the 278 ppm of the control
states (Fig. 6). This corresponds to an increase in ocean carbon storage of 139 PgC and 173 PgC, respectively (Table S.1). The drawdown is thus 25% larger in $GAM$ than in $RED$. In $Acomb$, where the perturbation in wind stress is omitted, drawdown

of $pCO_2^{atm}$ is smaller than in $GLcomb$ and the difference between model versions is less pronounced, but there is still a 14 % difference between $Acomb_{RED}$ and $Acomb_{GAM}$ ($-56$ and $-63ppm$, respectively) (Fig. 6).

## 4 Discussion

### 4.1 Accounting for variable C/P in ocean carbon cycle models

The representation of ocean biology in General Circulation Models (GCMs) tends to be over–simplified (Le Quéré et al., 2005a) and the development of the models is often held back by constraints imposed by maintaining the computational efficiency of the model. The Galbraith and Martiny (2015) empirical model is simple and based on nutrient variables that are already present in biogeochemical models (nitrate and phosphate). By implementing the $GAM$ parametrisation, or possibly a power law as that described by (Tanioka and Matsumoto, 2017), an additional facet of the complexity of ocean biology can be implicitly

accounted for at a relatively small computational cost. Previous model ensemble studies have shown that this type of dynamical response of the biology to changes in the modelled ocean state can improve the model's ability to realistically simulate ocean biogeochemistry (Buchanan et al., 2018). In pre–industrial and future simulations, respectively, Buchanan et al. (2018) and Tanioka and Matsumoto (2017) find that the flexible stoichiometry acts to stabilise the response of ocean DIC to changes in the physical (circulation) state. In our glacial–like simulations, we find that the response of ocean DIC, and thus $pCO_2^{atm}$, to the

combined perturbations is greater in the simulations with flexible stoichiometry. Nonetheless, our study confirms the potential importance of dynamical biological response for the outcome of model studies.

The approach taken by both Galbraith and Martiny (2015) and Tanioka and Matsumoto (2017) is to adapt a function to all the available species–independent observations. This means that they account for the adaptation of plankton to the surrounding conditions, both in terms of species composition and individuals being more frugal in low nutrient conditions. This is one of

the main advantages of such an approach, as it can be applied in a model without different plankton functional types, which is what we use here. Ganopolski and Brovkin (2017) appear to succeed with full glacial–interglacial CO$_2$ cyles in CLIMBER-2, which does not have flexible stoichiometry for primary producers, but which has a temperature limitation on growth and explicit phyto- and zooplankton with different C uptake rates. This combination may perhaps achieve a similar response in carbon export in their simulations, when moving into a colder climate, as the flexible stoichiometry does in our simulations.

The next step to approach a more realistic modelling of the biological pump would be to include a representation of preferential remineralisation of nutrients (e.g., Kolowith et al., 2001; Letscher and Moore, 2015), but this goes beyond the scope of the present study.

One drawback of the Galbraith and Martiny (2015) approach is that it assumes that the C/P ratio continues to increase continuously with increasing $[PO_4]$. As such, in a high surface $[PO_4]$ region like the Southern Ocean, $GAM$ does not represent

the effects on C/P of temperature and light, and the associated non–linear effects, that could be of importance here (e.g., Yvon-Durocher et al., 2015; Tanioka and Matsumoto, 2017; Moreno et al., 2018). Up to $[PO_4] = 1.7\mu M$, the $GAM$ parametrisation fits the binned observational data well (see Fig. 1 and S2 Galbraith and Martiny, 2015). To account for the lack of observational data at higher $[PO_4]$, we have tested the effect of saturation of the C/P ratio at higher $[PO_4]$ in $Ctrl$ and $GLcomb$ simulations

where we kept C/P = 55/1 at $[PO_4] > 1.7\mu M$. The increase in ocean carbon storage and decrease in $pCO_2^{atm}$ between the $Ctrl$ and $GLcomb$ are nearly identical with $GAM$, thus saturation of the C/P ratio at very high $[PO_4]$ causes no noticable impact on our results.

Our sensitivity experiments $RLS \times 0.75$ and $RLS \times 1.25$, reveal that the response in $pCO_2^{atm}$ to the perturbation is enhanced in $GAM$ compared to $RED$ for both increased and decreased RLS. While increased $RLS$ would be an effect of ocean cooling, and thus of interest for glacial studies, reduced $RLS$ would be a consequence of ocean warming (Matsumoto, 2007). Matsumoto (2007) describes how decreased RLS would have a positive feedback on $pCO_2^{atm}$ in future warming climate. Our results imply that flexible C/P could have a further re–inforcing effect on this feedback. It would therefore be of interest to apply a parametrisation of flexible C/P in models used for simulations of future climate feedbacks.

## 4.2 De–coupling of biologically sourced carbon and nutrient utilisation efficiency

Many studies have suggested increased ocean storage of organic carbon as a potentially important contributer to the low glacial $pCO_2^{atm}$ (e.g., Sarmiento and Toggweiler, 1984; Martin, 1990; Archer et al., 2000; Sigman and Boyle, 2000). However, biological production depends on water temperature (Eppley, 1972) and decreases in cold conditions. This temperature effect is parametrised in cGENIE as a local temperature–dependent uptake rate modifier proportional to $e^{(T/15.9^\circ C)}$, and we see overall reduced productivity in our cold climate simulations (see e.g. $LGM_{phy}$, Table S.1). As the climate cools, the temperature effect leads to a decrease in biological productivity and a subsequent decrease in $\overline{P^*}$ (see $LGM_{phy}$ in Fig. 9). If productivity decreases, other mechanisms are needed to offset this decrease, if the total ocean storage of organic carbon is to increase. Mechanisms that can contribute to increased deep ocean carbon retention are for example reduced SO overturning circulation (increased residence time of the AABW, Menviel et al., 2017) and deeper remineralisation (Matsumoto, 2007; Chikamoto et al., 2012), which we apply through our perturbations in winds (Section 3.2.2) and RLS (Section 3.2.3). From our results, it is evident that variable stoichiometry can be another contributing factor. In our simulations, global average export production decreases, both in terms of $POP$ (particulate organic phosporus) and $POC$ (particulate organic carbon), by 15 % between $Ctrl_{RED}$ and $GLcomb_{RED}$ because of the colder climate. However, the variable stoichiometry in $GAM$ partly offsets this decrease in biological carbon capture - while the export production in terms of $POP$ decreases by 17 %, the corresponding decrease in $POC$ is only 10 %. Thus, in $GLcomb_{GAM}$ we achieve an increase in ocean inventory of remineralised carbon, which exceeds the one of $GLcomb_{RED}$ (Fig. 9).

In summary, we suggest this is a result of changes in surface P fields (see Fig. 7)

Due to the competing effects between decreased export production and increased retention that are introduced by the different forcing components, the net change in $P_{rem}$, and thus in $\overline{P^*}$, is very small when moving from the control state ($Ctrl$) to the glacial–like state ($GLcomb$) (Fig. 9). In the fixed stoichiometry case ($RED$), there is a small net increase in $\overline{P^*}$ of 0.020 in $GLcomb$ compared to $Ctrl$, which is linearly related to a small increase in storage of $AC_{rem}$ in the ocean. In the case with variable stoichiometry ($GAM$), there is instead a very small decrease in $\overline{P^*}$ of 0.003. With a linear response, we would thus expect a decrease in storage of $AC_{rem}$ as well. Instead, we see a reasonably large increase in global average $AC_{rem}$.

The reduced $pCO_2^{atm}$ in $GLcomb_{GAM}$, compared to $GLcomb_{RED}$, can be explained by the non–linearity introduced by the local variability in C/P. When changes in ocean circulation, remineralisation depth and dust deposition cause the local nutrient availability in the surface waters to change, this affects the elemental composition of the exported organic material. In $GLcomb$, there is a reduction of the surface layer P concentration compared to $Ctrl$ in some key (HNLC) regions. In $GAM$, this decrease in surface P results in an increase in C/P. This further strengthens the biological pump in these key regions, resulting in a non–linear relationship between storage of remineralised phosphate and biologically sourced carbon (Fig. 9). In $Ctrl_{GAM}$, the average elemental C/P composition is 121/1. In $GLcomb_{GAM}$, this average is 134/1. This means that even though the same amount of P is exported to the deep ocean, the organic molecules carry more carbon, which is released in the deep ocean during remineralisation. In $Ctrl_{GAM}$, the global average concentration of $P_{rem}$ is 1.16 $\mu mol kg^{-1}$ (c.f. 1.17 $\mu mol kg^{-1}$ in $GLcomb_{GAM}$). By increasing the average C/P composition of 1.16 $\mu mol kg^{-1}$ organic molecules from 121 to 134 (i.e. by 13 units), this causes an increase in $C_{soft}$ by $\sim 15$ $\mu mol kg^{-1}$, which corresponds to the observed increase in $AC_{rem}$. From this, we conclude that, in a system where stoichiometry is variable on a local scale, ocean storage of biologically sourced $CO_2$ can change while the amount of remineralised nutrients remains constant.

Note also that $Ctrl_{GAM}$ has a larger inventory of $DIC$ as well as $C_{soft}$ compared to $Ctrl_{RED}$. Ödalen et al. (2018) found that drawdown of $pCO_2^{atm}$ in response to a perturbation is larger when the control state has a smaller inventory of DIC and $C_{soft}$. Yet, the effect of applying the same perturbation results in a larger drawdown of $pCO_2^{atm}$ in $GAM$ than in $RED$. This is opposite of the conclusions of Ödalen et al. (2018). The reason is that the flexible stoichiometry in effect increases the drawdown potential, which more than compensates for the increased carbon inventory in the control state. In $GLcomb_{GAM}$, the C/P in global average export production increases from the control 121/1 to 134/1, which reflects the increased storage of organic carbon allowed by the flexible stoichiometry.

### 4.3   Implications of flexible C/P for deep ocean oxygen

Studies have shown that deep ocean $O_2$ concentrations were lower during the LGM than in the Holocene (e.g., Bradtmiller et al., 2010; Jaccard and Galbraith, 2012; Galbraith and Jaccard, 2015). Here, we discuss implications of using flexible C/P for the model's ability to reproduce ocean oxygen patterns and concentrations in the different time periods.

In the Atlantic, $Ctrl_{GAM}$ (Fig. 4 e) reproduces the observed extent of the Oxygen Minimum Zone (OMZ) (Fig. 4 a) better than $Ctrl_{RED}$ (Fig. 4 c) does. In the Pacific Ocean, the $O_2$ gradient in the observations (Fig. 4 b) is more gradual compared to that of the control states (Fig. d, f), but the core of the OMZ is well reproduced by the model. The forcings applied to $GLcomb$ are not sufficient to reproduce a full glacial state. Still, we do get a vertical expansion of the OMZ in $GLcomb_{RED}$ (Fig. 4 h) compared to $Ctrl_{RED}$ (Fig. 4 c), in agreement with the findings of Hoogakker et al (2018). In $GLcomb_{GAM}$, oxygen depletion is too extensive (see below), but the tendency of vertical expansion compared to the control state is present here as well.

In our glacial–like states, we see a significant reduction of $O_2$ compared to the control state in both $RED$ and $GAM$ (Fig. 4 d and e). This response is expected when we apply increased dust deposition and deeper remineralisation (Galbraith and Jaccard, 2015), and the direction of the overall response of deep ocean $O_2$ to the glacial–like forcings is in line with observations. The reduction in $O_2$ is stronger in $GAM$, due to the larger storage of respired carbon in this model version.

Finally, it should be noted that $Ctrl_{GAM}$ (Fig. 4 c) displays deeper oxygen minima in the oxygen minimum zones of both the Atlantic equatorial region and the North Pacific compared to what is seen in $Ctrl_{RED}$ (Fig. 4 b), and in observations. In $Ctrl_{GAM}$, a large part of the interior North Pacific is anoxic (Fig. 4 c), while observations (Fig. 4 a) indicate very low oxygen levels, but not anoxia. As illustrated by simulations in Galbraith and de Lavergne (2018), variable stoichiometry in

itself is not always sufficient to achieve widespread deep ocean de–oxygenation in a model under glacial–like climate change. Among other factors, model ocean oxygen conditions are also dependent on deep water formation characteristics of the model (Galbraith and de Lavergne, 2018). The deep water formation characteristics of a model affects the amount of time available for remineralisation and, consequently, the oxygen consumption. In addition, due to a lack of resolution deep water formation in climate models generally happens as open water convection, rather than as dense plumes along slopes (Heuzé et al., 2013).

This may cause too much oxygen to be entrained into the deep ocean Galbraith and de Lavergne (2018). In cGENIE, this effect is small enough not to cancel the increased $O_2$ consumption caused by the higher average C/P in $Ctrl_{GAM}$ compared to $Ctrl_{RED}$.

In summary, the $GAM$ version of the model reproduces quantitatively the observed $O_2$ patterns of both the LGM and the Holocene, but with too low concentrations. If this parametrisation of variable stoichiometry is to be used in cGENIE in future

studies, we suggest some re–tuning, for example by reducing the global average concentration of nutrients, to improve the representation of observed modern–day ocean $O_2$ concentrations in the $GAM$ control state.

## 4.4    Effect of modified but fixed C/P

Part of the observed difference in $pCO_2^{atm}$ between $GLcomb_{RED}$ and $GLcomb_{GAM}$ results from a difference in global average C/P in the control states ($Ctrl_{RED}$ and $Ctrl_{GAM}$). In $Ctrl_{GAM}$, the average C/P in the export production is close to

121/1, instead of 106/1 as in $Ctrl_{RED}$. We illustrate the consequences of this difference by running parallell simulations with fixed C/P of 121/1 (model version 121).

In a simulation with reduced wind stress ($WNS \times 0.5_{121}$), $pCO_2^{atm}$ is reduced by $-14.5$ ppm compared to $Ctrl_{RED}$ (Table S.2). The corresponding $pCO_2^{atm}$ drawdown in $WNS \times 0.5_{RED}$, and $WNS \times 0.5_{GAM}$, is $-12.9$, and $-16.3$ ppm, respectively. This indicates that, for the reduced wind stress case, about half of the enhanced drawdown achieved by the flexible

stoichiometry can be attributed to a to a difference in the mean C/P composition of the organic material between the control states $Ctrl_{RED}$ and $Ctrl_{GAM}$. Similarly, for the enhanced dust deposition experiments, $LGMdust_{121}$ explains about 1/3 of the difference in $pCO_2^{atm}$ between $LGMdust_{RED}$ and $LGMdust_{GAM}$ (Table S.2). In the combined experiment $GLcomb$, we can attribute about half ($-8$ ppm of $-17$ ppm) of the observed difference in drawdown between $RED$ and $GAM$ to the difference in control state average C/P. In $Acomb$, the control state difference in C/P accounts for about 2/3 of the difference

between model versions ($-5$ ppm of $-8$ ppm). As shown above, the individual perturbation simulations and $GLcomb$ have smaller fractions of the change in $pCO_2^{atm}$ that are due to changed average C/P. As shown above, the simulations with 121 indicate that, depending on the change in forcing, between 1/3 and 2/3 of the difference in drawdown between $RED$ and $GAM$ is due to the difference in average C/P between the control states (Fig. 6, Table S.2). From this, we conclude that the effects of the perturbations do not add linearly.

As outlined in Section 4.3, an increase in C/P reduces deep ocean $O_2$ concentrations, through an increase in regenerated carbon. In this respect, the 121 experiments fall between the corresponding $RED$ and $GAM$ experiments. For example, the global ocean average $O_2$ concentration, $\overline{O_2}$, in $Ctrl_{121}$ is 152 $\mu mol kg^{-1}$, which is lower than $Ctrl_{RED}$ (166 $\mu mol kg^{-1}$), but higher than $Ctrl_{GAM}$ (144 $\mu mol kg^{-1}$). Similarly, $GLcomb_{121}$ has a lower $\overline{O_2}$ than $GLcomb_{RED}$ (96 compared to 122 $\mu mol kg^{-1}$), but higher than $GLcomb_{GAM}$ (74 $\mu mol kg^{-1}$).

The observed effects of modified average C/P could have implications for model intercomparison projects, if they compare results from models that use different versions of fixed stoichiometry (for example, Anderson and Sarmiento (1994) or Takahashi et al. (1985) stoichiometries, compared to Redfield (1963)). The problem with different stoichiometry assumptions in models is extensively discussed by Paulmier et al. (2009). Our study shows a direct consequence of such differences, with different model response to the same perturbation.

## 4.5 What can we learn from the model–data comparison of $\delta^{13}C$ ?

Proxy records of benthic $\delta^{13}C$ indicate a change in ocean $\delta^{13}C$ across the deglaciation. The whole ocean deglacial change has been estimated to $\sim 0.35$ ‰(Peterson et al., 2014; Peterson and Lisiecki, 2018), and the surface–to–deep gradient weakened (shown in numerous studies, see e.g. Curry et al., 1988; Duplessy et al., 1988; Curry and Oppo, 2005; Herguera et al., 2010; Peterson and Lisiecki, 2018, , and references therein). Here we compare model ocean $\delta^{13}C$ of our simulations to the benthic $\delta^{13}C$ records, to see how well the simulations capture the observed patterns, and if there is a difference in model–data correlation between $RED$ and $GAM$.

The model–data comparison in $\delta^{13}C$ (Section 3.1.4) suggests that the $Ctrl$ simulations are overall well correlated with Holocene benthic $\delta^{13}C$ data (HOL in Table S.3). For the Atlantic, the correlation of the $Ctrl$ simulations is higher with the LGM benthic $\delta^{13}C$ (LGM in Table S.3) than with HOL. However, the standard deviations (STDs) suggest that the Atlantic North–South gradient is not as strong as in an LGM ocean state and thus more similar to the modern ocean.

When we apply our combined forcings in the $GLcomb$ simulations, we achieve a stronger $\delta^{13}C$ gradient in the Atlantic, allowing for a closer match with LGM data in terms of STDs. A stronger gradient in $\delta^{13}C$ and more depleted suggest weaker ventilation of the deep ocean. The poor correlation in the Indo–Pacific, which may be partly due to sparse mid–ocean observations for almost 70% of the ocean volume, makes the global statistics for LGM observations difficult to interpret. The forcings applied to $GLcomb$ are factors that are likely to be important for the glacial ocean circulation and biogeochemistry. However, these forcings are not sufficient to reproduce a full glacial state (i.e. the use of the term *glacial–like simulations*, rather than *LGM simulation*). Other forcings that have shown to be important for modelling of glacial $\delta^{13}C$ are, for example, brine rejection (Bouttes et al., 2010, 2011), and freshwater forcing (Schmittner et al., 2002; Hewitt et al., 2006; Bouttes et al., 2012). The fact that some important forcings are missing is likely the main cause for the model–data discrepancy, and the reason for why we do not achieve a glacial Pacific Ocean circulation consistent with observed $\delta^{13}C$ patterns.

Each of the two observational datasets (HOL and LGM) display similar correlations across the two model simulations. The correlation of the HOL $\delta^{13}C$ records with $Ctrl_{RED}$, $Ctrl_{GAM}$, $GLcomb_{RED}$, and $GLcomb_{GAM}$, is in all cases between 0.76–0.78. Meanwhile, the correlation of LGM $\delta^{13}C$ records with the same four simulations is in all cases between 0.55–0.58.

As our $GLcomb$–simulations still correlate so well with the HOL dataset, this suggests the applied forcings have not caused these simulations to be clearly different from $Ctrl$ in terms of water mass distribution. For the same reason, the correlation with LGM $\delta^{13}C$ records does not significantly improve from $Ctrl$ to $GLcomb$. The water mass distribution in cGENIE is strongly constrained by the resolution of the model, especially in the vertical. Changes in temperature and salinity that should cause changes in water mass volume may not be sufficient to allow a water mass to extend to the next vertical level of the model. As a consequence, while the gradient between water masses may become more or less pronounced, the interface of water masses may still remain at the same depth. The applied changes affect the chemical and biological conditions for ocean carbon storage, such as $CO_2$ solubility (temperature dependent) and nutrient availability, more than the physical conditions, such as water mass volume and turnover time. To achieve a full glacial state with cGENIE, with a more glacial–like water mass distribution, additional physical forcings (see above) are likely to be required.

How $\delta^{13}C$ is represented in cGENIE is detailed in Appendix B. The very small differences between $RED$ and $GAM$ suggests that using variable stoichiometry does not impact our ability to represent the $\delta^{13}C$ patterns seen in observational data. However, we note that some retuning of the modern control state is recommended before cGENIE with variable stoichiometry is used in other studies.

## 5   Conclusions

In this paper, we examine the potential role of variable stoichiometry in biological production for glacial ocean $CO_2$ storage. We show that flexible C/P composition of organic matter allows a stronger response of $pCO_2^{atm}$ to glacial–like changes in climate, remineralisation length scale and aeolian dust flux. We conclude that variable stoichiometry may be important for glacial ocean $CO_2$ storage and for achieving the full extent of drawdown of atmospheric $CO_2$ in model simulations. In the experiment $GLcomb_{RED}$, with glacial–like climate and Redfield stoichiometry (Redfield, 1963), ocean carbon storage increases by 139 PgC and atmospheric $CO_2$ decreases by 64 ppm. In $GLcomb_{GAM}$, with glacial–like climate changes and variable stoichiometry, the corresponding numbers are 173 PgC and 80 ppm. Hence, the drawdown of atmospheric $CO_2$ increases by 25 % when C/P is variable.

About half of the increased drawdown of $CO_2$ results from different global average C/P in the export production. In addition, flexible stoichiometry allows increased carbon capture through the biological pump, while maintaining or even decreasing the fraction of remineralised to total nutrients in the deep ocean. With fixed stoichiometry, an increase in remineralised carbon is inevitably tied to a corresponding increase in remineralised nutrients.

We apply variable C/P parametrised as a simple function of the surface water concentration of $PO_4$, as suggested by Galbraith and Martiny (2015). Tanioka and Matsumoto (2017) suggest that it is unrealistic for C/P to continue to increase indefinitely with increased $[PO_4]$, and therefore suggest a more complex power law function, which takes into account saturation of the C/P ratio at high concentrations of $PO_4$. However, we found that saturation of the C/P ratio at concentrations higher than the observational upper bound of 1.7 $\mu M$ causes no noticable impact on our results.

The representation of flexible stoichiometry used in this study (Galbraith and Martiny, 2015) can be used without large increases in computational cost. It makes it possible to take into account, to first approximation, the complex biological changes that occur in the ocean during long–timescale climate change scenarios (see e.g. McInerney and Wing, 2011). We show here that, for glacial–interglacial cycles, this complexity contributes to changes in atmospheric $CO_2$ through flexible C/P ratios. Flexible C/P also has the potential to be an additional positive feedback of ocean warming on $pCO_2^{atm}$ in future climate.

*Code and data availability.* The code for the cGENIE.muffin model is hosted on GitHub. The specific version used in this paper, tagged as release v0.9.5, can be obtained at https://github.com/derpycode/cgenie.muffin/releases/tag/v0.9.5 and is assigned a DOI 10.5281/zenodo.3235761 (cGENIE release v0.9.5, 2019).

Configuration files for the specific experiments presented in the paper can be found in the directory cgenie.muffin/genie-userconfigs/MS/odalenetal.BG.2019. Details of the different experiments, plus the command line needed to run each one, are given in readme.txt.

A corresponding user manual detailing software installation and configuration, plus cGENIE.muffin model tutorials, is available from https://github.com/derpycode/muffindoc/releases/tag/1.9.1b and is assigned a DOI 10.5281/zenodo.1407658. (cGENIE release 1.9.1b, 2018)

Datasets are available upon request (e-mail to malin.odalen@misu.su.se).

## Appendix A: Regenerated acidic carbon

Carbon enters the ocean mainly in the form of $CO_2$ and dissolved carbonate. Despite this, the major fraction of carbon in the ocean resides as bicarbonate ions. The source related state variables acidic and basic carbon (AC and BC, respectively) allow us to separate the ocean DIC inventory into the sources of $CO_2$ (AC) and dissolved carbonate (BC). The concept of the sourced related state variables was first described by Walin et al. (2014).

AC and BC are defined from DIC and alkalinity (ALK) as

$$AC = DIC - \frac{1}{2}ALK \tag{A1}$$

$$BC = \frac{1}{2}ALK \tag{A2}$$

Total AC and BC include all ocean sources of carbon, including (but not limited to) river runoff, air–sea gas exchange, and the biological pump.

To isolate the $CO_2$ that was supplied to the ocean via biological soft tissue, we make use of the separation of DIC and ALK into their preformed and remineralised fractions (see Section 2.5). Thus, we then compute the remineralised acidic carbon ($AC_{rem}$) as

$$AC_{rem} = DIC_{rem} - \frac{1}{2}ALK_{rem} \tag{A3}$$

## Appendix B: $\delta^{13}C$ in cGENIE

cGENIE represents $^{13}C$ as an explicit tracer (separate from and in addition to bulk carbon) in the model, tracking its concentration in all the same gaseous, dissolved, and solid forms that carbon exists in, reporting $\delta^{13}C$ in ‰ relative to the standard VPDB. The current scheme is based on that described in Ridgwell (2001) and updated as described in Ridgwell et al. (2007)

and is evaluated for the modern ocean (alongside simulated $\Delta^{14}C$) in cGENIE, in Turner and Ridgwell (2016).

In the aqueous phase, the isotopic partitioning of carbon between $CO_2$(aq), $HCO_3^-$, and $CO_3^{2-}$ is resolved and follows Zeebe and Wolf-Gladrow (2001) (their Section 3.2). The empirical fractionation factors used are from Zhang et al. (1995). The air–sea fractionation scheme follows that of Marchal et al. (1998) with the individual fractionation factors again taken from Zhang et al. (1995).

For the isotopic composition of organic carbon ($\delta^{13}C_{POC}$), the model of Rau et al. (1996) is adapted, assuming that the isotopic signature of exported POC reflects that of phytoplankton biomass. Following Ridgwell (2001), the full equation of Rau et al. (1996) is simplified to:

$$\delta^{13}C^{POC} = \delta^{13}C^{CO_{2(aq)}} - \epsilon_f + (\epsilon_f - \epsilon_d) \cdot \frac{K_Q}{[CO_{2(aq)}]} \tag{B1}$$

where [$CO_2$(aq)] is the ambient concentration of aqueous $CO_2$ and $\delta^{13}C_{aq}$ is its isotopic composition. $K_Q$ is a temperature–

only dependent approximation of the full cell–dependent size and growth rate parameterization in the Rau et al. (1996) model (see Ridgwell, 2001). We take an intermediate value for the enzymatic isotope fractionation factor associated with intracellular C fixation ($\epsilon_f$) of -25‰ following Rau et al. (1996, 1997), and assume a temperature–invariant value for $\epsilon_d$ of 0.7 ‰.

The result of applying this scheme in cGENIE, is a zonal mean profile characterized by $\delta^{13}C_{POC}$ of -22 to -21 ‰ in the tropics, declining with increasing latitude to reach -28 to -30 ‰ in the Southern Ocean. This latitudinal pattern is comparable

to measurements made on suspended particulate organic matter as discussed in Ridgwell (2001).

For $^{13}C$ fractionation into biogenic carbonates at the ocean surface (e.g. foraminiferal tests, and coccolithophorid coccoliths), cGENIE follows Mook (1986) and employs a simple temperature–dependent fractionation between the $\delta^{13}C$ of aqueous $HCO_3^-$ and calcite.

*Author contributions.* M. Ödalen, J. Nycander, K. I. C. Oliver and A. Ridgwell designed the model experiments. A. Ridgwell developed the

original cGENIE model code. M. Ödalen and K. I. C. Oliver adapted the model code and forcings for the experimental design. M. Ödalen performed the model simulations and produced the tables and figures. C. D. Peterson provided expertise on ocean $\delta^{13}C$ analysis. M. Ödalen prepared the manuscript with contributions from all co–authors.

*Competing interests.* No competing interests are present.

*Acknowledgements.* The authors thank Pierce James Buchanan and one anonymous referee for their helpful comments which improved the paper. Model simulations were performed on resources provided by the Swedish National Infrastructure for Computing (SNIC) at the National Supercomputer Centre (NSC), Sweden. MÖ would like to acknowledge support from the Bolin Centre for Climate Research, Research Areas 1 and 6. AR was supported by a Heising–Simons Foundation award, and by EU grant ERC 2013-CoG-617313.

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

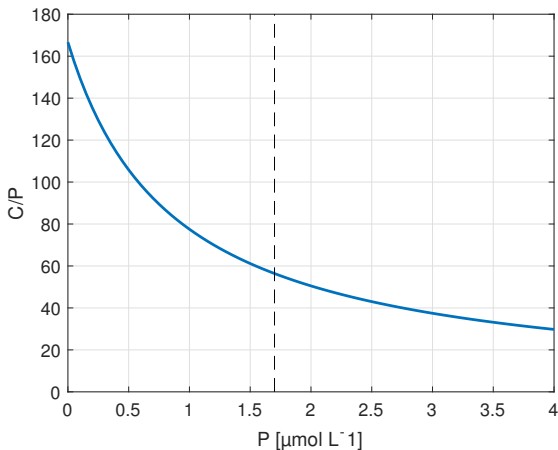

**Figure 1.** Flexible stoichiometry C/P (y-axis) dependent on the P-concentration $[\mu mol L^{-1}]$ (x-axis), as described by Eq. 1. Here, we extend the relationship beyond the observational interval 0–1.7 $\mu mol L^{-1}$ (bounded by dashed line) which form the basis of the relation derived by Galbraith and Martiny (2015).

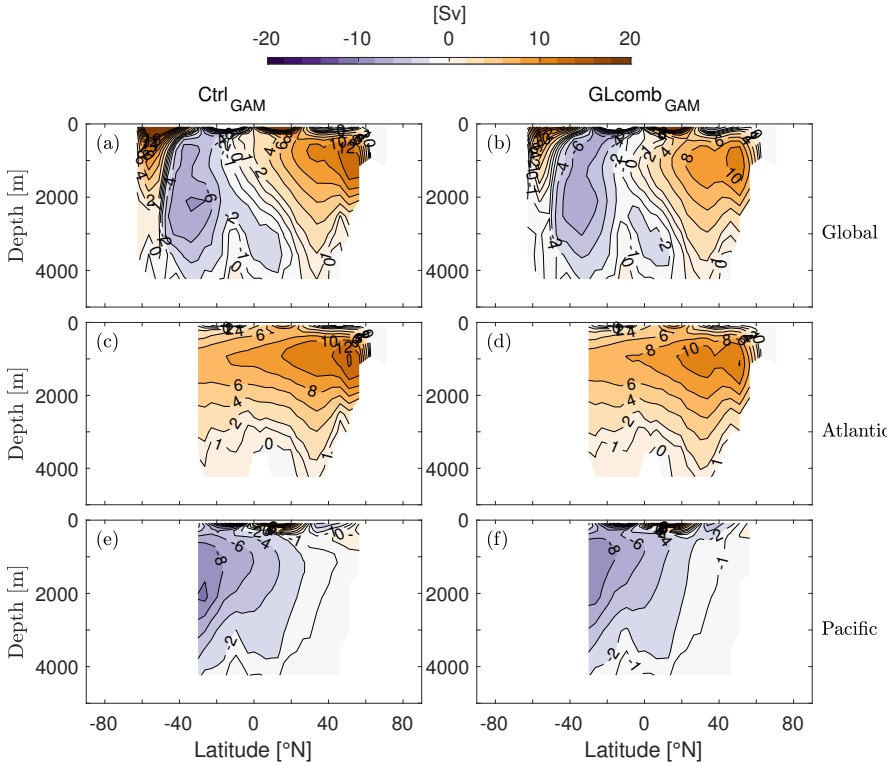

**Figure 2.** The eulerian component of the Global (panels a, b), Atlantic (panels c, d) , and Pacific (panels e, f) ocean meridional overturning streamfunction (1 Sv = $10^6 m^3 s^{-1}$) of $Ctrl_{GAM}$ (panels a, c, e) and $GLcomb_{GAM}$ (panels b, d, f). Note that the eddy–induced transport of tracers is taken into account through a skew–diffusive flux (Griffies, 1998) that is present in the velocity fields used to compute the eulerian stream function.

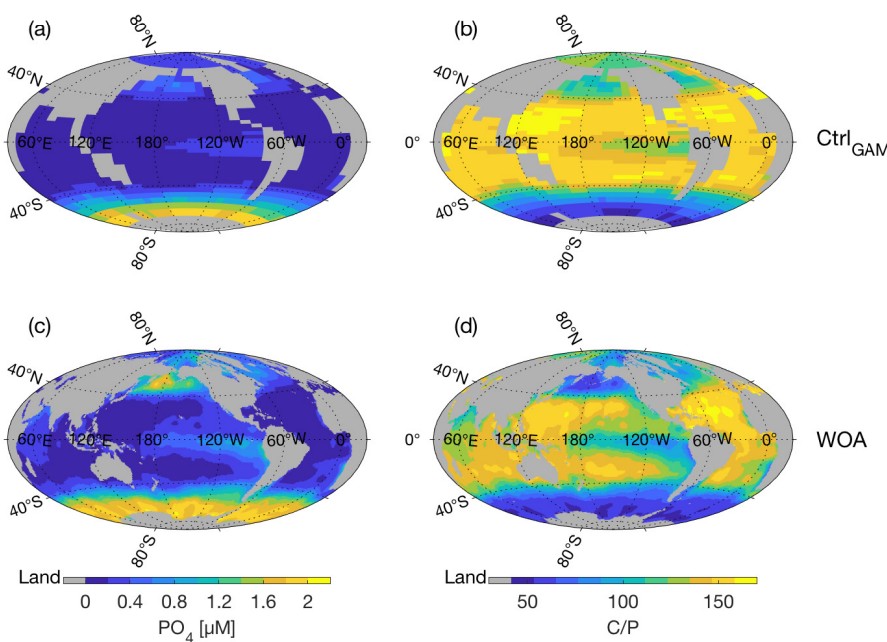

**Figure 3.** Surface $PO_4$ concentration ($\mu M$) (left hand column) and corresponding C/P as calculated using parametrisation of Galbraith and Martiny (2015) (right column). Panels show in $Ctrl_{GAM}$ (a, b) and observations (World Ocean Atlas 2018, Garcia et al., 2018b) (c, d).

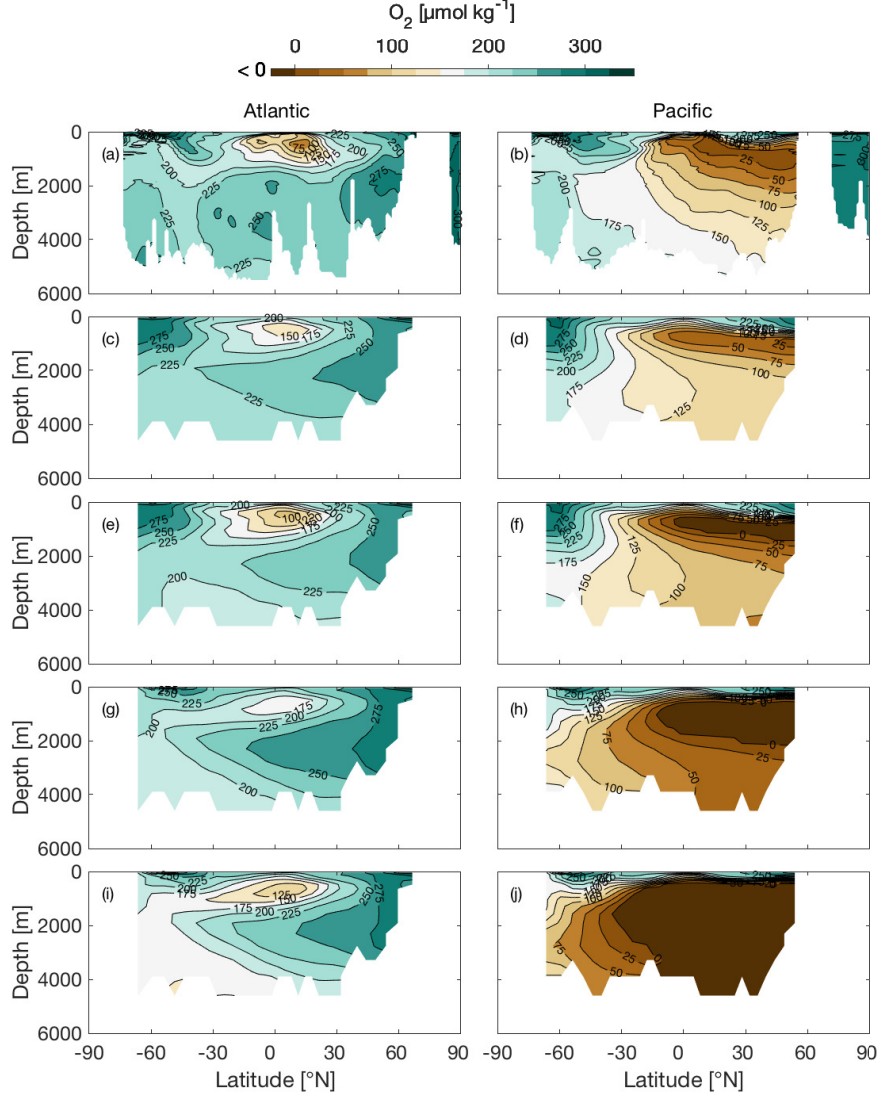

**Figure 4.** Sections of $O_2$ concentration ($\mu mol kg^{-1}$) along 25 °W in the Atlantic basin (left hand column) and along 135 °W in the Pacific basin (right hand column). Panels show observations (World Ocean Atlas 2018, Garcia et al., 2018c) (a, b), and model states $Ctrl_{RED}$ (c, d), $Ctrl_{GAM}$ (e, f), $GLcomb_{RED}$ (g, h) and $GLcomb_{GAM}$ (i, j).

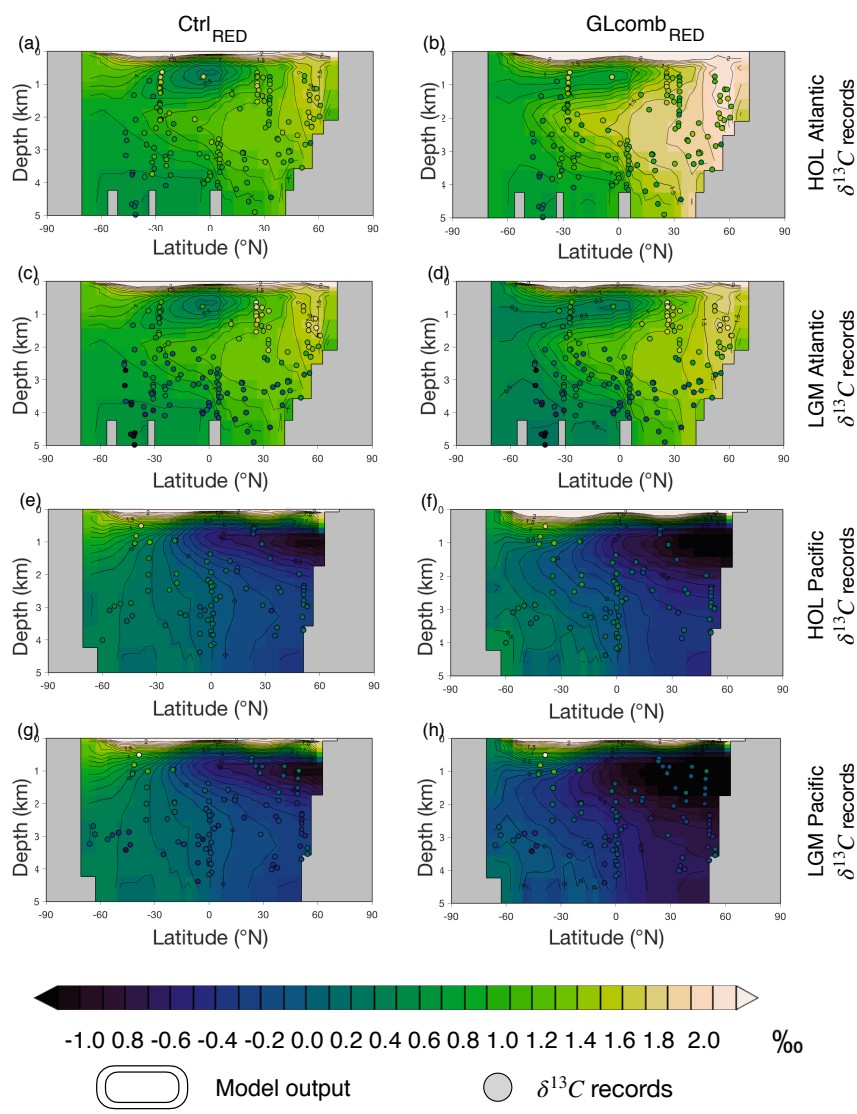

**Figure 5.** Model ocean $\delta^{13}C$ (contours) compared to the two proxy record time slices (HOL and LGM) of benthic $\delta^{13}C$ (circles) of Peterson et al. (2014). The upper half of the figure shows the Atlantic Ocean (panels a–d), while the lower half shows the Pacific Ocean (panels e–h). The columns represent the model simulations ($Ctrl_{RED}$ or $Ctrl_{GAM}$), while each row represents one of the proxy record time slices (HOL or LGM). The left hand column shows $Ctrl_{RED}$ (panels a, c, e, g) and the right hand column shows $GLcomb_{RED}$ (panels b, d, f, h). The rows show, from top to bottom, a–b) HOL Atlantic, c–d) LGM Atlantic, e–f) HOL Pacific, g–h) LGM Pacific. Note that, before we compare $GLcomb_{RED}$ to LGM observations (panels d and h), a constant of 0.32 ‰ is subtracted from the simulated $\delta^{13}C$, to account for terrestrial release of $\delta^{13}C$–depleted terrestrial carbon which is not modelled. The corresponding comparison for model version GAM is shown in Fig. S.3.

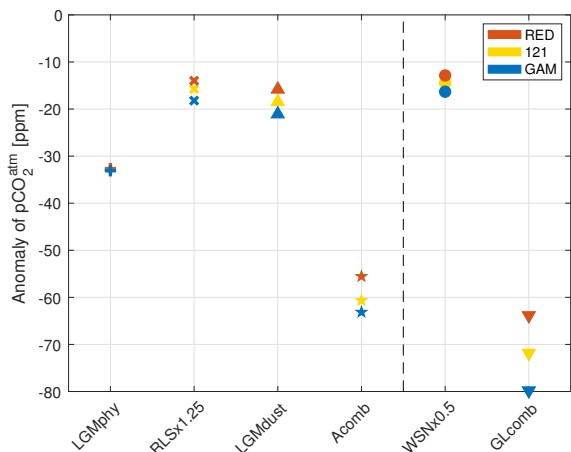

**Figure 6.** Resulting $CO_2$ anomaly, with respect to the control state 278 ppm, of the sensitivity experiments $LGMphy$ (plus–symbol), $RLS \times 1.25$ (×–symbol), $LGMdust$ (upward arrowheads), and $WSN \times 0.5$ (circles), and of the combined experiments $Acomb$ (stars) and $GLcomb$ (downward arrowheads). Results of the different model versions $RED$, 121 and $GAM$ are shown in red, yellow and blue, respectively. The vertical dashed line separates simulations without (left) and with (right) wind perturbation.

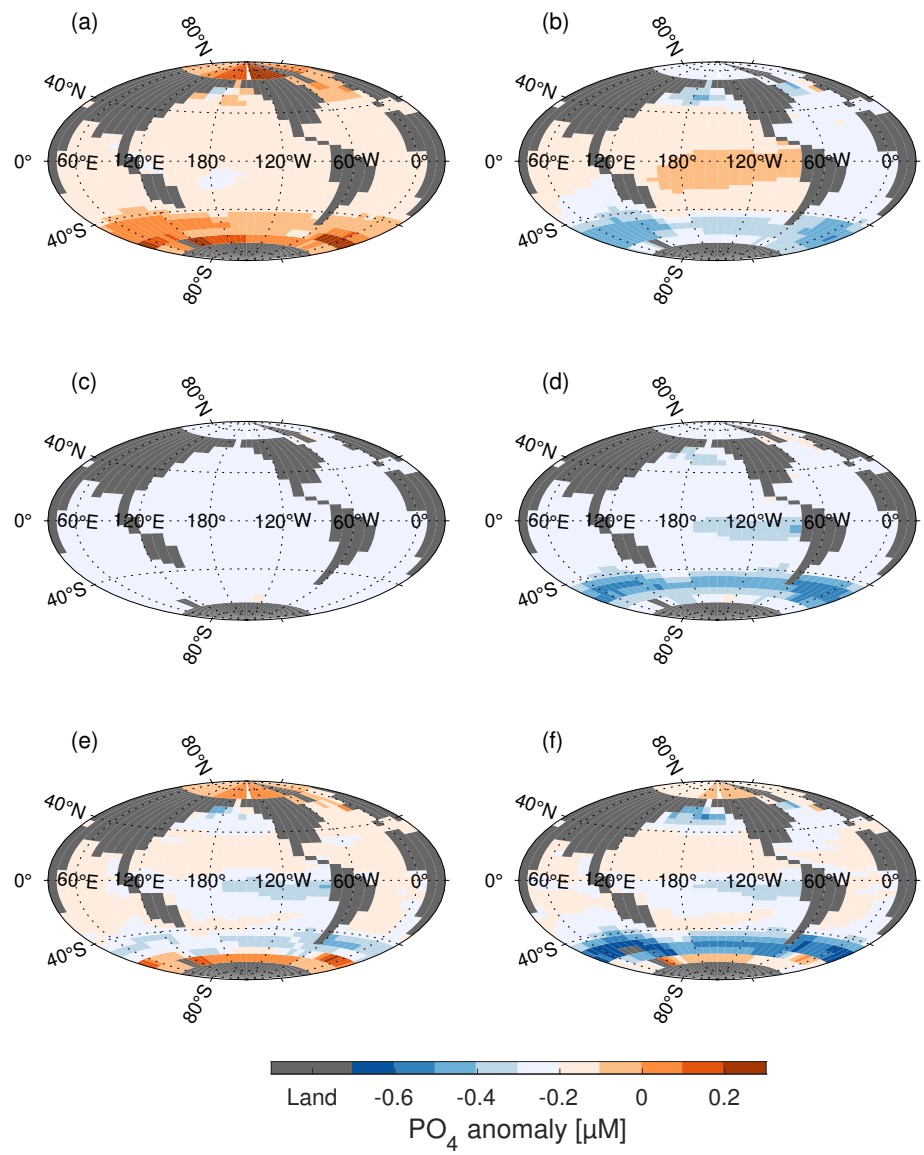

**Figure 7.** Surface $PO_4$ anomaly [$\mu M$], with respect to surface concentration of $PO_4$ in $Ctrl_{GAM}$ (Fig. 3a), for a) $LGMphy_{GAM}$, b) $WSN \times 0.5_{GAM}$ c) $RLS \times 1.25_{GAM}$, d) $LGMdust_{GAM}$, e) $Acomb_{GAM}$, f) $GLcomb_{GAM}$. Changes in surface nutrient fields are similar for all three model versions ($RED$, $GAM$, 121), thus only $GAM$ is shown.

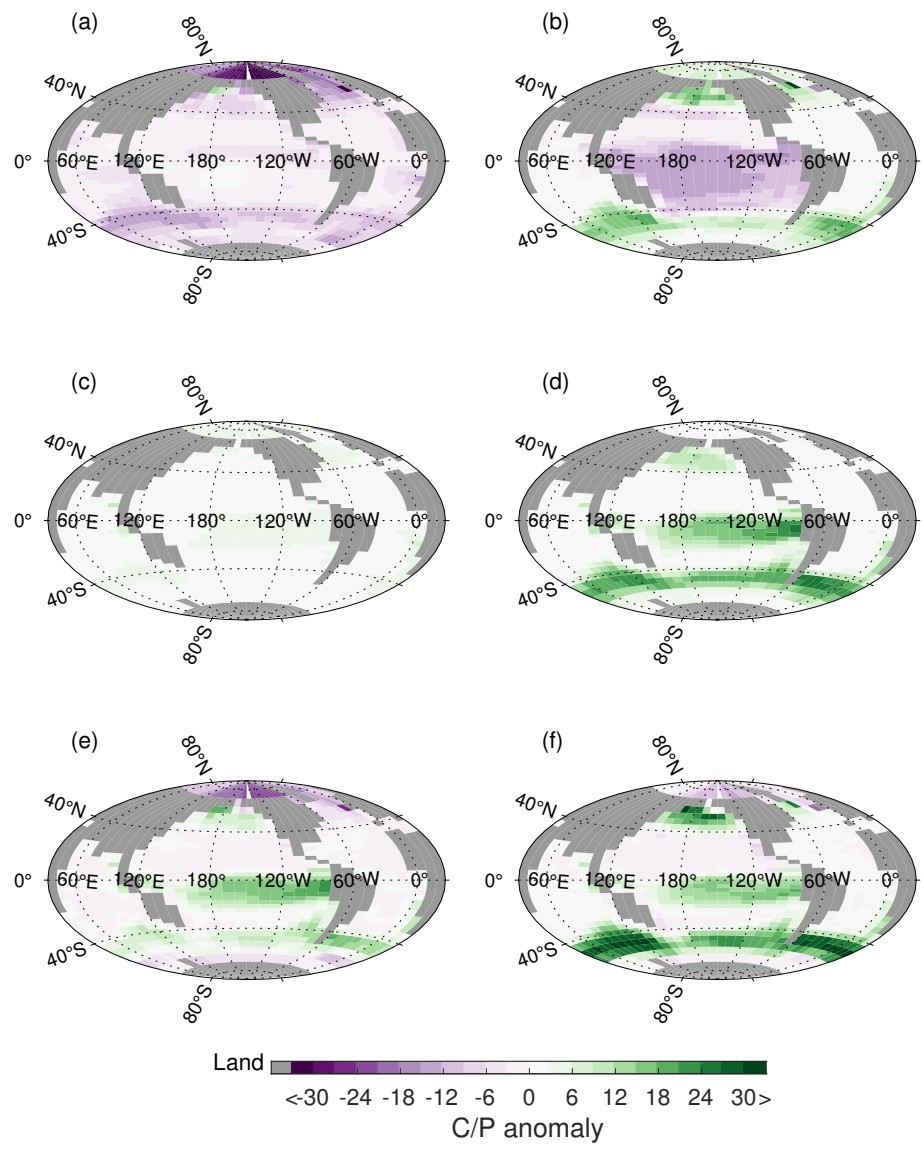

**Figure 8.** Surface $C/P$ anomaly, with respect to C/P of $Ctrl_{GAM}$ (Fig. 3b), for a) $LGMphy_{GAM}$, b) $WSN \times 0.5_{GAM}$ c) $RLS \times 1.25_{GAM}$, d) $LGMdust_{GAM}$, e) $Acomb_{GAM}$, f) $GLcomb_{GAM}$.

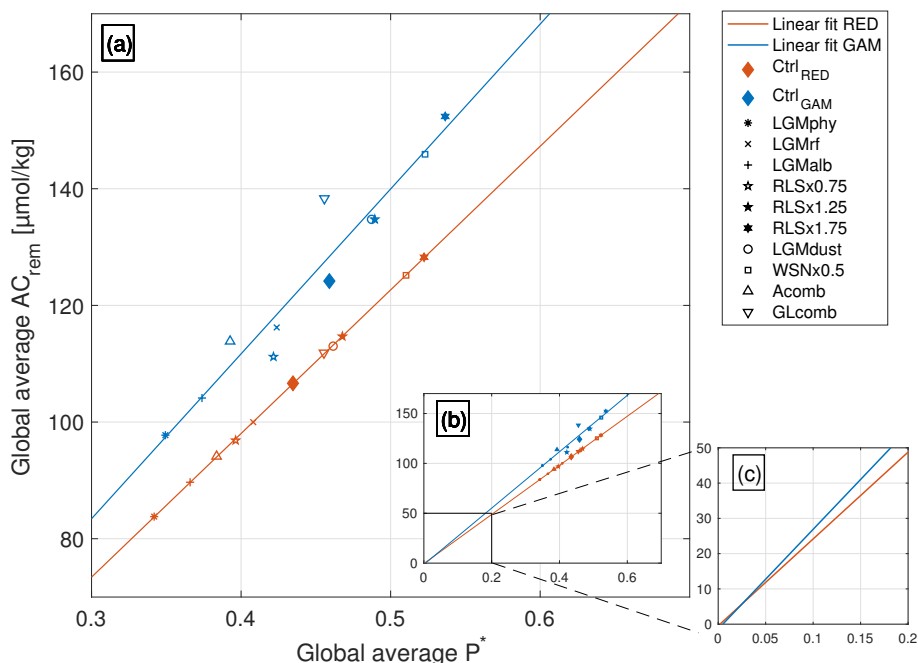

**Figure 9.** Remineralised acidic carbon ($AC_{rem} = DIC_{rem} - (1/2) * ALK_{rem}$, see Appendix A) as a function of $\overline{P*}$ . Simulations using model versions $RED$ and $GAM$ are shown in red and blue, respectively. Different symbols indicate the sensitivity experiments, listed in the panel on the right hand side. Red and blue lines show linear least–squares fits to the separate ensembles for $RED$ and $GAM$. Panels illustrate a) the deviation from the least–squares fit of the $GAM$ ensemble as opposed to the $RED$ ensemble, b) the linear fits extrapolated to origo, and c) a zoom–in around the origin, where the $RED$ (red) line goes through the origin, while the $GAM$ (blue) does not.

**Table 1.** List of experiments. Ensemble member acronyms, short descriptions of what the simulation tests, and specifications of parameter settings for each ensemble member. The $pCO_2^{atm}$ is either prescribed to pre-industrial (PI) = 278 ppm, or freely varying with changes in climate and ocean circulation. The radiative forcing is either coupled to the $pCO_2^{atm}$ of the atmospheric chemistry module of the model, or fixed at a value corresponding $pCO_2^{atm} = 185$ ppm. The zonal albedo profile is either representative of modern/PI conditions or of the LGM. The wind stress is either modern/PI, or has an adjusted peak in wind stress at $\pm 50°$ N. The modern/PI remineralisation length scale (RLS) is 590 m. If RLS is changed, it is multiplied by a factor $fr$. Dust forcing is either modern/PI or representative of LGM. Each experiment is conducted using model versions with C/P fixed at 106/1 (Redfield, 1963, , denoted *RED*), C/P variable with surface ocean $PO_4$ concentration (Galbraith and Martiny, 2015, denoted *GM*15), and C/P fixed at 121/1 (denoted 121, see Section 2.3).

| Ensemble member | Short description | $pCO_2^{atm}$ | Radiative forcing | Zonal albedo | Wind stress[1] at ±50°N | RLS[2] (m) | Dust forcing |
|---|---|---|---|---|---|---|---|
| *Ctrl* | Control | 278 ppm (restored) | coupled to $pCO_2^{atm}$ | modern/PI | modern/PI | 590 | modern/PI |
| *LGMrf* | LGM radiative forcing | variable | fixed at 185 ppm | modern/PI | modern/PI | 590 | modern/PI |
| *LGMalb* | LGM zonal albedo | variable | coupled to $pCO_2^{atm}$ | LGM[3] | modern/PI | 590 | modern/PI |
| *LGMphy* | LGMrf+ LGMalb | variable | fixed at 185 ppm | LGM | modern/PI | 590 | modern/PI |
| *WNS × 0.5* | Wind stress at ±50°N reduced | variable | coupled to $pCO_2^{atm}$ | modern/PI | PI ×0.5 | 590 | modern/PI |
| *RLS × fr*[1] | RLS changed by factor $fr$ | variable | coupled to $pCO_2^{atm}$ | modern/PI | modern/PI | $590 \times fr$ | modern/PI |
| *LGMdust* | LGM dust flux | variable | coupled to $pCO_2^{atm}$ | modern/PI | modern/PI | 590 | LGM[4] |
| *GLcomb* | "Glacial-like" (GL) (all forcings) | variable | fixed at 185 ppm | LGM | PI ×0.5 | $590 \times 1.25$ | LGM |
| *Acomb* | "GL" with PI wind | variable | fixed at 185 ppm | LGM | modern/PI | $590 \times 1.25$ | LGM |

1) See Lauderdale et al. (2013) for example of reduced peak wind profile for the Southern Hemisphere.

2) $fr$ = multiplication factor for remineralisation length scale. We test multiplication factors between $0.75$ and $1.75$, corresponding to a change in RLS between -25 % to +75 %.

3) Calculated from HADCM3 LGM (21ka) simulation of Davies-Barnard et al. (2017)

4) Re-gridded LGM dust flux from Mahowald et al. (2006)

**Table 2.** Atmospheric $CO_2$ ($pCO_2^{atm}$, ppm), global ocean averages of temperature ($\overline{T_{oce}}$, °C), $\overline{P^*}$ and $\overline{O_2}$ ($\mu mol\ kg^{-1}$), and Atlantic overturning streamfunction ($\psi$, Sverdrups (Sv)), maximum and minimum north of -30 °N, for observations, and for selected ensemble members in each model version ($RED and GAM$). Observed modern day $\overline{T_{oce}}$ and $\overline{O_2}$ were computed using the World Ocean Atlas 2018 (Locarnini et al., 2018; Garcia et al., 2018c). $\overline{P^*}$ for the modern day ocean is estimated by Ito and Follows (2005). Average modern day AMOC strength is estimated by McCarthy et al. (2015) from the RAPID-MOCHA array at 26 °N (corresponding to Atlantic $\psi_{max}$ in the model). Note that observed $pCO_2^{atm}$ is given for pre-industrial (PI) climate, as we do not model anthropogenic release of $CO_2$.

| Ensemble member | $pCO_2^{atm}$ (ppm) | $\overline{T_{oce}}$ (°C) | $\overline{P^*}$ | $\overline{O_2}$ ($\mu mol\ kg^{-1}$) | Atlantic $\psi_{max}$ (Sv) | Atlantic $\psi_{min}$ (Sv) |
|---|---|---|---|---|---|---|
| Obs. modern | 278 (PI) | 3.49 | 0.36 | 172 | 17.2±0.9 | - |
| Ensemble member | | | | | | |
| $Ctrl_{RED}$ | 278.0 | 3.56 | 0.43 | 166 | 14.2 | -0.8 |
| $Ctrl_{GAM}$ | 278.0 | 3.56 | 0.46 | 144 | 14.3 | -0.7 |
| $LGMphy_{RED}$ | 245.4 | 1.45 | 0.34 | 171 | 13.7 | -0.4 |
| $LGMphy_{GAM}$ | 244.9 | 1.45 | 0.35 | 149 | 13.7 | -0.4 |
| $RLS \times 1.25_{RED}$ | 264.0 | 3.50 | 0.47 | 155 | 14.4 | -0.5 |
| $RLS \times 1.25_{GAM}$ | 262.3 | 3.49 | 0.49 | 129 | 14.5 | -0.4 |
| $LGMdust_{RED}$ | 262.2 | 3.48 | 0.46 | 152 | 14.4 | -0.5 |
| $LGMdust_{GAM}$ | 256.9 | 3.47 | 0.49 | 124 | 14.5 | -0.4 |
| $WNS \times 0.5_{RED}$ | 265.1 | 3.98 | 0.51 | 141 | 12.7 | -0.8 |
| $WNS \times 0.5_{GAM}$ | 261.7 | 3.99 | 0.52 | 114 | 12.7 | -0.7 |
| $Acomb_{RED}$ | 222.5 | 1.45 | 0.48 | 148 | 13.7 | -0.4 |
| $Acomb_{GAM}$ | 214.9 | 1.45 | 0.39 | 116 | 13.7 | -0.4 |
| $GLcomb_{RED}$ | 214.1 | 1.94 | 0.46 | 122 | 12.0 | -0.4 |
| $GLcomb_{GAM}$ | 198.2 | 1.91 | 0.46 | 74 | 12.1 | -0.4 |