# Peer review of "Variable C/P composition of organic production and its effect on ocean carbon storage in glacial-like model simulations"

_Biogeosciences, 2019_

## Referee Comment (RC1) · Anonymous Referee #1 · 22 Jun 2019

Odalen et al., study the impact of a variable C/P on ocean carbon storage during glacial times. The variable C/P is combined with other mechanisms possibly leading to a glacial atmospheric $CO_2$ decrease, such as halved winds, a deepening of the remineralization depth, and increased iron fertilization. To my knowledge this is the first systematic study of the impact of variable C/P. This is well-written and interesting manuscript. I thus recommend publication in Biogeosciences, provided the comments below are taken into account.

Introduction: It might be good to add a sentence detailing the evidence for lower terrestrial carbon storage during glacial times (p2, L.1).

[Figure]

Section 2.3.1: It would be good to precise whether the wind changes impact the air-sea gas exchange of $CO_2$.

Methods and section 3.2.1: If I understand correctly global salinity is not increased during glacial times. If correct, it might be good to clearly state it as well as its impact on solubility changes.

Section 3.2.3: p10, L. 29: Please quantify magnitude and direction of "small" p11, L.3-4: This is an interesting result that should be emphasized. p10, L. 31-32: This sentence is unclear p11, L. 11: Shouldn't iron fertilization lead to an increase in Prem (instead of P*)? p11, L. 16: It is unclear what you mean here with "radionuclide proxy data"

Section 4.2: - Please consider amending the title of that section - I would suggest to add all the results of experiment 121 here and thus all the finishing sentences of the diverse paragraph (ex: p10, L. 31-32).

Figure 5: I'm confused as to what is shown here. I think mistakes have been made in the plots or legends as it does not make any sense. How can both HOL and LGM can be shown for CTR? How can both HOL and LGM can be shown for GLcomb? It is really not obvious LGM Pacific is HOL Pac-0.32 permil (h compared to f). Similarly how do you go from Hol Pac to LGM Pac in CTR (g compared to e)? g looks much more like an Atlantic section than a Pacific one. How can d be LGM Atl and f) Hol Pac? d) might be Pacific.

Figure 9: With a fixed Redfield ratio ACrem should increase with Prem. I am confused as to why ACrem increases with P* here.

Table 2: I find the format of this table not ideal and wonder if it would make sense to split the RED and GAM results. Also, it might not be necessary to show the AMOC strength for both experiments. The AABW transport in the Atlantic is extremely low (where did you take it?).

Minor points and typos: Section 2.3.3, L. 17: Missing table number P6, L. 22: "retrieved" P7, L. 14: missing reference P8: Sverdrup is usually noted "Sv" P11, L. 10: remove one "the"

---

## Referee Comment (RC2) · Pearse Buchanan (Referee) · 7 Jul 2019

Pearse James Buchanan

July 7, 2019

**Department of Earth, Ocean and Ecological Sciences, University of Liverpool, L69 3GP, United Kingdom.**

In their manuscript, Odalen and coauthors explore the role of variable C:P stoichiometry for carbon storage in the ocean under glacial conditions using an Earth System Model of Intermediate Complexity. Oceanic sequestration of carbon is almost certainly responsible for the recurring glacial-interglacial cycles that have been observed in sediment and ice-core records. While we know that carbon entered and exited the ocean over these cycles, we are still understanding why. What mechanisms caused the regular recurring reorganisation of the carbon system? Many mechanisms have been proposed and are still being proposed, but the suite of mechanisms that are of primary importance has alluded the palaeo-community for some time. Now, many scientists are working steadily towards "putting the pieces of this grand puzzle" together. Variable stoichiometry of organic matter, a phenomenon known for many years and even acknowledged by Redfield himself (often forgotten), is one possible piece that may help to explain why carbon entered (and more importantly stayed!) in the ocean for many thousands of years. It offers a positive-feedback under stratified and iron-rich glacial conditions to drive and **hold** $CO_2$ within the ocean. Variable stoichiometry achieves this by increasing the carbon that is assimilated into organic matter per unit of phosphorus, making the biological carbon pump more efficient. Effectively, a higher mean C:P ratio "tightens the lid".

The manuscript presents results from numerous simulations that have importantly been run to steady state in a logical progression. The simulations involve controls, physical perturbations and biological perturbations that are run in isolation and in combination. In this sense, the manuscript is easy to follow. The overarching finding of the paper is that variable C:P ratios make an important contribution to $p\mathrm{CO_2}$ decline under glacial conditions of roughly 15 ppm. This phenomenon therefore cannot be ignored, and the manuscript makes a meaningful contribution to the growing body of work that is highlighting the importance of biological stoichiometry on biogeochemistry and climate.

I therefore **advocate** for publication in *Climate of the Past*. However, I suggest *further*

*revisions* that I believe would improve the readability and hence its dissemination within the field. My comments for improvement (and sometimes clarification) are below.

**1 General comments**

The manuscript is well organised in how it presents the methods and results. In this sense, it is very easy as a reader to understand what the authors have done. However, the writing itself requires a lot of polishing. The common usage of "e.g.", as an example (pun not intended), in mid-sentence makes things simultaneously difficult to read and also leaves me thinking that the authors are purposely choosing to not discuss all the evidence/processes/knowledge on a subject. I strongly suggest that the authors take some time to improve the writing.

One particularly important part of the manuscript that is not conveyed clearly is their diagnosis of $CO_2$ capture via variable C:P ratios, as opposed to a simple whole ocean increase in C:P. For instance, in the conclusion the authors state that "About half of the increased drawdown of CO2 results from different global average C/P in the export production. In addition, flexible stoichiometry allows increased carbon capture through the biological pump, while maintaining or even decreasing the fraction of remineralised to total nutrients in the deep ocean.". The reader is therefore left confused about what C:P ratios are actually doing. Aren't they contributing to the *whole* of the *increased* drawdown? I think what the authors are trying to say is that the spatial and temporal variations in C:P are important for strengthening the biological pump, because they react dynamically to changes in nutrient supply to reinforce further $CO_2$ drawdown over a simple prescribed whole ocean increase. I would therefore advocate for the authors to make this more clear.

I also think that many of the concepts discussed in this paper with relevance for the sensitivity experiments could be introduced better in the Introduction. This includes previous glacial modelling studies, paloeproxy evidence and the theories that have been consequently generated for increased oceanic carbon storage due to these prior studies. Temp-dependent remin rates, changes in wind stresses, overturning circulation, polar stratification, solubility (salt and heat), Si-leakage, $CaCO_3$ compensation and production rates, Fe fertilisation, increased $N_2$ fixation, sea ice expansion, even volcanism. All of these, except volcanism, are feedbacks that were somehow kick-started by changes in solar insolation. While you obviously do not need to discuss all of these in detail, laying out the current "pieces of the puzzle" would help to ground your work in the current stream of consciousness in the paleoclimate community.

**2 Specific comments**

**Abstract**

- Page 1, line 8 : surely you mean increases rather than decreases? And also you mean Phosphate. Because an increase in Fe deposition, which is a nutrient, has been linked to an increase in C:P ratios. Garcia et al (2018) Nutrient supply controls particulate elemental concentrations in the low latitude eastern Indian Ocean. *Nature Communications*.

**Introduction**

- Page 2, line 2 : Also because of the rapid release of carbon to the atmosphere over the deglaciation, implying its storage somewhere during the glacial.
- Page 2, line 5 : What is the e.g. here referring to?

**Methods**

- Page 4, line 20 : suggest citing the more recent estimate of 2.6°C by Bereiter et al (2018) Mean global ocean temperatures during the last glacial transition. *Nature*.
- Page 6, line 1 : suggest citing Moore et al (2013) Processes and patterns of oceanic nutrient limitation. *Nature Geoscience*. Another paper of note that would be useful for your work is the recent article by Garcia et al (2018) Nutrient supply controls particulate elemental concentrations in the low latitude eastern Indian Ocean. *Nature Communications*.
- Page 6, line 17 : "see Table ??"... Please make sure your document is properly formatted before submitting.
- Page 6, line 22 : Why not use more recent WOA 2018 product?
- Page 7, line 13 : Again a question mark is present. Please format properly before submitting.

**Results**

- Page 8, line 29 : The oxygen content of the ocean should be lower in $Ctrl_{121}$ than for $Ctrl_{RED}$. This is because a higher C:P ratio of organic matter should also require more $O_2$ to remineralise that organic matter. I would like an explanation of why $O_2$ is higher in $Ctrl_{121}$ than for $Ctrl_{RED}$. See Paulmier, Kriest & Oschlies (2009) Stoichiometries of remineralisation and denitrification in global biogeochemical ocean models. *Biogeosciences*.
- Page 11, line 1 : Please explain why a shallow RLS is shallower in a warmer climate. Alternatively, you could provide a more thorough explanation of the effect of temperature on remineralisation rates in the Introduction.
- Page 11, line 9 : This sentence needs to be clearer with what it's trying to say. Roughly 10% of what? Change in the average composition of what? Of course I can guess what you mean when I stop to think about it, but please make it easier for the reader by saying what you mean.
- Page 11, lines 22-25 : But not cool enough to align with the more recent estimate of Bereiter et al (2018) Mean global ocean temperatures during the last glacial transition. *Nature*.
- Page 12, lines 21-25 : Again, I am unsure how you are treating P:$O_2$ remineralisation requirements in your variable stoichiometry experiments. I think this should be explained. It is also strange once again that your $GLcomb_{121}$ experiment is better oxygenated than your $GLcomb_{RED}$ experiment.
- Figure 5 : The panels in this figure do not seem to be arranged correctly.
- Page 13, lines 1-8 : What conditions affect the fractionation strength of biological carbon assimilation? Is it constant or variable?

- Page 13, line 12 : Why not add a figure of sea ice cover in the supplement? Also, can you separate the effects of sea ice cover expansion from the other physical changes in terms of $CO_2$? A few studies since the Stephens & Keeling (2000) paper have found that an increase in sea ice cover under glacial conditions actually reduces ocean carbon storage because it prevents organic carbon production. It would be worthwhile to separate this effect from temperature and circulation and note if it is positive or negative on atmospheric $CO_2$. Stephens & Keeling (200) The influence of Antarctic sea ice on glacial-interglacial CO2 variations. *Nature*. Kurahashi-Nakamura et al (2007) Compound effects of Antarctic sea ice on atmospheric pCO2 change during glacial-interglacial cycle. *Geophysical Research Letters*. Sun & Matsumoto (2010) Effects of sea ice on atmopsheric pCO2: A revised view and implications for glacial and future climates. *Journal of Geophysical Research*. Buchanan et al (2016) The simulated climate of the Last Glacial Maximum and insights into the global marine carbon cycle. *Climate of the Past*.

- Page 13, lines 10-23 : This paragraph would benefit from being clearer in its findings of $CO_2$ sequestration regarding C:P ratio changes. I have to read this multiple times to understand what the authors are trying to say when comparing Redfield, variable C:P and C:P=121.

**Discussion**

- Page 13, line 28 : your reference to "model" should be an "empirical model" to avoid confusion with the Earth System Model, GCM, etc.

- Page 14, lines 1-5 : The advantages of using empirical/statistical models within biogeo-chemical ocean GCMs, including the Galbraith & Martiny (2015) parameterisation, was explored rigorously in my 2018 paper in Global Biogeochemical Cycles. It not only improved that model's biogeochemistry significantly, but also altered the long-term behaviour of the carbon cycle as you have also found. It may be interesting, but I of course leave it up to you whether it's useful. Buchanan et al (2018) The importance of dynamic biological functioning for simulating and stabilizing ocean biogeochemistry. *Global Biogeochemical Cycles*.

- Page 14, line 32 : the conclusions of Odalen et al (2018)... which were? What did Odalen et al (2018) do?

- Page 15, lines 19-24 : So the proportion of remineralised to preformed phosphorus effectively doesn't change in the simulations? And yet, you find a large increase in respired C? This must mean that the remineralised phosphorus that is exported into the ocean interior in your *GLcomb* simulation is being quickly circulated into the lower overturning cell and returned to the Antarctic Zone, where sea ice prevents gas exchange and biological production, at which point this P is recirculated and becomes preformed, while respired C remains respired and is also recirculated. If this is the case, it merits more discussion in comparison with previous literature on the subject of a more efficient biological pump that invokes more regenerated nutrients as a *must* for a more efficient biological pump. I suggest Hain, Sigman & Haug (2014) The biological pump in the past. *Treatise on Geochemistry, $2^{nd}$ Ed.*.

- Page 16, lines 1-14 : I would like to see how variations in $P:O_2$ requirements were treated in this model. Also, can you please explain why the deep water formation characteristics of a model affects $O_2$? Overall, I find this section a bit sparse and I'm not entirely sure what the point of it is. Can you comment on the size of the OMZs? New evidence shows that

the OMZs in the Pacific expanded vertically during the glacial. Hoogakker et al (2018) Glacial expansion of oxygen-depleted seawater in the eastern tropical Pacific. *Nature*.

- Page 16, line 22 : And yet others achieve relatively good correlations in the Pacific basin using the same data? Menviel et al (2017) Poorly ventilated deep ocean at the Last Glacial Maximum inferred from carbon isotopes: A data-model comparison study. *Paleoceanography*. Muglia et al (2018) Weak overturning circulation and high Southern Ocean nutrient utilization maximized glacial ocean carbon. *Earth and Planetary Science Letters*. I think you cannot say that your model doesn't provide good fits to the LGM proxy data because of poor data coverage, and it would be more useful to discuss why the model doesn't fit with the data. It seems like your glacial circulation in the Pacific basin is therefore not accurate?

- Page 16, line 26 : Temperature is not chemical. I also do not understand how you could alter the temperature and salinity of the ocean without altering water mass distributions, and if this is indeed the case, it requires further description as to why earlier in the paper. Also a good spot to talk about why the data in the Pacific are not well reproduced by the model.

- Page 16, lines 24-31 : I don't follow this paragraph. You state that "Each of the two observational datasets (HOL and LGM) display similar correlations across the two model simulations. This implies that our changes in forcings do not achieve any obvious changes in water mass distribution." But doesn't the distribution of $\delta^{13}$C change across the glaciation and into the Holocene? $\delta^{13}$C in the Atlantic, for instance, is often used to show that the Atlantic meridional overturning was shallower during the glacial, and that this change occurred during Marine Isotope Stage 4 (Oliver et al (2010) A synthesis of marine sediment core $\delta^{13}$C data over the last 150000 years. *Climate of the Past*.)? Moreover, $\delta^{13}$C is used as a way to show that the water mass distribution between the Atlantic and Pacific was considerably different during the glacial as compared to the Holocene (Sikes et al (2017) Enhanced $\delta^{13}$C and $\delta^{18}$O Differences Between the South Atlantic and South Pacific During the Last Glaciation: The Deep Gateway Hypothesis. *Paleoceanography*.) These studies conflict with what you are saying.

- The ability for simulated $\delta^{13}$C to reproduce the proxy data at the LGM will depend strongly on water mass distribution (which apparently doesn't change appreciably) and how biological fractionation is parameterised. If it is constant, the 10% loss in C fixation will cause the ocean to be more positive overall by some constant factor. However, if the parameterisation contains a dependence on aqueous $CO_2$ and growth rate, both of which are lower, then the fractionation will vary. It would be worthwhile telling the reader what parameterisation is used and, if it does involve growth rate and aqueous $CO_2$, what effect this has.

**Conclusion**

- No comments

**3 Technical corrections**

1. Page 1, line 13 : remove repeated "with"
2. Page 5, line 16 : replace "reduced half" with "halved"

3. Table 1 : "witg" to "with"

4. Page 11, line 8 : Inadvisable to begin a sentence with "~"

5. Page 11, line 29 : "SV" to "Sv"

6. Page 13, line 26 : "GCMs" this acronym has not been defined previously.

7. Page 15, line 23 : I assume you mean "0.003" rather than "0003"?

---

## Author Comment (AC1) · 14 Oct 2019

**1   Introduction**

We thank Anonymous Referee #1 for the helpful comments and for the compliments on our paper. In this response, we will respond to the comments by the referee in the order the commented parts appear in the manuscript.

[Figure]

**2   General comments**

- ***Introduction: It might be good to add a sentence detailing the evidence for lower terrestrial carbon storage during glacial times (p2, L.1).***
  We agree and will add to the introduction: *Studies of paleoproxy records indicate that carbon storage in the glacial terrestrial biosphere was smaller compared to in interglacial climate* (Shackleton, 1977; Duplessy et al., 1988; Curry et al., 1988; Crowley, 1995; Adams and Faure, 1998; Ciais et al., 2012; Peterson et al., 2014).

- ***Section 2.3.1: It would be good to precise whether the wind changes impact the air–sea gas exchange of $CO_2$.***
  Yes. In cGENIE, gas transfer velocities are calculated as a function of wind speed (described in Ridgwell et al., 2007), and following Wanninkhof (1992). We will add this information to Section 2.3.1.

- ***Methods and section 3.2.1: If I understand correctly global salinity is not increased during glacial times. If correct, it might be good to clearly state it as well as its impact on solubility changes.***
  This is correct. We do not aspire to simulate a full glacial state, but rather to explore the effect of flexible C/P for biological carbon capture in response to a few common glacial forcings. As we do not change salinity, we are likely to overestimate the increase in solubility between $Ctrl$ and $GL_{comb}$, by $\sim 6$ ppm (Kohfeld and Ridgwell, 2009). This effect is consistent for any choice of C/P parametrisation, and is therefore not explored further. We will add this information to Section 3.2.1.

- ***Section 3.2.3: p. 10, L. 29: Please quantify magnitude and direction of "small".***
  We will clarify the sentence by changing it to: *Increases in RLS cause very small, but global, decreases in surface $PO_4$ concentrations (global average anomaly =*

*-0.016 $\mu M$) [...]*

- **Section 3.2.3: p. 11, L. 3–4: This is an interesting result that should be emphasized.**
  We add here: *The potential implications of this result for warm climate scenarios is further discussed in Section 4.1.*
  In Section 4.1, we add: *Our sensitivity experiments $RLS \times 0.75$ and $RLS \times 1.25$, reveal that the response in $pCO_2^{atm}$ to the perturbation is enhanced in $GAM$ compared to $RED$ for both increased and decreased RLS. While increased $RLS$ would be an effect of ocean cooling, and thus of interest for glacial studies, reduced $RLS$ would be a consequence of ocean warming (Matsumoto, 2007). Matsumoto (2007) describes how decreased RLS would have a positive feedback on $pCO_2^{atm}$ in future warming climate. Our results imply that flexible C/P could have a further re–inforcing effect on this feedback. It would therefore be of interest to apply a parametrisation of flexible C/P in models used for simulations of future climate feedbacks.* We also add the following sentence to Conclusions: *Flexible C/P also has the potential to be an additional positive feedback of ocean warming on $pCO_2^{atm}$ in future climate.*

- **Section 3.2.3: p. 10, L. 31–32: This sentence is unclear.**
  We suggest clarifying this sentence by changing it to: *Experiments with deeper RLS in 121 ($RLS \times 1.25_{121}$ and $RLS \times 1.75_{121}$), suggest that about 40 % of the observed differences in $pCO_2^{atm}$ between $GAM$ and $RED$ can again be attributed to the difference in export flux average C/P.*

- **Section 3.2.3: p. 11, L. 11: Shouldn't iron fertilization lead to an increase in Prem (instead of $P^*$)?**
  As $\overline{P^*} = \overline{P_{rem}}/\overline{P_{tot}}$, and $P_{tot}$ is constant, an increase in $P_{rem}$ is equal to an increase in $P^*$. We will clarify the sentence by changing it to: *[...] the iron added by the dust forcing allows more efficient usage of P in the HNLC-regions, which*

increases the ocean storage of biologically sourced carbon $P_{rem}$ (thus, $P^*$ increases).

- **Section 3.2.3: p. 11, L. 16: It is unclear what you mean here with "radionuclide proxy data"**
  We will clarify the sentence by changing it to: *This subantarctic increase in biological efficiency is consistent with radionuclide proxy data ($^{10}$Be, $^{230}$Th, $^{231}$Pa) from the LGM [...].*

- **Section 4.2: - Please consider amending the title of that section - I would suggest to add all the results of experiment 121 here and thus all the finishing sentences of the diverse paragraph (ex: p. 10, L. 31–32).**
  We agree that grouping all the results of experiment 121 and the associated discussion in Section 4.2 is a good idea, and will follow this recommendation. We will thereby also change the title of the subsection to *Effect of modified but fixed C/P.*

- **Figure 5: I'm confused as to what is shown here. I think mistakes have been made in the plots or legends as it does not make any sense. How can both HOL and LGM can be shown for CTR? How can both HOL and LGM can be shown for GLcomb? It is really not obvious LGM Pacific is HOL Pac -0.32‰ (h compared to f). Similarly, how do you go from Hol Pac to LGM Pac in CTR (g compared to e)? g looks much more like an Atlantic section than a Pacific one. How can d be LGM Atl and f) Hol Pac? d) might be Pacific.**
  In this figure, the sub–panels have by mistake been shifted to the wrong positions, which naturally causes unnecessary confusion. We apologise for the mistake and show in Fig. 1 the corrected version of the figure, which will replace Fig. 5 in the revised manuscript. The referee is also confused by how both time slices HOL and LGM can be shown for the $Ctrl$-simulation. This is simply because we have chosen to compare each of the simulations ($Ctrl$ and $GLcomb$) with

both time slices (HOL and LGM) of the proxy data. Even though we expect, of course, that $Ctrl$ should to a higher extent reproduce the patterns we see in the HOL data than in the LGM data, we do not want to assume that the model is successful in this respect. For transparency in the process, we therefore show both comparisons. In order to present this figure in a more accessible way, we will clarify that the columns represent the two model simulations, and that the rows represent the proxy records to which we compare the simulations (see updated caption on page C9 of this comment). Note that the update of the caption will also apply to Fig. S3.

Also, it seems the referee may have misunderstood our subtraction of -0.32‰, though the confusion may be a result of the panels being organised in the wrong order. The contours in panel f shows the Pacific Ocean of the $GLcomb$ simulation compared to HOL Pacific proxy records (circles). The contours in panel h also shows the Pacific Ocean of the $GLcomb$ simulation, but here, 0.32‰ has been subtracted from the simulation data. In panel h, the circles show LGM Pacific proxy records. We describe in Section 2.4 that proxy records of $\delta^{13}C$ indicate that the LGM ocean was more depleted in $\delta^{13}C$ than the Holocene ocean. We will clarify that this is also true for the dataset we use here (see Peterson et al., 2014). Gebbie et al. (2015) estimated this difference in whole–ocean $\delta^{13}C$ to -0.32±0.20‰. The low LGM whole–ocean value is attributed to glacial contraction of the terrestrial biosphere, and an associated addition of $\delta^{13}C$–depleted carbon of terrestrial origin to the ocean. As we do not simulate this terrestrial contribution of $\delta^{13}C$–depleted carbon, we do not expect our $GLcomb$–simulations to reproduce this change in whole–ocean $\delta^{13}C$. We therefore subtract 0.32‰ from each point of the $GLcomb$ simulation output, before we compare to the LGM proxy records.

- **Figure 9: With a fixed Redfield ratio $AC_{rem}$ should increase with $P_{rem}$. I am confused as to why $AC_{rem}$ increases with P\* here.**

As stated above, $\overline{P^*} = \overline{P_{rem}/P_{tot}}$, thus an increase in $P_{rem}$ is analogous to an increase in $P^*$.

- **Table 2: I find the format of this table not ideal and wonder if it would make sense to split the RED and GAM results. Also, it might not be necessary to show the AMOC strength for both experiments. The AABW transport in the Atlantic is extremely low (where did you take it?).**
  We agree, and will re–organise the table to show the GAM results below RED results, rather than showing them side by side. However, if the results are presented on different rows, the table would look incomplete if we out the AMOC strength for one of the model versions. We would therefore prefer to keep those results in the table. $\psi_{min}$ in this table is simply the minimum of the Atlantic overturning streamfunction below below 556m depth and north of 30°N. As seen in Fig. 2 c–d, the AABW circulation is weak in the Atlantic north of 30°N, while its peak strength is located in the Southern and Pacific Oceans. Due to the lack of a boundary between the Atlantic and the Pacific south of 30°N, it is not possible to compute the basin–specific streamfunction further south, though the Atlantic AABW circulation is likely to be stronger there.

**3   Minor points and typos**

- **Section 2.3.3, p. 6, L. 17: Missing table number**
  Here, *Table ??* should be corrected to *Table 2*.

- **Section 2.3.3, p. 6, L. 22:**
  Typo, *retreived* should be corrected to *retrieved*.

- **Section 2.5, p. 7, L. 13:**
  Missing reference marked by *(?)* should be (Ödalen et al., 2018).

- ***Section 3.1.1, p. 8: Sverdrup is usually noted "Sv".***
  *SV* will be changed to *Sv* throughout the paper.

- ***Section 3.2.4, p. 11, L. 10: remove one "the".***
  Typo, will be corrected.

**4 References**

Bouttes, N., Paillard, D., and Roche, D.: Impact of brine-induced stratification on the glacial carbon cycle, Climate of the Past, 6, 575–589, 2010.

Bouttes, N., Paillard, D., Roche, D. M., Brovkin, V., and Bopp, L.: Last Glacial Maximum $CO_2$ and $\delta^{13}C$ successfully reconciled, Geophysical Research Letters, 38, 2011.

Bouttes, N., Roche, D. M., and Paillard, D.: Systematic study of the impact of fresh water fluxes on the glacial carbon cycle, Climate of the Past, 8, 589–607, https://doi.org/10.5194/cp-8-589-2012, https://www.clim-past.net/8/589/2012/, 2012.

Matsumoto, K.: Biology-mediated temperature control on atmospheric pCO2 and ocean biogeochemistry, Geophysical Research Letters, 34, 2007.

Ödalen, M., Nycander, J., Oliver, K. I. C., Brodeau, L., and Ridgwell, A.: The influence of the ocean circulation state on ocean carbonstorage and $CO_2$ drawdown potential in an Earth system model, Biogeosciences, 15, 1367–1393, https://doi.org/10.5194/bg-15-1367- 2018, https://www.biogeosciences.net/15/1367/2018/, 2018.

Peterson, C. D., Lisiecki, L. E., and Stern, J. V.: Deglacial whole–ocean $\delta^{13}C$ change estimated from 480 benthic foraminiferal records, Paleoceanography, 29, 549–563, 2014.

Ridgwell, A., Hargreaves, J. C., Edwards, N. R., Annan, J. D., Lenton, T. M., Marsh, R., Yool, A., and Watson, A.: Marine geochemical data assimilation in an efficient Earth System Model of global biogeochemical cycling, Biogeosciences, 4, 87–104, 2007.

Ridgwell, A. J.: Glacial–interglacial perturbations in the global carbon cycle., Ph.D. thesis, University of East Anglia, 2001.

Zeebe, R. E. and Wolf-Gladrow, D. A.: $CO_2$ in seawater: equilibrium, kinetics, isotopes, 65, Gulf Professional Publishing, 2001.

[Figure]

**5  Figures**

**5.1  Fig. 5 - updated caption**

Model ocean $\delta^{13}C$ (contours) compared to the two proxy record time slices (HOL and LGM) of benthic $\delta^{13}C$ (circles) of Peterson et al. (2014). The upper half of the figure shows the Atlantic Ocean (panels a–d), while the lower half shows the Pacific Ocean (panels e–h). The columns represent the model simulations ($Ctrl_{RED}$ or $Ctrl_{GAM}$), while each row represents one of the proxy record time slices (HOL or LGM). The left hand column shows $Ctrl_{RED}$ (panels a, c, e, g), and the right hand column shows $GLcomb_{RED}$ (panels b, d, f, h). The rows show, from top to bottom, a–b) HOL Atlantic, c–d) LGM Atlantic, e–f) HOL Pacific, g–h) LGM Pacific. Note that, before we compare $GLcomb_{RED}$ to LGM observations (panels d and h), a constant of 0.32 ‰ is subtracted from the simulated $\delta^{13}C$, to account for terrestrial release of $\delta^{13}C$–depleted terrestrial carbon which is not modelled. The corresponding comparison for model version $GAM$ is shown in Fig. S.3.
* * *
Ctrl$_{RED}$   GLcomb$_{RED}$

(a)   (b)   HOL Atlantic proxy data

(c)   (d)   LGM Atlantic proxy data

(e)   (f)   HOL Pacific proxy data

(g)   (h)   LGM Pacific proxy data

-1.0 0.8 -0.6 -0.4 -0.2 0.0 0.2 0.4 0.6 0.8 1.0 1.2 1.4 1.6 1.8 2.0   ‰

⬭ Model output   ◯ Proxy data

**Fig. 1.** Updated version of Fig. 5. The caption of Fig. 5 is updated according to the text on the previous page. on

---

## Author Comment (AC2) · 14 Oct 2019

We thank Pearse James Buchanan (henceforth PJB) for a thourough and helpful review of our paper. In this response, we will respond to the comments by the referee in the order they were presented in the referee report.

[Figure]

**1 General comments**

As suggested by PJB, we will work to make the writing clearer. Specifically, PJB points out the common usage of e.g. mid–sentence, and questions whether this is intentionally used *"to not discuss all the evidence/processes/knowledge on a subject"*. This was not our intention, but rather an attempt to avoid the text getting too long. This is why we decided in some cases to only bring up the most relevant examples. We will go through these parts of the text in detail, and add more information.

Both Referee #1 and PJB find the discussion of Section 4.2 (Implications of changed average C/P) unclear, and both give useful suggestions on how to clarify this section. The section will thus be re–written based on these suggestions.

PJB details how the Introduction could be extended with a more extensive discussion of concepts, in order to help the reader. We agree that this could be helpful, and we will make additions according to PJB's suggestions.

**2 Specific comments**

**2.1 Abstract**

- **Page 1, line 8: surely you mean increases rather than decreases? And also you mean Phosphate. Because an increase in Fe deposition, which is a nutrient, has been linked to an increase in C:P ratios. Garcia et al (2018): Nutrient supply controls particulate elemental concentrations in the low latitude eastern Indian Ocean. Nature Communications.**
  Yes, "decreases" should be corrected to "increases", and "nutrients" to "phosphorus" (as we are discussing elemental ratios).

**2.2 Introduction**

- **Page 2, line 2 : Also because of the rapid release of carbon to the atmosphere over the deglaciation, implying its storage somewhere during the glacial.**
  We agree and will add this information, along with relevant references.

- **Page 2, line 5 : What is the e.g. here referring to?**
  This is a formatting issue. The *e.g.,* should be placed before the reference (Broecker, 1982a), which was intended to be given as an example of a key study for the role of ocean sedimentary processes for increased glacial oceanic storage of carbon.

**2.3 Methods**

- **Page 4, line 20 : suggest citing the more recent estimate of 2.6°C by Bereiter et al (2018): Mean global ocean temperatures during the last glacial transition. Nature.**
  We thank PJB for the suggestion and will add this reference.

- **Page 6, line 1 : suggest citing Moore et al (2013) Processes and patterns of oceanic nutrient limitation. Nature Geoscience. Another paper of note that would be useful for your work is the recent article by Garcia et al (2018) Nutrient supply controls particulate elemental concentrations in the low latitude eastern Indian Ocean. Nature Communications.**
  We thank PJB for the suggestions and add the reference to Moore et al. (2013) here. The paper by Garcia et al. (2018) is certainly relevant to our work, and this reference will be added already in Section 2.2 (Page 3).

- **Page 6, line 17 : "see Table ??"... Please make sure your document is**

*properly formatted before submitting.*
We apologise for the formatting error. *Table ??* should be corrected to *Table 2*.

• *Page 6, line 22 : Why not use more recent WOA 2018 product?*
When the paper was submitted, WOA18 was only available as a pre–release. We therefore chose to stay with the most up–to–date official release for the initial submission. The official release is now available, and we will therefore use WOA18 for the updated manuscript.

• *Page 7, line 13 : Again a question mark is present. Please format properly before submitting.*
We are very sorry for having overlooked the formatting errors in our final check before submission. The missing reference marked by *(?)* should be (Ödalen et al, 2018).

2.4  Results

• *Page 8, line 29 : The oxygen content of the ocean should be lower in $Ctrl_{121}$ than for $Ctrl_{RED}$. This is because a higher C:P ratio of organic matter should also require more $O_2$ to remineralise that organic matter. I would like an explanation of why $O_2$ is higher in $Ctrl_{121}$ than for $Ctrl_{RED}$. See Paulmier, Kriest & Oschlies (2009): Stoichiometries of remineralisation and denitrification in global biogeochemical ocean models. Biogeosciences.*
We thank PJB for having identified this inconsistency in our results, which we had failed to notice. This was caused by an issue in the code, which caused an unintentional change in $O_2$:C when the fixed C:P stoichiometry was changed. This issue was not present in the runs with flexible stoichiometry, and thus only affected the $O_2$ in the $121$–ensemble. We have re–run $Ctrl_{121}$ after having corrected the model code. The new resulting average $O_2$-concentration is 152 $\mu mol kg^{-1}$, which is lower than $Ctrl_{RED}$ (166 $\mu mol kg^{-1}$), but higher than $Ctrl_{GAM}$ (144 $\mu mol kg^{-1}$).

- ***Page 11, line 1 : Please explain why a shallow RLS is shallower in a warmer climate. Alternatively, you could provide a more thorough explanation of the effect of temperature on remineralisation rates in the Introduction.***
  In line with PJBs suggestion in the General comments, this will be described in more detail in the Introduction, based on the results of Matsumoto (2007).

- ***Page 11, line 9 : This sentence needs to be clearer with what it's trying to say. Roughly 10% of what? Change in the average composition of what? Of course I can guess what you mean when I stop to think about it, but please make it easier for the reader by saying what you mean.***
  We agree with PJB and in order to clarify the sentence, we will change it to:
  *About a third ($\sim 10$ % of 30 %) of the increase in drawdown can be explained by a change in average C/P composition of the organic material that is exported out of the surface ocean.* This will then be discussed in more detail in Section 4.2, where we will gather all the results and discussion that concerns the $121-$ ensemble (as suggested by Referee #1).

- ***Page 11, lines 22–25 : But not cool enough to align with the more recent estimate of Bereiter et al (2018): Mean global ocean temperatures during the last glacial transition. Nature.***
  We will add a note on this, and clarify that we are not applying all forcings that are expected to be needed to reproduce a full glacial state.

- ***Page 12, lines 21–25 : Again, I am unsure how you are treating P:O$_2$ remineralisation requirements in your variable stoichiometry experiments. I think this should be explained. It is also strange once again that your $GLcomb_{121}$ experiment is better oxygenated than your $GLcomb_{RED}$ experiment.***
  In remineralisation, there was an issue in the code which affected the C:O$_2$ ratio in the 121–experiments. This made the P:O$_2$ requirements appear strange, which

PJB noticed. This has been corrected, and the 121–experiments re–run. After correction, C:$O_2$ remains the same in all experiments, while the P:$O_2$ ratio, as a result, changes between experiments. We will clarify this in the Methods section. After the correction described abobe, $GLcomb_{121}$ has a lower global average $O_2$ concentration than $GLcomb_{RED}$ (96 compared to 122 $\mu mol kg^{-1}$), but higher than $GLcomb_{GAM}$ (74 $\mu mol kg^{-1}$).

- ***Figure 5 : The panels in this figure do not seem to be arranged correctly.***
  We are grateful to both referees for having identified this error. We here provide the updated figure (see Fig. 1), where the sub–panels have been re–arranged in the correct order. Per request of Referee #1, we will also clarify the caption of this figure (see Section 5).

- ***Page 13, lines 1–8 : What conditions affect the fractionation strength of biological carbon assimilation? Is it constant or variable?***
  The fractionation strength is variable and will be detailed in an appendix (see below, in author's response to the final bullet point of Discussion).

- ***Page 13, line 12 : Why not add a figure of sea ice cover in the supplement? Also, can you separate the effects of sea ice cover expansion from the other physical changes in terms of $CO_2$? A few studies since the Stephens & Keeling (2000) paper have found that an increase in sea ice cover under glacial conditions actually reduces ocean carbon storage because it prevents organic carbon production. It would be worthwhile to separate this effect from temperature and circulation and note if it is positive or negative on atmospheric $CO_2$. Stephens & Keeling (2000) The influence of Antarctic sea ice on glacial–interglacial $CO_2$ variations. Nature. Kurahashi–Nakamura et al. (2007) Compound effects of Antarctic sea ice on atmospheric $pCO_2$ change during glacial–interglacial cycle. Geophysical Research Letters. Sun & Matsumoto (2010) Effects of sea ice on atmopsheric $pCO_2$: A re-***

*vised view and implications for glacial and future climates. Journal of Geo-physical Research. Buchanan et al (2016) The simulated climate of the Last Glacial Maximum and insights into the global marine carbon cycle. Climate of the Past.*

We will add a figure of the sea ice anomaly ($GLcomb$-$Ctrl$) to the supplement. The separation of the effects of sea ice cover expansion from the other physical changes would be highly interesting to isolate. However, we find them to be beyond the scope of this paper, and we wish to keep this paper focused on the effects of flexible C/P on $pCO_2$. If PJB is interested in the separation of physical effects in cGENIE, this is explored in Ödalen et al. (2018). There, we want to point specifically to the simulation $AD/2$ in Figure 2, where panel e) details the contributions from the changes in each of the different carbon capture processes to the net change in $pCO_2$. The simulation $AD/2$ has a reduced biological pump compared to the pre–industrial control, but an enhanced carbon capture due to increased disequilibrium and saturation carbon. This is attributed mainly to ocean cooling and a resulting expansion of sea ice. However, no further separation of the effect of sea ice was made.

- *Page 13, lines 10–23 : This paragraph would benefit from being clearer in its findings of $CO_2$ sequestration regarding C:P ratio changes. I have to read this multiple times to understand what the authors are trying to say when comparing Redfield, variable C:P and C:P=121.*

This paragraph will be clarified by concentrating all the results and discussion concerning C/P = 121 to Section 4.2, as suggested by Referee #1. We are confident that this paragraph will become clearer when it is rewritten to focus only on the difference in $CO_2$ sequestration between Redfield ($RED$) and flexible C/P ($GAM$).

2.5  Discussion

- ***Page 13, line 28 : your reference to "model" should be an "empirical model"
  to avoid confusion with the Earth System Model, GCM, etc.***
  Agreed. We will add "empirical".

- ***Page 14, lines 1–5 : The advantages of using empirical/statistical models
  within biogeochemical ocean GCMs, including the Galbraith & Martiny (2015) parameterisation, was explored rigorously in my 2018 paper in
  Global Biogeochemical Cycles. It not only improved that model's biogeochemistry significantly, but also altered the long–term behaviour of the carbon cycle as you have also found. It may be interesting, but I of course
  leave it up to you whether it's useful. Buchanan et al (2018) The importance
  of dynamic biological functioning for simulating and stabilizing ocean biogeochemistry. Global Biogeochemical Cycles.***
  We thank PJB for pointing us to this paper, and find the conclusions very interesting. They are well aligned with the study we perform in this paper, and by citing
  this paper, we will strengthen our arguments. We will thus include the suggested
  paper in our discussion.

- ***Page 14, line 32 : the conclusions of Odalen et al (2018)... which were?
  What did Odalen et al (2018) do?***
  Here, lines 30–33 should read: *Note that $Ctrl_{GAM}$ has a larger inventory of
  DIC, as well as $C_{soft}$, compared to $Ctrl_{RED}$. Ödalen et al. (2018) found that
  drawdown of $CO_2$ in response to a perturbation is larger when the control state
  has a smaller inventory of DIC and $C_{soft}$. Yet, the effect of applying the same
  perturbation results in a larger drawdown of $CO_2$ in $GAM$ than in $RED$. This is
  thus opposite of the conclusions of Ödalen et al. (2018). The reason is that the
  flexible stoichiometry in effect increases the drawdown potential, which more than
  compensates for the increased carbon inventory in the control state.* As stated

above, Section 4.2 will be re–written, and clarified. In this way, we will assure that the conclusions of Ödalen et al. (2018) are clearly stated.

- ***Page 15, lines 19–24 : So the proportion of remineralised to preformed phosphorus effectively doesn't change in the simulations? And yet, you find a large increase in respired C? This must mean that the remineralised phosphorus that is exported into the ocean interior in your $GLcomb$ simulation is being quickly circulated into the lower overturning cell and returned to the Antarctic Zone, where sea ice prevents gas exchange and biological production, at which point this P is recirculated and becomes preformed, while respired C remains respired and is also recirculated. If this is the case, it merits more discussion in comparison with previous literature on the subject of a more efficient biological pump that invokes more regenerated nutrients as a must for a more efficient biological pump. I suggest Hain, Sigman & Haug (2014) The biological pump in the past. Treatise on Geochemistry, 2nd Ed.***
The referee has correctly identified that, despite the fact there is no change in remineralised P ($P_{rem}$) in $GLcomb_{GAM}$ compared to $Ctrl_{GAM}$, there is an increase in remineralised C. However, the process described by the referee focuses on what happens after remineralisation, while we suggest that this decoupling happens before the organic material is exported to the deep ocean (see lines 25–31). The forcing components applied to $GLcomb_{GAM}$ have competing effects on the amount of organic matter that remineralises in the deep ocean. The net effect is that this amount does not change globally (reflected by a constant $P_{rem}$ compared to $Ctrl_{GAM}$). Meanwhile, changes in ocean circulation, remineralisation depth and dust deposition still cause the local nutrient availability in the surface waters to change. This affects the elemental composition of the exported organic material. In $Ctrl_{GAM}$, the average elemental C/P composition is 121/1. In $GLcomb_{GAM}$, this average is 134/1. This means that even though the

same amount of P is exported to the deep ocean, the organic molecules carry more carbon, which is released in the deep ocean during remineralisation. In $Ctrl_{GAM}$, the global average concentration of $P_{rem}$ is 1.16 $\mu mol kg^{-1}$ (c.f. 1.17 $\mu mol kg^{-1}$ in $GLcomb_{GAM}$). By increasing the average C/P composition of 1.16 $\mu mol kg^{-1}$ organic molecules from 121 to 134 (i.e. by 13 units), this causes an increase in $C_{rem}$ by $\sim 15$ $\mu mol kg^{-1}$, which corresponds to the observed increase in $C_{rem}$. In summary, we suggest this is a result of changes in surface P fields (see Fig. 7), rather than a change in the partitioning between $C_{rem}$ and $P_{pre}$ in the recirculation area in the Antarctic Zone. We will clarify this part of the discussion, which comprises lines 19–31 on page 15.

- **Page 16, lines 1–14 :**
  Here, each of the questions will be treated separately.

  - *I would like to see how variations in P:$O_2$ requirements were treated in this model.*
    C:$O_2$ requirements were meant to be held constant throughout the simulations, consequently causing changes in P:$O_2$. However, an inconsistency in the code caused C:$O_2$ requirements to change in the simulations of the 121–ensemble, causing an inconsistent behaviour of P:$O_2$. This has been corrected, as outlined above.

  - *Also, can you please explain why the deep water formation characteristics of a model affects $O_2$?*
    Deep water formation characteristics of a model affects the amount of time available for remineralisation and, consequently, the oxygen consumption. In addition, due to a lack of resolution deep water formation in climate models generally happens as open water convection, rather than as dense plumes along slopes. This causes too much oxygen to be entrained into the deep ocean. We will add this explanation in the updated manuscript.

– ***Overall, I find this section a bit sparse and I'm not entirely sure what the point of it is.***
The section aims to discuss 1) to what extent our $GLcomb$ simulations reproduce proxy observations, in this case for $O_2$, and 2) to discuss one of the problems that arose from applying the flexible C/P in GENIE, i.e. that $O_2$ concentrations in $Ctrl_{GAM}$ are too low, and its implications for the glacial–like simulation. We agree that the section could be expanded, and we will add the information requested by PJB to make the discussion more comprehensive.

– ***Can you comment on the size of the OMZs? New evidence shows that the OMZs in the Pacific expanded vertically during the glacial. Hoogakker et al (2018) Glacial expansion of oxygen–depleted seawater in the eastern tropical Pacific. Nature***
In the Atlantic, $Ctrl_{GAM}$ (Fig. 4 e) reproduces the observed extent of the OMZ (Fig. 4 a) better than $Ctrl_{RED}$ (Fig. 4 c) does. In the Pacific Ocean, the $O_2$ gradient in the observations (Fig. 4 b) is more gradual compared to that of the control states (Fig. d, f), but the core of the OMZ is well reproduced by the model. The forcings applied to $GLcomb$ are not sufficient to reproduce a full glacial state (see also author's response to the next main point, regarding Page 16, line 22). Still, we do get a vertical expansion of the OMZ in $GLcomb_{RED}$ (Fig. 4 h) compared to $Ctrl_{RED}$ (Fig. 4 c), in agreement with the findings of Hoogakker et al (2018). In $GLcomb_{GAM}$, oxygen depletion is too extensive, but the tendency of vertical expansion compared to the control state is present here as well.

• ***Page 16, line 22 : And yet others achieve relatively good correlations in the Pacific basin using the same data? Menviel et al (2017) Poorly ventilated deep ocean at the Last Glacial Maximum inferred from carbon isotopes: A data–model comparison study. Paleoceanography. Muglia et al (2018)***

*Weak overturning circulation and high Southern Ocean nutrient utilization maximized glacial ocean carbon. Earth and Planetary Science Letters. I think you cannot say that your model doesn't provide good fits to the LGM proxy data because of poor data coverage, and it would be more useful to discuss why the model doesn't fit with the data. It seems like your glacial circulation in the Pacific basin is therefore not accurate?*

The forcings applied to $GLcomb$ are factors that are likely to be important for the glacial ocean circulation and biogeochemistry. However, these forcings are not sufficient to reproduce a full glacial state (i.e. the use of the term "glacial–like simulations", rather than "LGM simulation"). Other forcings that have shown to be important for modelling of glacial $\delta^{13}C$ are, for example, brine rejection (Bouttes et al., 2010; Bouttes et al., 2011), and freshwater forcing (e.g., Schmittner et al., 2002; Hewitt et al., 2006; Bouttes et al., 2012). The fact that some important forcings are missing (mentioned on lines 27–28) is likely the main cause for the model–data discrepancy, and the reason for why we do not achieve an accurate glacial Pacific Ocean circulation. This will be clarified in the section. We will also clarify, throughout the paper, the fact that we do not aim to produce a full LGM state. This may also call for adjusting the title of the paper.

- *Page 16, line 26 : Temperature is not chemical. I also do not understand how you could alter the temperature and salinity of the ocean without altering water mass distributions, and if this is indeed the case, it requires further description as to why earlier in the paper. Also a good spot to talk about why the data in the Pacific are not well reproduced by the model.*
  We agree that temperature in itself is not chemical. Here, we were referring to the changes in solubility of $CO_2$, which is a chemical response to changes in temperature. This will be clarified in the revised version of the paper. The water mass distribution in cGENIE is strongly constrained by the resolution of the model, especially in the vertical. Changes in temperature and salinity that should cause

changes in water mass volume may not be sufficient to allow a water mass to extend to the next vertical level of the model. As a consequence, while the gradient between water masses may become more or less pronounced, the interface of water masses may still remain at the same depth. The section will be clarified, including the above description of why Pacific glacial circulation is not fully reproduced.

- **Page 16, lines 24–31 : I don't follow this paragraph. You state that "Each of the two observational datasets (HOL and LGM) display similar correlations across the two model simulations. This implies that our changes in forcings do not achieve any obvious changes in water mass distribution." But doesn't the distribution of $\delta^{13}C$ change across the glaciation and into the Holocene? $\delta^{13}C$ in the Atlantic, for instance, is often used to show that the Atlantic meridional overturning was shallower during the glacial, and that this change occurred during Marine Isotope Stage 4 (Oliver et al (2010) A synthesis of marine sediment core $\delta^{13}C$ data over the last 150000 years. Climate of the Past.)? Moreover, $\delta^{13}C$ is used as a way to show that the water mass distribution between the Atlantic and Pacific was considerably different during the glacial as compared to the Holocene (Sikes et al (2017) Enhanced $\delta^{13}C$ and $\delta^{18}O$ Differences Between the South Atlantic and South Pacific During the Last Glaciation: The Deep Gateway Hypothesis. Paleoceanography.) These studies conflict with what you are saying.**

The proxy data do imply a change in $\delta^{13}C$ across the deglaciation (whole ocean change 0.34 $\pm 0.19$ ‰, Peterson et al., 2014). What we are trying to say is that our model simulations do not fully reproduce this change. Here we are referring to the fact that the correlation of the HOL proxy records with $Ctrl_{RED}$, $Ctrl_{GAM}$, $GLcomb_{RED}$, and $GLcomb_{GAM}$, is in all cases between 0.76–0.78. On the other hand, the correlation of LGM proxy records with the same four simulations is in all cases between 0.55–0.58. As our $GLcomb$–simulations still correlate so well

with the HOL dataset, this suggests the applied forcings have not caused these simulations to be clearly different from $Ctrl$ in terms of water mass distribution. For the same reason, the correlation with LGM proxy data does not significantly improve from $Ctrl$ to $GLcomb$. Thus, the deglacial change in $\delta^{13}C$ reflected in the proxy data is not fully captured by the model. This will be clarified in the updated version of the manuscript.

- *(Section 4.5 ?): The ability for simulated $\delta^{13}C$ to reproduce the proxy data at the LGM will depend strongly on water mass distribution (which apparently doesn't change appreciably) and how biological fractionation is parameterised. If it is constant, the 10% loss in C fixation will cause the ocean to be more positive overall by some constant factor. However, if the parameterisation contains a dependence on aqueous $CO_2$ and growth rate, both of which are lower, then the fractionation will vary. It would be worthwhile telling the reader what parameterisation is used and, if it does involve growth rate and aqueous $CO_2$, what effect this has.*
The fractionation is dependent on both aqueous $CO_2$ and growth rate (represented in $K_Q$, see appendix). This dependence and its consequences will be detailed by adding the following text in an Appendix (see Section 2.6)

2.6   Appendix: $\delta^{13}C$ in cGENIE

cGENIE represents $^{13}C$ as an explicit tracer (separate from and in addition to bulk carbon) in the model, tracking its concentration in all the same gaseous, dissolved, and solid forms that carbon exists in, reporting $\delta^{13}C$ in ‰ relative to the standard VPDB. The current scheme is based on that described in Ridgwell (2001) and updated as described in Ridgwell et al. (2007), and is evaluated for the modern ocean (alongside simulated $\Delta^{14}C$) in cGENIE, in Kirtland–Turner and Ridgwell (2016).

In the aqueous phase, the isotopic partitioning of carbon between $CO_2(aq)$, $HCO_3^-$, and $CO_3^{2-}$ is resolved and follows Zeebe and Wolf–Gladrow (2001) (their Section 3.2). The empirical fractionation factors used are from Zhang et al. (1995). The air–sea fractionation scheme follows that of Marchal et al. (1998) with the individual fractionation factors again taken from Zhang et al. (1995).

For the isotopic composition of organic carbon ($\delta^{13}C_{POC}$), the model of Rau et al. (1996) is adapted, assuming that the isotopic signature of exported POC reflects that of phytoplankton biomass. Following Ridgwell (2001), the full equation of Rau et al. (1996) is simplified to:

$$\delta^{13}C^{POC} = \delta^{13}C^{CO_{2(aq)}} - \epsilon_f + (\epsilon_f - \epsilon_d) \cdot \frac{K_Q}{[CO_{2(aq)}]} \qquad (1)$$

where $[CO_2(aq)]$ is the ambient concentration of aqueous $CO_2$ and $\delta^{13}C_{aq}$ is its isotopic composition. $K_Q$ is a temperature–only dependent approximation of the full cell–dependent size and growth rate parameterization in the Rau et al. (1996) model (see Ridgwell, 2001). We take an intermediate value for the enzymatic isotope fractionation factor associated with intracellular C fixation ($\epsilon_f$) of -25‰ following Rau et al. (1996,1997), and assume a temperature–invariant value for $\epsilon_d$ of 0.7 ‰.

The result of applying this scheme in cGENIE, is a zonal mean profile characterized by $\delta^{13}C_{POC}$ of -22 to -21 ‰ in the tropics, declining with increasing latitude to reach -28 to -30 ‰ in the Southern Ocean. This latitudinal pattern is comparable to measurements made on suspended particulate organic matter as discussed in Ridgwell (2001).

For $^{13}C$ fractionation into biogenic carbonates at the ocean surface (e.g. foraminiferal tests, and coccolithophorid coccoliths), cGENIE follows Mook (1986) and employs a simple temperature–dependent fractionation between the $\delta^{13}C$ of aqueous $HCO_3^-$ and calcite.

**3  Technical corrections**

- ***Page 1, line 13 : remove repeated "with"***
  We will correct this typo.

- ***Page 5, line 16 : replace "reduced half" with "halved"***
  The sentence will be changed according to the suggestion.

- ***Table 1 : "witg" to "with"***
  We will correct this typo in the caption of Table 1.

- ***Page 11, line 8 : Inadvisable to begin a sentence with "∼"***
  The sentence has been rewritten (see above comment for Page 11, line 9).

- ***Page 11, line 29 : "SV" to "Sv"***
  We will change "SV" to "Sv" throughout the manuscript.

- ***Page 13, line 26 : "GCMs" this acronym has not been defined previously.***
  We will define the acronym here.

- ***Page 15, line 23 : I assume you mean "0.003" rather than "0003"?***
  Yes, we will correct this to 0.003.

[Figure]

**4 References**

Bouttes, N., Paillard, D., and Roche, D.: Impact of brine–induced stratification on the glacial carbon cycle, Climate of the Past, 6, 575–589, 2010.

Bouttes, N., Paillard, D., Roche, D. M., Brovkin, V., and Bopp, L.: Last Glacial Maximum $CO_2$ and $\delta^1 3C$ successfully reconciled, Geophysical Research Letters, 38, 2011.

Bouttes, N., Roche, D. M., and Paillard, D.: Systematic study of the impact of fresh water fluxes on the glacial carbon cycle, Climate of the Past, 8, 589–607, https://doi.org/10.5194/cp-8-589-2012, https://www.clim-past.net/8/589/2012/, 2012.

Hewitt, C. D., Broccoli, A., Crucifix, M., Gregory, J., Mitchell, J., and Stouffer, R.: The effect of a large freshwater perturbation on the glacial North Atlantic Ocean using a coupled general circulation model, Journal of Climate, 19, 4436–4447, 2006.

Marchal, O., Stocker, T. F., and Joos, F.: A latitude-depth, circulation-biogeochemical ocean model for paleoclimate studies. Development and sensitivities, Tellus B: Chemical and Physical Meteorology, 50, 290–316, 1998.

Matsumoto, K.: Biology-mediated temperature control on atmospheric pCO2 and ocean biogeochemistry, Geophysical Research Letters, 34, 2007.

Mook, W.: 13C in atmospheric CO2, Netherlands Journal of Sea Research, 20, 211–223, 1986.

Ödalen, M., Nycander, J., Oliver, K. I. C., Brodeau, L., and Ridgwell, A.: The influence of the ocean circulation state on ocean carbon storage and $CO_2$ drawdown potential in an Earth system model, Biogeosciences, 15, 1367–1393, https://doi.org/10.5194/bg-15-1367- 2018, https://www.biogeosciences.net/15/1367/2018/, 2018.

Peterson, C. D., Lisiecki, L. E., and Stern, J. V.: Deglacial whole-ocean $\delta^1 3C$ change estimated from 480 benthic foraminiferal records, Paleoceanography, 29, 549–563,

2014.

Rau, G., Riebesell, U., and Wolf–Gladrow, D.: $CO_{2aq}$–dependent photosynthetic $^{13}C$ fractionation in the ocean: A model versus measurements, Global Biogeochemical Cycles, 11, 267–278, 1997.

Rau, G. H., Riebesell, U., and Wolf–Gladrow, D.: A model of photosynthetic $^{13}C$ fractionation by marine phytoplankton based on diffusive molecular $CO_2$ uptake, Marine Ecology Progress Series, 133, 275–285, 1996.

Ridgwell, A., Hargreaves, J. C., Edwards, N. R., Annan, J. D., Lenton, T. M., Marsh, R., Yool, A., and Watson, A.: Marine geochemical data assimilation in an efficient Earth System Model of global biogeochemical cycling, Biogeosciences, 4, 87–104, 2007.

Ridgwell, A. J.: Glacial-interglacial perturbations in the global carbon cycle., Ph.D. thesis, University of East Anglia, 2001.

Schmittner, A., Meissner, K., Eby, M., and Weaver, A.: Forcing of the deep ocean circulation in simulations of the Last Glacial Maximum, Paleoceanography, 17, 5–1, 2002.

Turner, S. K. and Ridgwell, A.: Development of a novel empirical framework for interpreting geological carbon isotope excursions, with implications for the rate of carbon injection across the PETM, Earth and Planetary Science Letters, 435, 1–13, 2016.

Zeebe, R. E. and Wolf–Gladrow, D. A.: $CO_2$ in seawater: equilibrium, kinetics, isotopes, 65, Gulf Professional Publishing, 2001.

Zhang, J., Quay, P., and Wilbur, D.: Carbon isotope fractionation during gas-water exchange and dissolution of CO2, Geochimica et Cosmochimica Acta, 59, 107–114, 1995.

**5  Figures**

**5.1  Fig. 5 - updated caption**

Model ocean $\delta^{13}C$ (contours) compared to the two proxy record time slices (HOL and LGM) of benthic $\delta^{13}C$ (circles) of Peterson et al. (2014). The upper half of the figure shows the Atlantic Ocean (panels a–d), while the lower half shows the Pacific Ocean (panels e–h). The columns represent the model simulations ($Ctrl_{RED}$ or $Ctrl_{GAM}$), while each row represents one of the proxy record time slices (HOL or LGM). The left hand column shows $Ctrl_{RED}$ (panels a, c, e, g), and the right hand column shows $GLcomb_{RED}$ (panels b, d, f, h). The rows show, from top to bottom, a–b) HOL Atlantic, c–d) LGM Atlantic, e–f) HOL Pacific, g–h) LGM Pacific. Note that, before we compare $GLcomb_{RED}$ to LGM observations (panels d and h), a constant of 0.32‰ is subtracted from the simulated $\delta^{13}C$, to account for terrestrial release of $\delta^{13}C$–depleted terrestrial carbon which is not modelled. The corresponding comparison for model version $GAM$ is shown in Fig. S.3.
* * *
[Figure]

**Fig. 1.** Updated version of Fig. 5. The caption of Fig. 5 is updated according to the text on the previous page.

---

## Author Response (AR1)

*Author's response to Anonymous Referee #1 for*
**Variable C/P composition of organic production and its effect on ocean carbon storage in glacial model simulations**

Malin Ödalen[1], Jonas Nycander[1], Andy Ridgwell[2,3], Kevin I. C. Oliver[4], Carlye D. Peterson[2], and Johan Nilsson[1]

[1]Department of Meteorology, Bolin Centre for Climate Research, Stockholm University, 106 91 Stockholm, Sweden
[2]Department of Earth Sciences, University of California–Riverside, Riverside, CA 92521, USA
[3]School of Geographical Sciences, Bristol University, Bristol BS8 1SS, UK
[4]National Oceanography Centre, Southampton, University of Southampton, Southampton SO14 3ZH, United Kingdom

**Correspondence:** Malin Ödalen (malin.odalen@misu.su.se)

**1 Introduction**

In this document, referee comments are shown in ***bold and black italics***, and author's discussion response directly below in plain black. For each author's response, the corresponding changes in the manuscript are shown in blue.

**2 General comments**

5    – ***Introduction: It might be good to add a sentence detailing the evidence for lower terrestrial carbon storage during glacial times (p2, L.1).***

We agree and will add to the introduction: *Studies of paleoproxy records indicate that carbon storage in the glacial terrestrial biosphere was smaller compared to in interglacial climate* (Shackleton, 1977; Duplessy et al., 1988; Curry et al., 1988; Crowley, 1995; Adams and Faure, 1998; Ciais et al., 2012; Peterson et al., 2014).

10    Added to p. 2, L. 2-5: *In addition, studies of paleoproxy records indicate that carbon storage in the glacial terrestrial biosphere was smaller compared to in interglacial climate* (Shackleton, 1977; Duplessy et al., 1988; Curry et al., 1988; Crowley, 1995; Adams and Faure, 1998; Ciais et al., 2012; Peterson et al., 2014).

   – ***Section 2.3.1: It would be good to precise whether the wind changes impact the air–sea gas exchange of $CO_2$.***

Yes. In cGENIE, gas transfer velocities are calculated as a function of wind speed (described in Ridgwell et al. (2007), and following Wanninkhof (1992)). We will add this information to Section 2.3.1.

15    Added to p. 6, L. 24-26: *In cGENIE, gas transfer velocities are calculated as a function of wind speed (described in Ridgwell et al. (2007)), and following Wanninkhof (1992). Consequently, weaker winds also lead to reduced gas exchange with the atmosphere.*

– *Methods and section 3.2.1: If I understand correctly global salinity is not increased during glacial times. If correct, it might be good to clearly state it as well as its impact on solubility changes.*

This is correct. We do not aspire to simulate a full glacial state, but rather to explore the effect of flexible C/P for biological carbon capture in response to a few common glacial forcings. As we do not change salinity, we are likely to overestimate the increase in solubility between $Ctrl$ and $GL_{comb}$, by $\sim 6$ ppm (Kohfeld and Ridgwell, 2009). This effect is consistent for any choice of C/P parametrisation, and is therefore not explored further. We will add this information to Section 3.2.1.

Added to p. 11, L. 16-18: *As we do not change salinity, we are simultaneously likely to overestimate the increase in solubility between Ctrl and a glacial–like state, by $\sim 6$ ppm (Kohfeld and Ridgwell, 2009). This effect is consistent for any choice of C/P parametrisation, and is therefore not explored further.*

– *Section 3.2.3: p. 10, L. 29: Please quantify magnitude and direction of "small".*

We will clarify the sentence by changing it to: *Increases in RLS cause very small, but global, decreases in surface $PO_4$ concentrations (global average anomaly = -0.016 $\mu M$) [...]*

Sentence, now on p. 12, L. 8-9, has been changed to: *Our changes in RLS cause very small, but global, changes in $PO_4$ concentrations (global average anomaly = -0.016 $\mu M$, RLS $\times$ 1.25 in Fig. 7 c), [...].*

– *Section 3.2.3: p. 11, L. 3–4: This is an interesting result that should be emphasized.*

We add here: *The potential implications of this result for warm climate scenarios is further discussed in Section 4.1.*

The suggested change was added to Section 3.2.3, now on p. 12, L. 14-15.

In Section 4.1, we add: *Our sensitivity experiments $RLS \times 0.75$ and $RLS \times 1.25$, reveal that the response in $pCO_2^{atm}$ to the perturbation is enhanced in $GAM$ compared to $RED$ for both increased and decreased RLS. While increased $RLS$ would be an effect of ocean cooling, and thus of interest for glacial studies, reduced RLS would be a consequence of ocean warming (Matsumoto, 2007). Matsumoto (2007) describes how decreased RLS would have a positive feedback on $pCO_2^{atm}$ in future warming climate. Our results imply that flexible C/P could have a further re–inforcing effect on this feedback. It would therefore be of interest to apply a parametrisation of flexible C/P in models used for simulations of future climate feedbacks.* The suggested text was added to Section 4.1, p. 16, L. 4-9.

We also add the following sentence to Conclusions: *Flexible C/P also has the potential to be an additional positive feedback of ocean warming on $pCO_2^{atm}$ in future climate.* The suggested text was added on p. 21, L. 5.

– *Section 3.2.3: p. 10, L. 31–32: This sentence is unclear.*

We suggest clarifying this sentence by changing it to: *Experiments with deeper RLS in 121 ($RLS \times 1.25_{121}$ and $RLS \times 1.75_{121}$), suggest that about 40 % of the observed differences in $pCO_2^{atm}$ between $GAM$ and $RED$ can again be attributed to the difference in export flux average C/P.*

As suggested by both referees, all information about experiments with the model version 121 has been moved to Section 4.4 (p. 18). Here, the suggested sentence proved to be redundant, and the information is instead conveyed in the sentence

*'[...] the simulations with 121 indicate that, depending on the change in forcing, between 1/3 and 2/3 of the difference in drawdown between RED and GAM is due to the difference in average C/P between the control states (Fig. 6, Table S.2).'*, found on p. 18, L. 31-33.

– **Section 3.2.4: p. 11, L. 11: Shouldn't iron fertilization lead to an increase in $P_{rem}$ (instead of $P^*$)?**

As $\overline{P^*} = \overline{P_{rem}/P_{tot}}$, and $P_{tot}$ is constant, an increase in $P_{rem}$ is equal to an increase in $P^*$. We will clarify the sentence by changing it to: *[...] the iron added by the dust forcing allows more efficient usage of P in the HNLC-regions, which increases the ocean storage of biologically sourced carbon $P_{rem}$ (thus, $P^*$ increases).*

The suggested change has been made, and is now found on p. 12, L. 22-23.

– **Section 3.2.4: p. 11, L. 16: It is unclear what you mean here with "radionuclide proxy data"**

We will clarify the sentence by changing it to: *This subantarctic increase in biological efficiency is consistent with radionuclide proxy data ($^{10}Be$, $^{230}Th$, $^{231}Pa$) from the LGM [...].*

The suggested change has been made, and is now found on p. 12, L. 27-28.

– **Section 4.2: - Please consider amending the title of that section - I would suggest to add all the results of experiment 121 here and thus all the finishing sentences of the diverse paragraph (ex: p. 10, L. 31–32).**

We agree that grouping all the results of experiment 121 and the associated discussion in Section 4.2 is a good idea, and will follow this recommendation. We will thereby also change the title of the subsection to *Effect of modified but fixed C/P*.

As suggested, all results from experiments with model version 121 have been grouped into Section 4.4, titled *Effect of modified but fixed C/P*.

– **Figure 5: I'm confused as to what is shown here. I think mistakes have been made in the plots or legends as it does not make any sense. How can both HOL and LGM can be shown for CTR? How can both HOL and LGM can be shown for GLcomb? It is really not obvious LGM Pacific is HOL Pac -0.32 permil (h compared to f). Similarly, how do you go from Hol Pac to LGM Pac in CTR (g compared to e)? g looks much more like an Atlantic section than a Pacific one. How can d be LGM Atl and f) Hol Pac? d) might be Pacific.**

In this figure, the sub–panels have by mistake been shifted to the wrong positions, which naturally causes unnecessary confusion. We apologise for the mistake and show in Fig. 1 the corrected version of the figure, which will replace Fig. 5 in the revised manuscript. The referee is also confused by how both time slices HOL and LGM can be shown for the $Ctrl$-simulation. This is simply because we have chosen to compare each of the simulations ($Ctrl$ and $GLcomb$) with both time slices (HOL and LGM) of the proxy data. Even though we expect, of course, that $Ctrl$ should to a higher extent reproduce the patterns we see in the HOL data than in the LGM data, we do not want to assume that the model is successful in this respect. For transparency in the process, we therefore show both comparisons. In order to present this figure in a more accessible way, we will clarify that the columns represent the two model simulations, and that the rows represent the proxy records to which we compare the simulations (see caption of Fig. 1). Note that the update of

the caption will also apply to Fig. S3. Also, it seems the referee may have misunderstood our subtraction of -0.32‰, though the confusion may be a result of the panels being organised in the wrong order. The contours in panel f shows the Pacific Ocean of the $GLcomb$ simulation compared to HOL Pacific proxy records (circles). The contours in panel h also shows the Pacific Ocean of the $GLcomb$ simulation, but here, 0.32 ‰has been subtracted from the simulation data. In panel h, the circles show LGM Pacific proxy records. We describe in Section 2.4 that proxy records of $\delta^{13}C$ indicate that the LGM ocean was more depleted in $\delta^{13}C$ than the Holocene ocean. We will clarify that this is also true for the dataset we use here (see Peterson et al., 2014). Gebbie et al. (2015) estimated this difference in whole–ocean $\delta^{13}C$ to -0.32±0.20‰. The low LGM whole–ocean value is attributed to glacial contraction of the terrestrial biosphere, and an associated addition of $\delta^{13}C$–depleted carbon of terrestrial origin to the ocean. As we do not simulate this terrestrial contribution of $\delta^{13}C$–depleted carbon, we do not expect our $GLcomb$–simulations to reproduce this change in whole–ocean $\delta^{13}C$. We therefore subtract 0.32‰from each point of the $GLcomb$ simulation output, before we compare to the LGM proxy records.

The figure and caption have been corrected and updated in accordance with the author's response, and have been added to the manuscript. Note that the clarifications also apply to Fig. S.3.

– *Figure 9: With a fixed Redfield ratio $AC_{rem}$ should increase with $P_{rem}$. I am confused as to why $AC_{rem}$ increases with $P^*$ here.*

As stated above, $\overline{P^*} = \overline{P_{rem}}/\overline{P_{tot}}$, thus an increase in $P_{rem}$ is analogous to an increase in $P^*$.

As an increase in $\overline{P^*}$ is analogous to an increase in $P_{rem}$, no changes have been made to the manuscript in response to this comment.

– *Table 2: I find the format of this table not ideal and wonder if it would make sense to split the RED and GAM results. Also, it might not be necessary to show the AMOC strength for both experiments. The AABW transport in the Atlantic is extremely low (where did you take it?).*

We agree, and will re–organise the table to show the GAM results below RED results, rather than showing them side by side. However, if the results are presented on different rows, the table would look incomplete if we leave out the AMOC strength for one of the model versions. We would therefore prefer to keep those results in the table. $\psi_m in$ in this table is simply the minimum of the Atlantic overturning streamfunction below below 556m depth and north of 30°N. As seen in Fig. 2 c–d, the AABW circulation is weak in the Atlantic north of -30°N, while its peak strength is located in the Southern and Pacific Oceans. Due to the lack of a boundary between the Atlantic and the Pacific south of -30°N, it is not possible to compute the basin–specific streamfunction further south, though the Atlantic AABW circulation is likely to be stronger there.

Table 2 has been updated according to the suggestions in the author's response. Note that a clarification has been made in the caption, to specify that the maximum and minimum Atlantic streamfunction was taken north of -30°N. In the published author's response, there was a typo which suggested that this latitude was +30°N, but naturally, the cutoff latitude is that corresponding to the southernmost tip of Africa. We apologise for this typo.

**3   Minor points and typos**

– *Section 2.3.3, p. 6, L. 17: Missing table number*

Here, *Table ??* should be corrected to *Table 2*.

Missing table number corrected to Table 2, see p. 7, L. 28.

– *Section 2.4, p. 6, L. 22:*

Typo, *retreived* should be corrected to *retrieved*.

Typo corrected, see p. 7, L. 32.

– *Section 2.5, p. 7, L. 13:*

Missing reference marked by *(?)* should be (Ödalen et al., 2018).

Missing reference corrected to (Ödalen et al., 2018), see p. 8, L. 23.

– *Section 3.1.1, p. 8: Sverdrup is usually noted "Sv".*

*SV* will be changed to *Sv* throughout the paper.

Correction has been made throughout the paper.

– *Section 3.2.4, p. 11, L. 10: remove one "the".*

Typo, will be corrected.

Sentence removed when information was moved to Section 4.4.

**References**

[revised manuscript text omitted]

**Correspondence:** Malin Ödalen (malin.odalen@misu.su.se)

**1 Introduction**

In this document, referee comments are shown in ***bold and black italics***, and author's discussion response directly below in plain black. For each author's response, the corresponding changes in the manuscript are shown in blue.

**2 General comments**

5 ***The manuscript is well organised in how it presents the methods and results. In this sense, it is very easy as a reader to understand what the authors have done. However, the writing itself requires a lot of polishing. The common usage of "e.g.", as an example (pun not intended), in mid-sentence makes things simultaneously difficult to read and also leaves me thinking that the authors are purposely choosing to not discuss all the evidence/processes/knowledge on a subject. I strongly suggest that the authors take some time to improve the writing.***

10 As suggested by PJB, we will work to make the writing clearer. Specifically, PJB points out the common usage of e.g. mid–sentence, and questions whether this is intentionally used *"to not discuss all the evidence/processes/knowledge on a subject"*. This was not our intention, but rather an attempt to avoid the text getting too long. This is why we decided in some cases to only bring up the most relevant examples. We will go through these parts of the text in detail, and add more information.
We have worked through the manuscript to improve the writing, especially in the Introduction and Discussion sections. We

15 have re-written sentences where e.g. was used mid-sentence.

***One particularly important part of the manuscript that is not conveyed clearly is their diagnosis of $CO_2$ capture via variable C:P ratios, as opposed to a simple whole ocean increase in C:P. For instance, in the conclusion the authors state that "About half of the increased drawdown of $CO_2$ results from different global average C/P in the export production. In addition, flexible stoichiometry allows increased carbon capture through the biological pump, while maintaining or even***

20 ***decreasing the fraction of remineralised to total nutrients in the deep ocean.". The reader is therefore left confused about***

*what C:P ratios are actually doing. Aren't they contributing to the whole of the increased drawdown? I think what the authors are trying to say is that the spatial and temporal variations in C:P are important for strengthening the biological pump, because they react dynamically to changes in nutrient supply to reinforce further $CO_2$ drawdown over a simple prescribed whole ocean increase. I would therefore advocate for the authors to make this more clear.*

Both Referee #1 and PJB find the discussion of Section 4.2 (Implications of changed average C/P) unclear, and both give useful suggestions on how to clarify this section. The section will thus be re–written based on these suggestions.

In the updated manuscript, Sections 4.2 (now 4.4), and 4.3 (now 4.2) have been re-written, and we have worked to clarify the contributions by $CO_2$ capture via variable C:P ratios, as opposed to a simple whole ocean increase in C:P. For a reader who has read the updated manuscript, the quoted sentence from the Discussion should now be easier to understand, and it was therefore left unchanged.

*I also think that many of the concepts discussed in this paper with relevance for the sensitivity experiments could be introduced better in the Introduction. This includes previous glacial modelling studies, paloeproxy evidence and the theories that have been consequently generated for increased oceanic carbon storage due to these prior studies. Temp-dependent remin rates, changes in wind stresses, overturning circulation, polar stratification, solubility (salt and heat), Si-leakage, $CaCO_3$ compensation and production rates, Fe fertilisation, increased $N_2$ fixation, sea ice expansion, even volcanism. All of these, except volcanism, are feedbacks that were somehow kick-started by changes in solar insolation. While you obviously do not need to discuss all of these in detail, laying out the current "pieces of the puzzle" would help to ground your work in the current stream of consciousness in the paleoclimate community*

PJB details how the Introduction could be extended with a more extensive discussion of concepts, in order to help the reader. We agree that this could be helpful, and we will make additions according to PJB's suggestions.

The Introduction has been extended with more detailed information about evidence of contributers to glacial oceanic carbon storage from previous studies (page 2, lines 7-30), and more detailed information about the background of the concepts behind the sensitivity experiments (page 3, lines 11-20).

**3   Specific comments**

**3.1   Abstract**

– *Page 1, line 8: surely you mean increases rather than decreases? And also you mean Phosphate. Because an increase in Fe deposition, which is a nutrient, has been linked to an increase in C:P ratios. Garcia et al (2018): Nutrient supply controls particulate elemental concentrations in the low latitude eastern Indian Ocean. Nature Communications.*

Yes, "decreases" should be corrected to "increases", and "nutrients" to "phosphorus" (as we are discussing elemental ratios).

Page 1, line 8-9: After re–reading the sentence and PJB's comment, we agree with PJB, and the sentence has been changed to *'If the C/P ratio increases when phosphate availability is scarce, [...]'*, in accordance with his comment.

**3.2 Introduction**

– *Page 2, line 2 : Also because of the rapid release of carbon to the atmosphere over the deglaciation, implying its storage somewhere during the glacial.*

We agree and will add this information, along with relevant references.

On page 2, lines 5-6, we have added: *'During deglaciation, radiocarbon evidence indicate that $CO'_2$ was rapidly released from the ocean back to the atmosphere (Marchitto et al., 2007; Skinner et al., 2010).'*

– *Page 2, line 5 : What is the e.g. here referring to?*

This is a formatting issue. The *e.g.,* should be placed before the reference (Broecker, 1982a), which was intended to be given as an example of a key study for the role of ocean sedimentary processes for increased glacial oceanic storage of carbon.

The Introduction has been re-written, and this citation is now located on page 2, line 20, together with other relevant references. The formatting issue is thus no longer present.

**3.3 Methods**

– *Page 4, line 20 : suggest citing the more recent estimate of 2.6°C by Bereiter et al (2018): Mean global ocean temperatures during the last glacial transition. Nature.*

We thank PJB for the suggestion and will add this reference.

The sentence has been changed to: *'By applying the above perturbations, we aim to approach, but not fully resolve, some of the characteristics of the Last Glacial Maximum (LGM) ocean, which appears to have had a global average ocean temperature ($\overline{T_{oce}}$) 2.57 ±0.24 °C colder than the Holocene (Bereiter et al., 2018), [...]'*, see page 5, lines19-21. See also page 2, line 10; page 7, line 34; and page 13, line 4.

– *Page 6, line 1 : suggest citing Moore et al (2013) Processes and patterns of oceanic nutrient limitation. Nature Geoscience. Another paper of note that would be useful for your work is the recent article by Garcia et al (2018) Nutrient supply controls particulate elemental concentrations in the low latitude eastern Indian Ocean. Nature Communications.*

We thank PJB for the suggestions and add the reference to Moore et al. (2013) here. The paper by Garcia et al. (2018) is certainly relevant to our work, and this reference will be added already in Section 2.2 (Page 3).

References added in text, see Page 7, line 13 (Moore et al., 2013), and Page 4, line 16 (Garcia et al., 2018a).

– *Page 6, line 17 : "see Table ??"... Please make sure your document is properly formatted before submitting.*

We apologise for the formatting error. *Table ??* should be corrected to *Table 2*.

Missing table number corrected to Table 2, see page 7, line 28.

– *Page 6, line 22 : Why not use more recent WOA 2018 product?*

When the paper was submitted, WOA18 was only available as a pre–release. We therefore chose to stay with the most up–to–date official release for the initial submission. The official release is now available, and we will therefore use WOA18 for the updated manuscript.

In the updated manuscript, we have replaced WOA13 for WOA18 throughout the manuscript, in text, figures and tables. See e.g., page 7, line 32: *'Modern data of ocean temperature, oxygen and nutrients are retrieved from the World Ocean Atlas 2018 (Locarnini et al., 2018; Garcia et al., 2018c, b)'.*

– *Page 7, line 13 : Again a question mark is present. Please format properly before submitting.*

We are very sorry for having overlooked the formatting errors in our final check before submission. The missing reference marked by *(?)* should be (Ödalen et al., 2018).

Missing reference corrected to (Ödalen et al., 2018), see page 8, line 23.

**3.4 Results**

– *Page 8, line 29 : The oxygen content of the ocean should be lower in $Ctrl_{121}$ than for $Ctrl_{RED}$. This is because a higher C:P ratio of organic matter should also require more $O_2$ to remineralise that organic matter. I would like an explanation of why $O_2$ is higher in $Ctrl_{121}$ than for $Ctrl_{RED}$. See Paulmier, Kriest & Oschlies (2009): Stoichiometries of remineralisation and denitrification in global biogeochemical ocean models. Biogeosciences.*

We thank PJB for having identified this inconsistency in our results, which we had failed to notice. This was caused by an issue in the code, which caused an unintentional change in $O_2$:C when the fixed C:P stoichiometry was changed. This issue was not present in the runs with flexible stoichiometry, and thus only affected the $O_2$ in the 121–ensemble. We have re–run $Ctrl_{121}$ after having corrected the model code. The new resulting average $O_2$-concentration is 152 $\mu mol kg^{-1}$, which is lower than $Ctrl_{RED}$ (166 $\mu mol kg^{-1}$), but higher than $Ctrl_{GAM}$ (144 $\mu mol kg^{-1}$).

The $O_2$-concentrations of the 121-experiments have been corrected throughout the manuscript, and we have added to Methods Section 2.2, Stoichiometry, page 4, lines 25-26: *'Note that, while we change the ratio C/P, the ratio $C/O_2$ remains the same in all experiments. As a result, the $P/O_2$ ratio changes between experiments.'.*

– *Page 11, line 1 : Please explain why a shallow RLS is shallower in a warmer climate. Alternatively, you could provide a more thorough explanation of the effect of temperature on remineralisation rates in the Introduction.*

In line with PJBs suggestion in the General comments, this will be described in more detail in the Introduction, based on the results of Matsumoto (2007).

The detailed description of temperature dependent remineralisation rates was too long for the Introduction, and in the updated manuscript it is found in Section 2.3.2, where the sensitivity experiments are described.

Added to Introduction, page 3, lines 15-16: *'Ocean cooling reduces the degradation rate of sinking particulate organic carbon, which increases the average depth of remineralisation of organic carbon (Matsumoto, 2007).'*

Added to Section 2.3.2, first paragraph: *'In the ocean, phytoplankton growth rates and remineralisation of particulate*

*organic carbon are processes that both work more slowly at colder temperatures (Eppley, 1972; Laws et al., 2000). Cooling of the ocean would thus lead to decreased production of particulate organic matter (POC), and simultaneously to a slower degradation of POC, with competing effects on export production (i.e. the amount of C captured by primary production that leaves the surface ocean without being remineralised) (Matsumoto, 2007). However, Matsumoto (2007) shows that the effect of slower remineralisation dominates the effect on export production. It has therefore been hypothesised that the cooling of the glacial ocean led to a deepening of the remineralisation length scale (henceforth denoted RLS) in the ocean, and thereby more efficient retention of organic carbon in the deep ocean (Matsumoto, 2007; Chikamoto et al., 2012), which in turn caused a lowering of $pCO_2^{atm}$.'*

– **Page 11, line 9 : This sentence needs to be clearer with what it's trying to say. Roughly 10% of what? Change in the average composition of what? Of course I can guess what you mean when I stop to think about it, but please make it easier for the reader by saying what you mean.**

We agree with PJB and in order to clarify the sentence, we will change it to: *About a third ($\sim 10$ % of 30 %) of the increase in drawdown can be explained by a change in average C/P composition of the organic material that is exported out of the surface ocean.* This will then be discussed in more detail in Section 4.2, where we will gather all the results and discussion that concerns the 121–ensemble (as suggested by Referee #1).

Sentence re–written as suggested in author's response, see page 12, lines 19-21. Results from experiments with changed fixes C/P described in more detail in Section 4.4. (former Section 4.2).

– **Page 11, lines 22–25 : But not cool enough to align with the more recent estimate of Bereiter et al (2018): Mean global ocean temperatures during the last glacial transition. Nature.**

We will add a note on this, and clarify that we are not applying all forcings that are expected to be needed to reproduce a full glacial state.

Added on page 13, lines 7-9: *'Compared to the Bereiter et al. (2018) estimate, our combined experiments $GL_{comb}$ and $A_{comb}$ achieve 64 and 82 % of the glacial–interglacial difference in $\overline{T_{oce}}$, respectively. As anticipated, our combined forcings do not induce a full glacial maximum state, but a state with glacial–like climate conditions.'*

– **Page 12, lines 21–25 : Again, I am unsure how you are treating P:O$_2$ remineralisation requirements in your variable stoichiometry experiments. I think this should be explained. It is also strange once again that your $GLcomb_{121}$ experiment is better oxygenated than your $GLcomb_{RED}$ experiment.**

In remineralisation, there was an issue in the code which affected the C:O$_2$ ratio in the 121–experiments. This made the P:O$_2$ requirements appear strange, which PJB noticed. This has been corrected, and the 121–experiments re–run. After correction, C:O$_2$ remains the same in all experiments, while the P:O$_2$ ratio, as a result, changes between experiments. We will clarify this in the Methods section. After the correction described abobe, $GLcomb_{121}$ has a lower global average $O_2$ concentration than $GLcomb_{RED}$ (96 compared to 122 $\mu mol kg^{-1}$), but higher than $GLcomb_{GAM}$ (74 $\mu mol kg^{-1}$). The O$_2$-concentrations of the 121-experiments have been corrected throughout the manuscript, and we have added to

Methods Section 2.2, Stoichiometry, page 4, lines 25-26: *'Note that, while we change the ratio C/P, the ratio C/O$_2$ remains the same in all experiments. As a result, the P/O$_2$ ratio changes between experiments.'*.

- *Figure 5 : The panels in this figure do not seem to be arranged correctly.*

  We are grateful to both referees for having identified this error. We here provide the updated figure (see Fig. 1), where the sub–panels have been re–arranged in the correct order. Per request of Referee #1, we will also clarify the caption of this figure.

  The figure and caption have been corrected and updated in accordance with the author's responses to both referees, and have been added to the manuscript. Note that the clarifications also apply to Fig. S.3.

- *Page 13, lines 1–8 : What conditions affect the fractionation strength of biological carbon assimilation? Is it constant or variable?*

  The fractionation strength is variable and will be detailed in an appendix (see below, in author's response to the final bullet point of Discussion).

  Added Appendix B: $\delta^{13}C$ in cGENIE, page 22.

- *Page 13, line 12 : Why not add a figure of sea ice cover in the supplement? Also, can you separate the effects of sea ice cover expansion from the other physical changes in terms of CO$_2$? A few studies since the Stephens & Keeling (2000) paper have found that an increase in sea ice cover under glacial conditions actually reduces ocean carbon storage because it prevents organic carbon production. It would be worthwhile to separate this effect from temperature and circulation and note if it is positive or negative on atmospheric CO$_2$. Stephens & Keeling (2000) The influence of Antarctic sea ice on glacial–interglacial CO$_2$ variations. Nature. Kurahashi–Nakamura et al. (2007) Compound effects of Antarctic sea ice on atmospheric $pCO_2$ change during glacial–interglacial cycle. Geophysical Research Letters. Sun & Matsumoto (2010) Effects of sea ice on atmopsheric $pCO_2$: A revised view and implications for glacial and future climates. Journal of Geophysical Research. Buchanan et al (2016) The simulated climate of the Last Glacial Maximum and insights into the global marine carbon cycle. Climate of the Past.*

  We will add a figure of the sea ice anomaly ($GLcomb-Ctrl$) to the supplement. The separation of the effects of sea ice cover expansion from the other physical changes would be highly interesting to isolate. However, we find them to be beyond the scope of this paper, and we wish to keep this paper focused on the effects of flexible C/P on $pCO_2$. If PJB is interested in the separation of physical effects in cGENIE, this is explored in (Ödalen et al., 2018). There, we want to point specifically to the simulation $AD/2$ in Figure 2, where panel e) details the contributions from the changes in each of the different carbon capture processes to the net change in $pCO_2$. The simulation $AD/2$ has a reduced biological pump compared to the pre–industrial control, but an enhanced carbon capture due to increased disequilibrium and saturation carbon. This is attributed mainly to ocean cooling and a resulting expansion of sea ice. However, no further separation of the effect of sea ice was made.

  Added Figure S.4, see reference on page 14, line 24, and Supplementary Material.

– *Page 13, lines 10–23 : This paragraph would benefit from being clearer in its findings of $CO_2$ sequestration regarding C:P ratio changes. I have to read this multiple times to understand what the authors are trying to say when comparing Redfield, variable C:P and C:P=121.*

This paragraph will be clarified by concentrating all the results and discussion concerning C/P = 121 to Section 4.2, as suggested by Referee #1. We are confident that this paragraph will become clearer when it is rewritten to focus only on the difference in $CO_2$ sequestration between Redfield ($RED$) and flexible C/P ($GAM$).

Section 3.3.5, referred to here by PJB, has been re–phrased, shortened and information regarding experiments using model version 121 has been moved to Section 4.4

**3.5   Discussion**

– *Page 13, line 28 : your reference to "model" should be an "empirical model" to avoid confusion with the Earth System Model, GCM, etc.*

Agreed. We will add "empirical".

Added *'empirical'*, page 15, line 7.

– *Page 14, lines 1–5 : The advantages of using empirical/statistical models within biogeochemical ocean GCMs, including the Galbraith & Martiny (2015) parameterisation, was explored rigorously in my 2018 paper in Global Biogeochemical Cycles. It not only improved that model's biogeochemistry significantly, but also altered the long–term behaviour of the carbon cycle as you have also found. It may be interesting, but I of course leave it up to you whether it's useful. Buchanan et al (2018) The importance of dynamic biological functioning for simulating and stabilizing ocean biogeochemistry. Global Biogeochemical Cycles.*

We thank PJB for pointing us to this paper, and find the conclusions very interesting. They are well aligned with the study we perform in this paper, and by citing this paper, we will strengthen our arguments. We will thus include the suggested paper in our discussion.

Added to Introduction, page 3, lines 7-10: *In addition, Buchanan et al. (2018) explored the importance of dynamic response of ocean biology, such as flexible stoichiometry, for modelled ocean biogeochemistry in pre–industrial simulations. They found that the dynamic response was fundamental for stabilising the response of ocean DIC to changes in the physical circulation state.*

Added to Section 4.1, page 15, lines 10-16: *'Previous model ensemble studies have shown that this type of dynamical response of the biology to changes in the modelled ocean state can improve the model's ability to realistically simulate ocean biogeochemistry (Buchanan et al., 2018). In pre–industrial and future simulations, respectively, Buchanan et al. (2018) and Tanioka and Matsumoto (2017) find that the flexible stoichiometry acts to stabilise the response of ocean DIC to changes in the physical (circulation) state. In our glacial–like simulations, we find that the response of ocean DIC, and thus $pCO_2^{atm}$ , to the combined perturbations is greater in the simulations with flexible stoichiometry. Nonetheless, our study confirms the potential importance of dynamical biological response for the outcome of model studies.'*

- *Page 14, line 32 : the conclusions of Odalen et al (2018)... which were? What did Odalen et al (2018) do?*

  Here, lines 30–33 should read: *Note that $Ctrl_{GAM}$ has a larger inventory of DIC, as well as $C_{soft}$, compared to $Ctrl_{RED}$. Ödalen et al. (2018) found that drawdown of $CO_2$ in response to a perturbation is larger when the control state has a smaller inventory of DIC and $C_{soft}$. Yet, the effect of applying the same perturbation results in a larger drawdown of $CO_2$ in GAM than in RED. This is thus opposite of the conclusions of Ödalen et al. (2018). The reason is that the flexible stoichiometry in effect increases the drawdown potential, which more than compensates for the increased carbon inventory in the control state.* As stated above, Section 4.2 will be re–written, and clarified. In this way, we will assure that the conclusions of Ödalen et al. (2018) are clearly stated.

  Added sentence *'Ödalen et al. (2018) found that drawdown of $CO_2$ in response to a perturbation is larger when the control state has a smaller inventory of DIC and $C_{soft}$.'* (see p. 17, lines 14-16), as suggested in author's response.

- *Page 15, lines 19–24 : So the proportion of remineralised to preformed phosphorus effectively doesn't change in the simulations? And yet, you find a large increase in respired C? This must mean that the remineralised phosphorus that is exported into the ocean interior in your $GLcomb$ simulation is being quickly circulated into the lower overturning cell and returned to the Antarctic Zone, where sea ice prevents gas exchange and biological production, at which point this P is recirculated and becomes preformed, while respired C remains respired and is also recirculated. If this is the case, it merits more discussion in comparison with previous literature on the subject of a more efficient biological pump that invokes more regenerated nutrients as a must for a more efficient biological pump. I suggest Hain, Sigman & Haug (2014) The biological pump in the past. Treatise on Geochemistry, 2nd Ed.*

  The referee has correctly identified that, depsite the fact there is no change in remineralised P ($P_{rem}$) in $GLcomb_{GAM}$ compared to $Ctrl_{GAM}$, there is an increase in remineralised C. However, the process described by the referee focuses on what happens after remineralisation, while we suggest that this decoupling happens before the organic material is exported to the deep ocean (see lines 25–31). The forcing components applied to $GLcomb_{GAM}$ have competing effects on the amount of organic matter that remineralises in the deep ocean. The net effect is that this amount does not change globally (reflected by a constant $P_{rem}$ compared to $Ctrl_{GAM}$). Meanwhile, changes in ocean circulation, remineralisation depth and dust deposition still cause the local nutrient availability in the surface waters to change. This affects the elemental composition of the exported organic material. In $Ctrl_{GAM}$, the average elemental C/P composition is 121/1. In $GLcomb_{GAM}$, this average is 134/1. This means that even though the same amount of P is exported to the deep ocean, the organic molecules carry more carbon, which is released in the deep ocean during remineralisation. In $Ctrl_{GAM}$, the global average concentration of $P_{rem}$ is 1.16 $\mu mol kg^{-1}$ (c.f. 1.17 $\mu mol kg^{-1}$ in $GLcomb_{GAM}$). By increasing the average C/P composition of 1.16 $\mu mol kg^{-1}$ organic molecules from 121 to 134 (i.e. by 13 units), this causes an increase in $C_{rem}$ by $\sim 15$ $\mu mol kg^{-1}$, which corresponds to the observed increase in $C_{rem}$. In summary, we suggest this is a result of changes in surface P fields (see Fig. 7), rather than a change in the partitioning between $C_{rem}$ and $P_{pre}$ in the recirculation area in the Antarctic Zone. We will clarify this part of the discussion, which comprises lines 19–31 on page 15.

Section 4.2 (i.e. Section 4.3 in the original manuscript) has been re–written to include the discussion outlined in the author's response, see page 16, line 17 through page 17, line 13.

– *Page 16, lines 1–14 :*

Here, each of the questions will be treated separately.

– *I would like to see how variations in P:$O_2$ requirements were treated in this model.*

C:$O_2$ requirements were meant to be held constant throughout the simulations, consequently causing changes in P:$O_2$. However, an inconsistency in the code caused C:$O_2$ requirements to change in the simulations of the 121– ensemble, causing an inconsistent behaviour of P:$O_2$. This has been corrected, as outlined above.

The $O_2$-concentrations of the 121-experiments have been corrected throughout the manuscript, and we have added to Methods Section 2.2, Stoichiometry, page 4, lines 25-26: *'Note that, while we change the ratio C/P, the ratio C/$O_2$ remains the same in all experiments. As a result, the P/$O_2$ ratio changes between experiments.'*

– *Also, can you please explain why the deep water formation characteristics of a model affects $O_2$?*

Deep water formation characteristics of a model affects the amount of time available for remineralisation and, consequently, the oxygen consumption. In addition, due to a lack of resolution deep water formation in climate models generally happens as open water convection, rather than as dense plumes along slopes. This causes too much oxygen to be entrained into the deep ocean. We will add this explanation in the updated manuscript.

Page 18, lines 1-12, paragraph extended by lines 6-12: *'Among other factors, model ocean oxygen conditions are also dependent on deep water formation characteristics of the model (Galbraith and de Lavergne, 2018). The deep water formation characteristics of a model affects the amount of time available for remineralisation and, consequently, the oxygen consumption. In addition, due to a lack of resolution deep water formation in climate models generally happens as open water convection, rather than as dense plumes along slopes (Heuzé et al., 2013). This may cause too much oxygen to be entrained into the deep ocean Galbraith and de Lavergne (2018). In cGENIE, this effect is small enough not to cancel the increased $O_2$ consumption caused by the higher average C/P in $Ctrl_{GAM}$ compared to $Ctrl_{RED}$.'*

– *Overall, I find this section a bit sparse and I'm not entirely sure what the point of it is.*

The section aims to discuss 1) to what extent our $GLcomb$ simulations reproduce proxy observations, in this case for $O_2$, and 2) to discuss one of the problems that arose from applying the flexible C/P in GENIE, i.e. that $O_2$ concentrations in $Ctrl_{GAM}$ are too low, and its implications for the glacial–like simulation. We agree that the section could be expanded, and we will add the information requested by PJB to make the discussion more comprehensive.

Discussion has been expanded according to suggestions and questions by PJB in the above and below bullet points.

– *Can you comment on the size of the OMZs? New evidence shows that the OMZs in the Pacific expanded vertically during the glacial. Hoogakker et al (2018) Glacial expansion of oxygen–depleted seawater in the eastern tropical Pacific. Nature*

In the Atlantic, $Ctrl_{GAM}$ (Fig. 4 e) reproduces the observed extent of the OMZ (Fig. 4 a) better than $Ctrl_{RED}$ (Fig. 4 c) does. In the Pacific Ocean, the $O_2$ gradient in the observations (Fig. 4 b) is more gradual compared to that of the control states (Fig. d, f), but the core of the OMZ is well reproduced by the model. The forcings applied to $GLcomb$ are not sufficient to reproduce a full glacial state (see also author's response to the next main point, regarding Page 16, line 22). Still, we do get a vertical expansion of the OMZ in $GLcomb_{RED}$ (Fig. 4 h) compared to $Ctrl_{RED}$ (Fig. 4 c), in agreement with the findings of Hoogakker et al (2018). In $GLcomb_{GAM}$, oxygen depletion is too extensive, but the tendency of vertical expansion compared to the control state is present here as well.

Added discussion of glacial vertical expansion of the OMZ in line with author's response (see page 17, lines 25-30).

– *Page 16, line 22 : And yet others achieve relatively good correlations in the Pacific basin using the same data? Menviel et al (2017) Poorly ventilated deep ocean at the Last Glacial Maximum inferred from carbon isotopes: A data–model comparison study. Paleoceanography. Muglia et al (2018) Weak overturning circulation and high Southern Ocean nutrient utilization maximized glacial ocean carbon. Earth and Planetary Science Letters. I think you cannot say that your model doesn't provide good fits to the LGM proxy data because of poor data coverage, and it would be more useful to discuss why the model doesn't fit with the data. It seems like your glacial circulation in the Pacific basin is therefore not accurate?*

The forcings applied to $GLcomb$ are factors that are likely to be important for the glacial ocean circulation and biogeochemistry. However, these forcings are not sufficient to reproduce a full glacial state (i.e. the use of the term "glacial–like simulations", rather than "LGM simulation"). Other forcings that have shown to be important for modelling of glacial $\delta^{13}C$ are, for example, brine rejection (Bouttes et al., 2010, 2011), and freshwater forcing (e.g., Schmittner et al., 2002; Hewitt et al., 2006; Bouttes et al., 2012). The fact that some important forcings are missing (mentioned on lines 27–28) is likely the main cause for the model–data discrepancy, and the reason for why we do not achieve an accurate glacial Pacific Ocean circulation. This will be clarified in the section. We will also clarify, throughout the paper, the fact that we do not aim to produce a full LGM state. This may also call for adjusting the title of the paper.

The discussion in this section (Section 4.5) has been re–phrased and extended to give more detailed explanations of the reasons for model–data discrepancies. For this referee comment, see specifically page 19, lines 25-31: *'The forcings applied to $GL_{comb}$ are factors that are likely to be important for the glacial ocean circulation and biogeochemistry. However, these forcings are not sufficient to reproduce a full glacial state (i.e. the use of the term glacial–like simulations, rather than LGM simulation). Other forcings that have shown to be important for modelling of glacial $\delta^{13}C$ are, for example, brine rejection (Bouttes et al., 2010, 2011), and freshwater forcing (Schmittner et al., 2002; Hewitt et al., 2006; Bouttes et al., 2012). The fact that some important forcings are missing is likely the main cause for the model–data discrepancy, and the reason for why we do not achieve a glacial Pacific Ocean circulation consistent with observed $\delta^{13}C$ patterns.'*

*In addition, the title of the paper has been adjusted to: 'Variable C/P composition of organic production and its effect on ocean carbon storage in glacial–like model simulations'*

– *Page 16, line 26 : Temperature is not chemical. I also do not understand how you could alter the temperature and salinity of the ocean without altering water mass distributions, and if this is indeed the case, it requires further description as to why earlier in the paper. Also a good spot to talk about why the data in the Pacific are not well reproduced by the model.*

We agree that temperature in itself is not chemical. Here, we were referring to the changes in solubility of $CO_2$, which is a chemical response to changes in temperature. This will be clarified in the revised version of the paper. The water mass distribution in cGENIE is strongly constrained by the resolution of the model, especially in the vertical. Changes in temperature and salinity that should cause changes in water mass volume may not be sufficient to allow a water mass to extend to the next vertical level of the model. As a consequence, while the gradient between water masses may become more or less pronounced, the interface of water masses may still remain at the same depth. The section will be clarified, including the above description of why Pacific glacial circulation is not fully reproduced.

*The discussion in this section (Section 4.5) has been re–phrased and extended to give more detailed explanations of the reasons for model–data discrepancies. For this referee comment, see specifically page 19, lines 32-34, and page 20, lines 1-10. (Due to the length of the added paragraph, we do not cite it here.)*

– *Page 16, lines 24–31 : I don't follow this paragraph. You state that "Each of the two observational datasets (HOL and LGM) display similar correlations across the two model simulations. This implies that our changes in forcings do not achieve any obvious changes in water mass distribution." But doesn't the distribution of $\delta^{13}C$ change across the glaciation and into the Holocene? $\delta^{13}C$ in the Atlantic, for instance, is often used to show that the Atlantic meridional overturning was shallower during the glacial, and that this change occurred during Marine Isotope Stage 4 (Oliver et al (2010) A synthesis of marine sediment core $\delta^{13}C$ data over the last 150000 years. Climate of the Past.)? Moreover, $\delta^{13}C$ is used as a way to show that the water mass distribution between the Atlantic and Pacific was considerably different during the glacial as compared to the Holocene (Sikes et al (2017) Enhanced $\delta^{13}C$ and $\delta^{18}O$ Differences Between the South Atlantic and South Pacific During the Last Glaciation: The Deep Gateway Hypothesis. Paleoceanography.) These studies conflict with what you are saying.*

The proxy data do imply a change in $\delta^{13}C$ across the deglaciation (whole ocean change 0.34 $\pm 0.19$ ‰, Peterson et al., 2014). What we are trying to say is that our model simulations do not fully reproduce this change. Here we are referring to the fact that the correlation of the HOL proxy records with $Ctrl_{RED}$, $Ctrl_{GAM}$, $GLcomb_{RED}$, and $GLcomb_{GAM}$, is in all cases between 0.76–0.78. On the other hand, the correlation of LGM proxy records with the same four simulations is in all cases between 0.55–0.58. As our $GLcomb$–simulations still correlate so well with the HOL dataset, this suggests the applied forcings have not caused these simulations to be clearly different from $Ctrl$ in terms of water mass distribution. For the same reason, the correlation with LGM proxy data does not significantly improve from $Ctrl$ to $GLcomb$. Thus, the deglacial change in $\delta^{13}C$ reflected in the proxy data is not fully captured by the model. This will be clarified in the

updated version of the manuscript.

The discussion in this section (Section 4.5) has been re–phrased and extended to give more detailed explanations of the reasons for model–data discrepancies. For this referee comment, see specifically page 19, lines 12-17; page 19, lines 32-34; and page 20, lines 1-10. (Due to the combined length of the added paragraphs, we do not cite them here.)

5    – *(Section 4.5 ?): The ability for simulated $\delta^{13}C$ to reproduce the proxy data at the LGM will depend strongly on water mass distribution (which apparently doesn't change appreciably) and how biological fractionation is parameterised. If it is constant, the 10% loss in C fixation will cause the ocean to be more positive overall by some constant factor. However, if the parameterisation contains a dependence on aqueous $CO_2$ and growth rate, both of which are lower, then the fractionation will vary. It would be worthwhile telling the reader what parameterisation is used and, if it does*

10    *involve growth rate and aqueous CO2, what effect this has.*

The fractionation is dependent on both aqueous $CO_2$ and growth rate (represented in $K_Q$, see appendix). This dependence and its consequences will be detailed by adding the following text in an Appendix:

*[... Appendix text given in Author's response removed, as it is now included in the updated manuscript.]*

Added sentence on page 20, line 11: *'How $\delta^{13}C$ is represented in cGENIE is detailed in Appendix B.'*

15    Added also Appendix B: $\delta^{13}C$ in cGENIE (page 22).

**4 Technical corrections**

– *Page 1, line 13 : remove repeated "with"*

We will correct this typo. Page 1, line 13: Removed repeated "with".

– *Page 5, line 16 : replace "reduced half" with "halved"*

20    The sentence will be changed according to the suggestion.

Page 6, line 16: Replaced "reduced half" with "halved".

– *Table 1 : "witg" to "with"*

We will correct this typo in the caption of Table 1.

Table 1, caption: Changed "witg" to "with".

25    – *Page 11, line 8 : Inadvisable to begin a sentence with "$\sim$"*

The sentence has been rewritten (see above comment for Page 11, line 9).

Page 12, lines 19-21: Sentence re–written.

– *Page 11, line 29 : "SV" to "Sv"*

We will change "SV" to "Sv" throughout the manuscript.

30    "SV" replaced by "Sv" throughout the manuscript.

– *Page 13, line 26 : "GCMs" this acronym has not been defined previously.*

We will define the acronym here.

Page 15, line 5: Acronym "GCMs" defined as General Circulation Models.

– *Page 15, line 23 : I assume you mean "0.003" rather than "0003"?*

5        Yes, we will correct this to 0.003.

Page 16, line 32: "0003" corrected to "0.003"

**References**

[revised manuscript text omitted]

**List of relevant changes**

- Abstract
  - o Minor changes as suggested by referee #2

- Introduction
  - o The introduction has been expanded to give more background to the ocean's role in the glacial-interglacial $CO_2$ problem, as suggested by referee #2.

- Methods
  - o 2.1-2.2, only minor changes
  - o 2.3 Replaced estimate of glacial-interglacial global ocean average temperature difference by Headly & Severinghaus (2007) for the more recent estimate by Bereiter et al. (2018).
    - ▪ 2.3.1 Clarified descriptions and added references, described air-sea gas exchange dependence on wind speed as suggested by referee #1
    - ▪ 2.3.2 Expanded description of temperature dependent remineralisation, and added references.
  - o 2.4 Replaced WOA13 for WOA18
  - o 2.5 Added missing reference to Ödalen et al. (2018)

- Results
  - o 3.1 Minor changes
  - o 3.2
    - ▪ 3.2.1 Added explanation of why salinity changes are not explored
    - ▪ 3.2.2-3.2.4 Moved information about 121-simulations to Section 4.4, as suggested by both referees.
  - o 3.3
    - ▪ 3.3.1 Replaced estimate of glacial-interglacial global ocean average temperature difference by Headly & Severinghaus (2007) for the more recent estimate by Bereiter et al. (2018).
    - ▪ 3.3.2-3.3.4 Minor changes
    - ▪ 3.3.5 Text has been clarified and information about 121-simulations has been moved to Section 4.4

- Discussion
  - o 4.1 Text has been clarified and expanded by discussing Buchanan et al. (2018). We have expanded the discussion on the implications for warm climate scenarios.
  - o 4.2 The former Section 4.2 is now Section 4.4. The section now includes all the information about the 121-simulations, which has been moved here from the results section.
  - o 4.3 The former Section 4.3 is now Section 4.2. This section has been clarified and expanded in response to referee comments.
  - o 4.4 The former Section 4.4 is now Section 4.3. This section has been expanded to align with suggestions from referee #2.
  - o 4.5 The section has been expanded in response to comments of referee #2.

- Conclusions
  - o Added sentence about potential implications for warm climate scenarios.

- Figures
  - o Fig. 3 Updated to WOA18
  - o Fig. 4 Updated to WOA18

- o Fig. 5 Both referees pointed out the original version of this figure was incorrect, and found it difficult to interpret. The order of figure panels has been corrected. The description of the figure in the legend and the figure caption has been improved.

- Tables
  - o Table 2 Format of table changed as suggested by referee #1.

- Supplementary material
  - o Fig. S.3 Improved for clarity (c.f. Fig. 5)
  - o Fig. S.4 New figure added (sea-ice anomaly)

[revised manuscript text omitted]

---

## Referee Report (RR1)

Pearse James Buchanan

February 26, 2020

**Department of Earth, Ocean and Ecological Sciences, University of Liverpool, Liverpool, UK.**

In their revised manuscript, Odalen and coauthors have made a strong attempt to address the many demands of the two reviewers. The writing is much clearer so I thank them for that, and as a result the discussion of their findings is much easier to understand. Their justification for focussing on the Atlantic Ocean given more observations of benthic $\delta^{13}$C records in this ocean basin than in the Indo-pacific is also more obvious and I find myself more convinced of this approach.

I have only a few minor, specific comments (see below). *I recommend publication of the article*.

**1  Specific comments**

**Results**

- Page 9, lines 23-30: But the C:P "observations" are not actually observations. You have applied the Galbraith & Martiny (2015) empirical model to the WOA PO$_4$ data. I would suggest either changing your wording from "observations", or actually using the observations by overlaying them on the map of C:P, which are available from Martiny (2014) Scientific Data. There may in fact be a newer version. If you wish to make a comparison with the actual observations, you'll find much greater model-data error given the high variability in these data, which is difficult to explain without invoking subseasonal variability and other environmental predictors, such as fine scale variations in light and nutrient stress that your model does not resolve. So I leave this up to you whether you wish to include and comment on. If not included, then a change in wording is necessary.

- Page 14, lines 1-6: This is a great result and the following is more comment than review. Achieving a decline in global $O_2$ as a result of variable stoichiometry is very interesting, as previous model studies have found it difficult to accomplish. Laurent Bopp's work, published in 2017 in Phil. Trans. Royal Society, for instance, achieved an LGM deoxygenation only by applying large amounts of freshwater to the North Atlantic. As such it can be thought of as a transient response, not a steady-state solution as you have achieved here. Again, just a comment, so feel free to include this new information or not.

**2   Technical corrections**

1. Page 1, line 21: missing closing parenthesis
2. Page 3, line 1: "suggest" –> "suggests"
3. Page 10, line 7: do you mean WOA 2018?
4. Page 16, line 27: Should this line be a new paragraph or is it attached to the previous paragraph?
5. Page 23, line 1: "Pierce" –> "Pearse"
6. Page 38, Figure 9: "origo" –> "origin" ?

---

## Author Response (AR2)

**Changes in manuscript based on Associate Editor report and interactive comment by referee #2, Pearse James Buchanan.**

Malin Ödalen[1,2], Jonas Nycander[1], Andy Ridgwell[3,4], Kevin I. C. Oliver[5], Carlye D. Peterson[3], and Johan Nilsson[1]

[1]Department of Meteorology, Bolin Centre for Climate Research, Stockholm University, 106 91 Stockholm, Sweden
[2]Department of Geosciences, University of Arizona, Tucson, AZ 85721, USA
[3]Department of Earth Sciences, University of California–Riverside, Riverside, CA 92521, USA
[4]School of Geographical Sciences, Bristol University, Bristol BS8 1SS, UK
[5]National Oceanography Centre, Southampton, University of Southampton, Southampton SO14 3ZH, United Kingdom

**Correspondence:** Malin Ödalen (malin.odalen@gmail.com)

**1 Introduction**

We thank referee #2, Pearse James Buchanan (henceforth PJB), for the final remarks on the paper, and for helping to add the final touches to the paper. PJB recommends some minor changes to the manuscript before publication. Below, in Sections 2–3, we address his comments and present the associated changes in the manuscript. We also make some additional minor changes to the manuscript, which are described in Section 4.

In this document, referee comments are shown in ***bold and black italics***, and author's discussion response directly below in plain black. For each author's response, the corresponding changes in the manuscript are shown in blue.

**2 Specific comments**

**2.1 Results**

– *Page 9, lines 23 –30 : But the C:P "observations" are not actually observations. You have applied the Galbraith & Martiny (2015) empirical model to the WOA $PO_4$ data. I would suggest either changing your wording from "observations", or actually using the observations by overlaying them on the map of C:P, which are available from Martiny (2014) Scientific Data. There may in fact be a newer version. If you wish to make a comparison with the actual observations, you'll find much greater model-data error given the high variability in these data, which is difficult to explain without invoking subseasonal variability and other environmental predictors, such as fine scale variations in light and nutrient stress that your model does not resolve. So I leave this up to you whether you wish to include and comment on. If not included, then a change in wording is necessary.*

We agree with PJB that the climatological mean WOA $PO_4$ data are not observations per se, though they are based on observations. Therefore, we also agree that a change in wording is appropriate. We choose not to overlay the figure

with actual observations, as we feel it would make the figure too busy. However, the caveats associated with the lack of variability associated with light and nutrient stress are discussed in Section 2.1. Associated changes are made throughout the manuscript.

In the updated version, Page 10, lines 2 + 8, and line 5, we replace "observations" by "climatological mean $PO_4$ fields", and "climatological mean $PO_4$ concentrations", respectively.

Associated changes are made on Page 8, line 4; Page 10, line 15; Page 17, lines 28–29; Page 18, lines 6–7; Figure 3, caption; and Figure 4, caption.

Minor adjustments to the text on Page 15, line 22, and Page 16, line 4, were also made based on this comment.

- *Page 14, lines 1–6 : This is a great result and the following is more comment than review. Achieving a decline in global $O_2$ as a result of variable stoichiometry is very interesting, as previous model studies have found it difficult to accomplish. Laurent Bopp's work, published in 2017 in Phil. Trans. Royal Society, for instance, achieved an LGM deoxygenation only by applying large amounts of freshwater to the North Atlantic. As such it can be thought of as a transient response, not a steady–state solution as you have achieved here. Again, just a comment, so feel free to include this new information or not.*

We thank PJB for the interesting comment. We feel that related information is already discussed, but from a different perspective, in Section 4.3 (see e.g. page 18, lines 6–14 in the updated manuscript). For the sake of the lenght of the paper, we therefore choose not to extend the discussion with the new information.

No change in manuscript.

**3 Technical corrections**

- *Page 1, line 21: missing closing parenthesis*

Page 2, line 2: Closing parenthesis added.

- *Page 3, line 1: "suggest" –> "suggests"*

Corrected. Note that the error was found on page 5, now on line 6.

- *Page 10, line 7: do you mean WOA 2018?*

Yes, we have missed changing this line after moving from using WOA 2013 to using WOA 2018.

Page 10, Line 15: World Ocean Atlas 2013 corrected to World Ocean Atlas 2018.

- *Page 16, line 27: Should this line be a new paragraph or is it attached to the previous paragraph?*

The line should be placed at the end of the paragraph that was starting on line 28. The line had accidentally been moved.

Line on page 16 moved from line 27 to Page 17, 2–3.

- *Page 23, line 1: "Pierce" –> "Pearse"*

  We apologise for misspelling PJB's first name, and have corrected this in the manuscript.

  Spelling corrected on Page 23, now line 6.

- *Page 38, Figure 9: "origo" –> "origin" ?*

  "origo" changed to "origin"

**4 Additional changes**

- The corresponding author has a new affiliation and email address.

  Affiliations and email address updated for the corresponding author.

- In the first round of revisions, Fig. S.4 was added to the supplementary. However, the figure had not been referenced in the updated manuscript as intended. This is now updated.

  Page 13, line 26: "(not shown)" updated to "(Fig. S.4)".

- During revisions, we have come to realise that a graphical representation of C/P as a function of $\overline{P^*}$ can help the reader in Section 4.2. This data is also available in Table S.1, and the figure should be considered a complement to Fig. 9 and Table S.1. We would therefore like to include it as Fig. S.5 in the supplementary information.

  Page 17, lines 6 and 10: added references to Fig. S.5 and Table S.1.

  In Supplementary material, added text on Page 2, lines 13–14. Added Fig. S.5 on Page 7 (see Fig. 1 below).

- Duplicate reference found to Le Quéré et al. (2005 a/b) on Page 26, line 35 to Page 27, line 3.

  Duplicate reference removed. Reference to Le Quéré et al. (2005) no longer needs to be specified by a or b (see Page 4, line 20; Page 15, line 9

- Note the PDF comparison suggests that the configuration of $pCO_2^{atm}$ and $PO_4$ has changed throughout the manuscript. This is not the case, but a result of a change in line and/or page number.

[revised manuscript text omitted]

As an example of sea–ice cover expansion due to the applied glacial–like perturbations, we show the change in sea–ice cover between $Ctrl_{RED}$ and $GLcomb_{RED}$ (Fig. S.4).

The supplementary tables S.1 and S.2 list diagnostic variables for climate (Table S.1, model versions $RED$ and $GAM$, all ensemble members), carbon and nutrients (Table S.2, all model versions, all ensemble members). Table S.3 lists model–data comparison statistics for the $Ctrl$ and $GLcomb$ simulations in each of the model versions $RED$ and $GAM$, and for the two benthic $\delta^{13}C$ data time slices HOL (0-6 ka) and LGM (19-23 ka).

Fig. S.5 shows the non–linear relationship between $\overline{P^*}$ and C/P in model version $GAM$. Note that the complete set of these data are also listed in Table S.1.

[revised manuscript text omitted]